# A Theoretical Framework for Modular Learning of Robust Generative Models

**Corinna Cortes** [1]   **Mehryar Mohri** [1 2]   **Yutao Zhong** [1]

## Abstract

Training large-scale generative models is resource-intensive and relies heavily on heuristic dataset weighting. We address two fundamental questions: Can we train Large Language Models (LLMs) modularly, combining small, domain-specific experts to match monolithic performance, and can we do so robustly for *any* data mixture, eliminating heuristic tuning? We present a theoretical framework for *modular* generative modeling where a set of pre-trained experts are combined via a gating mechanism. We define the space of normalized gating functions $\mathcal{G}_1$ and formulate the problem as a minimax game to find a single robust gate that minimizes divergence to the worst-case data mixture. We prove the existence of such a robust gate using Kakutani's fixed-point theorem and show that modularity acts as a strong regularizer, with generalization bounds scaling with the lightweight gate's complexity. Furthermore, we prove that this modular approach can theoretically outperform models retrained on aggregate data, with the gap characterized by the Jensen-Shannon Divergence. Finally, we introduce a scalable Stochastic Primal-Dual algorithm and a *Structural Distillation* method for efficient inference. Empirical results on synthetic and real-world datasets confirm that our modular architecture effectively mitigates gradient conflict and can robustly outperform monolithic baselines.

## 1. Introduction

Training large-scale generative models, such as Large Language Models (LLMs), is notoriously expensive and often impractical to repeat for every new dataset (Brown et al., 2020; Hoffmann et al., 2022). The computational cost and environmental footprint of these dense models have raised significant sustainability concerns (Strubell et al., 2019; Schwartz et al., 2020). This monolithic paradigm faces two critical challenges. First, *sustainability and adaptability:* can we train LLMs modularly, learning small, accurate models on individual domains (e.g., math, coding) and combining them to match a giant model? If so, training becomes dramatically cheaper and greener; updates require training only a new module and the lightweight combiner, avoiding catastrophic forgetting (Kirkpatrick et al., 2017; Parisi et al., 2019) and enabling the efficient reuse of pretrained experts (Pfeiffer et al., 2023). In the future, privacy regulations and publisher paywalls could also restrict access to data domains, smaller models trained by the data owners could constitute the only viable path to data access. Second, *robustness:* standard training relies on heuristic importance weights across datasets (Gao et al., 2020; Touvron et al., 2023), or static optimization targets (Xie et al., 2023), often failing when test distributions differ from training assumptions (Koh et al., 2021). Can we build a modular LLM that is accurate for *any* mixture of datasets, eliminating heuristic weighting entirely?

We provide an affirmative answer to both questions, offering the first rigorous game-theoretic framework for robust modularity. Unlike heuristic approaches like simple parameter averaging (Model Soups) (Wortsman et al., 2022), task arithmetic (Ilharco et al., 2023), or standard Mixture of Experts which rely on auxiliary load-balancing losses (Shazeer et al., 2017; Fedus et al., 2022), we seek a single system that is robust to *any* arbitrary mixture of the underlying source distributions. We propose a *gated solution*, $\pi_g(x) = \sum_k g(x,k)\widehat{\pi}_k(x)$, where an adaptive gate dynamically reweights frozen experts. Our goal is to find a robust gate $g^*$ that minimizes the divergence to the worst-case data mixture.

**Contributions.** Our main contributions are: (1) *Theoretical Framework:* We define the normalized gate space $\mathcal{G}_1$ and formulate robustness as a minimax game. We prove the existence of a robust gate using Kakutani's fixed-point theorem, establishing a stable upper bound on the worst-case risk (Theorem 3.3). (2) *Generalization Analysis:* We derive bounds showing that sample complexity scales with

---

[1]Google Research, New York, NY; [2]Courant Institute of Mathematical Sciences, New York, NY. Correspondence to: Corinna Cortes <corinna@google.com>, Mehryar Mohri <mohri@google.com>, Yutao Zhong <yutaozhong@google.com>.

*Proceedings of the 43rd International Conference on Machine Learning*, Seoul, South Korea. PMLR 306, 2026. Copyright 2026 by the author(s).

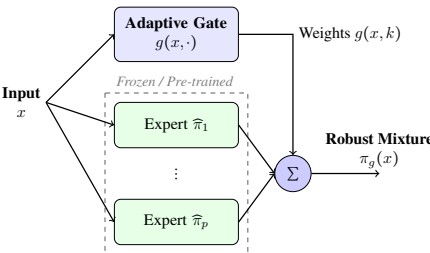

*Figure 1.* Conceptual Architecture of the Modular Gated Solution.

the lightweight gate complexity and the *expert coincidence norm* $C_\Pi$, rather than the massive expert parameters. (3) *Comparison with Retraining:* We prove an information-theoretic bound showing our modular approach can outperform monolithic retraining, with the performance gap characterized by the Jensen-Shannon Divergence (Theorem D.1). (4) *Scalable Algorithm & Inference:* We introduce a Stochastic Primal-Dual algorithm for the constrained game and a *Structural Distillation* method (Hinton et al., 2015) to map the non-causal gate to a causal router for efficient autoregressive inference. (5) *Empirical Validation:* We demonstrate on synthetic benchmarks and real-world datasets (Wikipedia, Code, FineWeb) that our approach mitigates gradient conflict (Yu et al., 2020), outperforming baselines in high-interference regimes.

**Organization.** Section 2 formalizes the problem. Section 3 presents existence proofs and comparisons. Section 4 details the optimization algorithm. Sections 5 and 6 addresses inference and distillation. Section 7 presents empirical results.

**Related Work.** Our framework bridges theoretical Multiple-Source Domain Adaptation (MSA) (Mansour et al., 2008) with modular architectures. While rooted in MSA, we diverge from recent value-based routing (Dann et al., 2025) by addressing the *generative* setting, which introduces the unique mathematical challenge of enforcing global normalization ($Z_g = 1$). Structurally, our approach differs from Mixture of Experts (Fedus et al., 2022) by operating on *frozen* experts; from static merging methods like Model Soups (Wortsman et al., 2022) by using dynamic, input-dependent gating; and from learning to defer (Cortes et al., 2016a) by optimizing for soft probabilistic mixing rather than hard selection. An extended discussion is provided in Appendix A.1.

## 2. Setup & Problem Formulation

Let $D_k$, $k \in [1,p]$, denote $p$ datasets with empirical distributions $\widehat{\mathsf{p}}_k$. We assume access to pre-trained models $\widehat{\pi}_k$ approximating each distribution with guarantees $\mathsf{D}_{\mathrm{KL}}(\widehat{\mathsf{p}}_k \parallel \widehat{\pi}_k) \le \epsilon_k$. We consider *gated solutions* $\pi_g(x) = \sum_{k=1}^p g(x,k)\widehat{\pi}_k(x)$, where $g(x,\cdot) \in \Delta$ is a gating function (see Figure 1). Our goal is to approximate any mixture $\widehat{\mathsf{p}}_\lambda = \sum_{k=1}^p \lambda_k \widehat{\mathsf{p}}_k$ for $\lambda \in \Delta$.

We define the space of *normalized* gating functions $\mathcal{G}_1$ as the subset of gates $g \in \prod_{x \in \mathcal{X}_0} \Delta([1,p])$, that is $g(x,k) \ge 0$ and $\sum_k g(x,k) = 1$ for all $x \in \mathcal{X}_0$, satisfying the global normalization constraint: $\mathcal{G}_1 = \left\{ g : Z_g = \sum_{x \in \mathcal{X}_0} \sum_{k=1}^p g(x,k)\widehat{\pi}_k(x) = 1 \right\}$, where $\mathcal{X}_0 = \bigcup_k \mathrm{supp}(\widehat{\mathsf{p}}_k)$ and is hence finite. For any $g \in \mathcal{G}_1$, the resulting model $\pi_g$ is a valid probability distribution.

**Lemma 2.1.** *The family $\mathcal{G}_1$ is non-empty, compact, and convex.*

See Appendix B.1 for the proof. We seek a single gate $g^* \in \mathcal{G}_1$ robust to the *worst-case* mixture $\lambda \in \Delta$. We formulate this as a minimax game:

$$\min_{g \in \mathcal{G}_1} \max_{\lambda \in \Delta([1,p])} \mathsf{D}_{\mathrm{KL}}(\widehat{\mathsf{p}}_\lambda \parallel \pi_g).$$

We use the relative entropy rather than cross-entropy because the entropy term $H(\widehat{\mathsf{p}}_\lambda)$ varies with the adversarial choice of $\lambda$. Minimizing the worst-case cross-entropy $\max_\lambda \mathbb{E}_{\widehat{\mathsf{p}}_\lambda}[-\log \pi]$ is thus not equivalent to minimizing the divergence $\max_\lambda \mathsf{D}_{\mathrm{KL}}(\widehat{\mathsf{p}}_\lambda \parallel \pi)$, which ensures the model approximates the distribution $\widehat{\mathsf{p}}_\lambda$ itself.

## 3. Theoretical Analysis

We now establish the existence of a robust gate and quantify its advantages over monolithic retraining.

### 3.1. Fixed Mixture & Robust Existence

First, we consider a simple non-adaptive baseline. If we fix the mixture weights $\lambda$, a constant gate $g_\lambda(x,k) = \lambda_k$ (which belongs to $\mathcal{G}_1$) achieves an average error bound.

**Proposition 3.1** (Fixed Mixture Guarantee). *For any fixed $\lambda \in \Delta$, the constant gate $\pi_\lambda = \sum_k \lambda_k \widehat{\pi}_k$ satisfies $\mathsf{D}_{\mathrm{KL}}(\widehat{\mathsf{p}}_\lambda \parallel \pi_\lambda) \le \sum_{k=1}^p \lambda_k \epsilon_k \le \max_k \epsilon_k$.*

See Appendix B.2.1 for the proof. This proposition guarantees average performance for a known mixture $\lambda$ using a simple non-adaptive gate. (We derive the exact, though complex, optimal gate for a fixed $\lambda$ in Appendix B.2.2). However, any static weighting scheme is fundamentally limited by a capacity lower bound of $\log(\sum_k e^{\epsilon_k})$ for disjoint domains, as formalized below.

**Theorem 3.2** (Fundamental Capacity Lower Bound for Static Gating). *Assume the datasets $D_k$ have mutually disjoint supports. Consider the class of static gating functions $\mathcal{G}_{const} \subset \mathcal{G}_1$, defined as gates where $g(x,k) = w_k$ is independent of $x$ for all $k$. For any static gate $\mathbf{w} \in \mathcal{G}_{const}$, the worst-case Kullback-Leibler divergence is lower-bounded by:*

$$\max_{k \in \{1,\dots,p\}} \mathsf{D}_{\mathrm{KL}}(\widehat{\mathsf{p}}_k \parallel \pi_\mathbf{w}) \ge \log\left(\sum_{j=1}^p e^{\epsilon_j}\right).$$

*Proof Sketch.* By the global normalization constraint, any static weight vector $\mathbf{w}$ must lie on the simplex. Because the supports are mutually disjoint, the mixture density simplifies exactly to $\pi_{\mathbf{w}}(x) = w_k \widehat{\pi}_k(x)$ on the support of task $k$. Evaluating the KL divergence yields $D_{\mathrm{KL}}(\widehat{\mathsf{p}}_k \| \pi_{\mathbf{w}}) = \epsilon_k - \log w_k$. Bounding this by the worst-case risk $\delta$ implies $w_k \geq e^{\epsilon_k - \delta}$. Summing over all $p$ experts gives $1 \geq e^{-\delta} \sum_k e^{\epsilon_k}$, which yields the stated bound. See Appendix B.3 for full details. $\square$

To overcome this lower bound barrier and achieve robustness to *unknown* $\lambda$, we require an input-dependent gate.

Next, we show the existence of a gate model with a favorable guarantee for any target mixture $\lambda$. We will use the *linearized game*, which is defined by the payoff $\widetilde{L}(\lambda, g) = \sum_{k=1}^{p} \lambda_k D_{\mathrm{KL}}(\widehat{\mathsf{p}}_k \| \pi_g)$. Our analysis is presented in terms of the *Jensen-Shannon Divergence* (JSD), which is a measure of diversity. For a mixture $\widehat{\mathsf{p}}_\lambda = \sum_k \lambda_k \widehat{\mathsf{p}}_k$, JSD is defined as the average KL divergence from each source to the mixture:

$$D_{\mathrm{JSD}}^{\lambda}(\{\widehat{\mathsf{p}}_k\}) = \sum_{k=1}^{p} \lambda_k D_{\mathrm{KL}}(\widehat{\mathsf{p}}_k \| \widehat{\mathsf{p}}_\lambda).$$

JSD is non-negative and upper bounded by the Shannon entropy $H(\lambda) = -\sum_{k=1}^{p} \lambda_k \log \lambda_k$.

**Theorem 3.3** (Robust Existence). *The linearized modular game admits a saddle point $(\lambda^*, g^*) \in \Delta([1, p]) \times \mathcal{G}_1$. For any mixture $\lambda \in \Delta$, the robust gate $g^*$ satisfies for any $\lambda \in \Delta([1, p])$:*

$$D_{\mathrm{KL}}(\widehat{\mathsf{p}}_\lambda \| \pi_{g^*}) \leq \log\left[\sum_{k=1}^{p} e^{\epsilon_k}\right] - H_\sigma^{\lambda^*}(K|X) - D_{\mathrm{JSD}}^{\lambda}(\{\widehat{\mathsf{p}}_k\}),$$

*where $H_\sigma^{\lambda^*}(K|X) = \sum_{k=1}^{p} \lambda_k^* \mathbb{E}_{x \sim \widehat{\mathsf{p}}_k}\left[-\log \frac{\sigma_k \widehat{\pi}_k(x)}{\pi_\sigma(x)}\right]$ is the target-weighted conditional entropy of the expert assignment under the robust constant gate $\pi_\sigma = \sum_{k=1}^{p} \sigma_k \widehat{\pi}_k$ defined by the softmax weights $\sigma_k = e^{\epsilon_k} / \sum_{j=1}^{p} e^{\epsilon_j}$.*

*Proof Sketch.* The proof relies on constructing a linearized auxiliary game to satisfy Kakutani's fixed-point theorem conditions; See Appendix B.4 for full details. $\square$

Here, $H_\sigma^{\lambda^*}(K|X)$ can be viewed as the *overlap gain* and $D_{\mathrm{JSD}}^{\lambda}$ as the *diversity gain*. The upper bound ($V^* \leq$ LSE $-$ Overlap $-$ Diversity) reveals how the robust gate leverages task geometry in three limiting regimes (Appendix B.5):

**Case 1: The Specialization Limit (Disjoint Experts).** When supports are mutually disjoint, overlap vanishes ($H_\sigma^{\lambda^*} = 0$) and diversity is maximal ($D_{\mathrm{JSD}}^{\lambda} = H(\lambda)$). For balanced tasks, this diversity gain cancels the capacity cost ($\log p$), guaranteeing minimax robustness: $V^* \leq \max_k \epsilon_k$.

**Case 2: The Redundancy Limit (Identical Experts).** When experts are identical with error $\epsilon$, diversity vanishes

($D_{\mathrm{JSD}} = 0$) but overlap is maximal ($H(\sigma) = \log p$). This overlap gain exactly refunds the capacity cost ($\log(p e^\epsilon) = \epsilon + \log p$), recovering the single-expert performance: $V^* \leq \epsilon$.

**Case 3: The Ensemble Mechanism (High Overlap).** For overlapping experts with distinct errors, the overlap gain becomes the entropy of the robust weights $H(\sigma)$. Substituting $H(\sigma) = \log Z - \sum_k \sigma_k \epsilon_k$ into the bound proves an exact cancellation of the capacity cost (LogSumExp). The bound collapses to the *weighted average error* $V^* \leq \sum \sigma_k \epsilon_k$, effectively acting as a static ensemble to minimize risk.

**Prior Knowledge on Mixture Weights.** In certain applications, we may have prior knowledge suggesting that the mixture weights encountered at test time will be restricted to a convex subset $\Lambda \subset \Delta([1, p])$. This knowledge can be leveraged to derive a more specialized solution with significantly stronger performance guarantees. Appendix C.1 gives a detailed analysis of this setting and its benefits.

**The Least-Favorable Mixture.** The optimization also yields the adversary's optimal strategy $\lambda^*$ for the linearized game $\widetilde{L}$. This vector identifies the specific mixture weights that maximize the weighted expert loss. In practical scenarios where a modular architecture is infeasible (e.g., extreme latency constraints), $\lambda^*$ provides a statistically principled target distribution for training a single static model, eliminating the need for heuristic weighting (see Appendix E.3).

## 3.2. Comparison with Monolithic Baselines: Interference vs. Decoupling

We rigorously contrast the proposed Modular Robustness with standard Monolithic training by analyzing how each architecture interacts with the geometry of the task distributions. See Appendix D for a more detailed comparison.

**The Monolithic Barrier: Diversity as Interference.** Consider a monolithic model $\pi_{\mathrm{mono}}$ trained to minimize the loss on the mixture $\widehat{\mathsf{p}}_\lambda = \sum_{k=1}^{p} \lambda_k \widehat{\mathsf{p}}_k$. The performance on individual tasks is governed by the *Jensen-Shannon Decomposition Identity*. For any model $\pi$, the average task risk decomposes exactly into two terms:

$$\underbrace{\sum_{k=1}^{p} \lambda_k D_{\mathrm{KL}}(\widehat{\mathsf{p}}_k \| \pi)}_{\text{Average Task Risk}} = \underbrace{D_{\mathrm{KL}}(\widehat{\mathsf{p}}_\lambda \| \pi)}_{\text{Mixture Fit}} + \underbrace{D_{\mathrm{JSD}}^{\lambda}(\widehat{\mathsf{p}}_1, \ldots, \widehat{\mathsf{p}}_p)}_{\text{Interference}}.$$

This equality reveals a fundamental limitation. Even in the limit of infinite capacity where the model fits the global mixture perfectly ($D_{\mathrm{KL}}(\widehat{\mathsf{p}}_\lambda \| \pi) \to 0$), the average risk is strictly determined by the task diversity, measured by the Jensen-Shannon Divergence ($D_{\mathrm{JSD}}$). We formalize this in Theorem D.1 (Appendix D.2). Since the average task risk is lower-bounded by $D_{\mathrm{JSD}}^{\lambda}$, the worst-case domain loss $\max_k D_{\mathrm{KL}}(\widehat{\mathsf{p}}_k \| \pi)$ is also necessarily lower-bounded by this quantity. For a monolithic architecture, diversity manifests

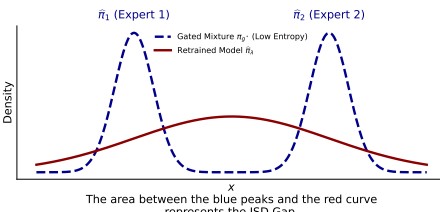

*Figure 2.* Visualizing the JSD Gap. A gated model (blue) fits distinct modes perfectly by routing inputs. A retrained model (red) suffers from capacity interference, forcing an entropy increase proportional to the JSD.

as *Geometric Interference*: the model is forced to collapse distinct distributions into a single centroid. Consequently, performance is dominated by the geometry of the problem rather than the difficulty of the tasks; even if tasks are trivial to solve individually ($\epsilon_k \approx 0$), the model fails if they are distinct ($D_{\mathrm{JSD}} \gg 0$).

**The Modular Advantage: Diversity as Separability.** In contrast, the modular gating network effectively inverts this relationship. The worst-case risk of the robust gate is bounded by (Theorem 3.3):

$$\mathrm{Risk}_{\mathrm{mod}} \le \underbrace{\log\left(\sum e^{\epsilon_k}\right)}_{\text{Capacity Cost}} - \underbrace{D_{\mathrm{JSD}}^\lambda(\widehat{\mathsf{p}}_1,\dots,\widehat{\mathsf{p}}_p)}_{\text{Diversity Gain}} - \underbrace{H_\sigma^{\lambda^*}(K|X)}_{\text{Overlap}}.$$

Here, the divergence term appears with a *negative* sign. For the modular system, task diversity acts as a *Geometric Gain*. In the high-diversity regime (disjoint supports), the overlap vanishes ($H \approx 0$). Crucially, if the test mixture is diverse, the separability gain becomes maximal ($D_{\mathrm{JSD}}^\lambda \to H(\lambda)$). As shown in our geometric analysis (Section 3.1), this gain effectively cancels the entropy term in the capacity cost ($\log(\sum e^{\epsilon_k}) \approx \epsilon_{\max} + H(\lambda)$). Consequently, for diverse mixtures, the bound simplifies to the intrinsic error of the worst-case expert: $D_{\mathrm{KL}}(\widehat{\mathsf{p}}_\lambda \| \pi_{g^*}) \le \max_k \epsilon_k$.

**The Decoupling Hypothesis.** This analysis identifies a structural phase transition governed by data geometry. We observe a *Symmetric Divergence Effect*: the same quantity $D_{\mathrm{JSD}}^\lambda$ that acts as an interference penalty for the monolithic model acts as a separability bonus for the modular gate. In the *High Diversity Regime* (large $D_{\mathrm{JSD}}^\lambda$), the Monolithic model hits an interference floor (Risk $\ge D_{\mathrm{JSD}}^\lambda$), forced to increase entropy to cover disjoint supports (broad red curve in Figure 2). In contrast, the Modular model exploits this separation to cancel the capacity cost, maintaining the low-entropy precision of the original experts (sharp blue peaks). By effectively *decoupling* risk from geometry, the modular system is guaranteed to outperform the monolithic baseline on *any* mixture $\lambda$ where the intrinsic task difficulty is lower than the geometric cost of mixing: $\max_k(\epsilon_k) < D_{\mathrm{JSD}}^\lambda(\widehat{\mathsf{p}})$.

**Safety in Convex Settings.** Finally, one might ask if modularity sacrifices performance in simpler settings. We prove that in convex settings, the answer is no. We assume $\Pi$ is a

linear model family (e.g., exponential families) in the sense of Csiszár (1975).

**Theorem 3.4** (Gated Model Coincides with Retraining). *Let $\Pi$ be a linear model. For a mixture $\widehat{\mathsf{p}}_\lambda$, let $\pi_k = \pi^*(\widehat{\mathsf{p}}_k)$ denote the projection of each component. Then, the best model trained on the mixture $\pi^*(\widehat{\mathsf{p}}_\lambda)$ coincides exactly with the gated mixture of the best component models:*

$$\pi^*(\widehat{\mathsf{p}}_\lambda) = \sum_{k=1}^p \lambda_k \pi_k.$$

*Proof Sketch.* The proof leverages the Pythagorean equality for linear models (Csiszár, 1975). By expanding the component divergences, we show that the aggregate mixture risk $D_{\mathrm{KL}}(\widehat{\mathsf{p}}_\lambda \| \pi)$ and the expected individual risk $\sum_k \lambda_k D_{\mathrm{KL}}(\widehat{\mathsf{p}}_k \| \pi)$ differ only by a constant (the Jensen-Shannon Divergence). Since they share the same unique minimizer, the optimal monolithic model exactly equals the convex combination of the individually projected experts. Full details are provided in Appendix D.3. □

Thus, modularity acts as a "safe" architectural prior: it loses nothing in convex regimes while providing strictly superior robustness guarantees in the presence of conflicting, non-convex settings.

### 3.3. Generalization and Sample Efficiency

The guarantees in Theorem 3.3 establish the existence of a robust gate on the empirical distributions $\widehat{\mathsf{p}}_k$. A critical advantage of the modular framework is that this robustness transfers efficiently to the true population distributions $\mathsf{p}_k$. The generalization gap of the gated model scales with the complexity of the lightweight gating network $\mathcal{G}_1$, rather than the massive complexity of the generative experts. To formalize this, we rely on the vector-valued Rademacher complexity of the gate class and also require expert log-likelihoods to be bounded (Assumption D.6). The formal assumption and definitions are provided in Appendix D.4.

**Theorem 3.5** (Generalization Bound for Modular Gate Models). *Under Assumption D.6, for any $\delta > 0$, with probability at least $1 - \delta$ over the draw of samples $S_k \sim \mathsf{p}_k^m$, the following inequality holds simultaneously for all $g \in \mathcal{G}_1$ and $\lambda \in \Delta$: for the generalization gap $\Gamma = \mathbb{E}_{x \sim \mathsf{p}_\lambda}[-\log \pi_g(x)] - \mathbb{E}_{x \sim \widehat{\mathsf{p}}_\lambda}[-\log \pi_g(x)]$: $\Gamma \le 2\sqrt{2} C_\Pi e^M \mathfrak{R}_m(\mathcal{G}_1) + M\sqrt{\frac{\log(p/\delta)}{2m}}$, where $\mathfrak{R}_m(\mathcal{G}_1)$ is the Rademacher complexity of the gate class and $C_\Pi = \sup_x \|(\widehat{\pi}_1(x),\dots,\widehat{\pi}_p(x))\|_2$ is the expert coincidence norm.*

See Appendix D.4 for the proof using vector contraction inequalities. The term $C_\Pi$ acts as a condition number for modularity, measuring expert overlap: (1) Specialized Experts (Ideal): If experts have disjoint supports, $\|\widehat{\pi}(x)\|_2 \approx 1$,

so $C_\Pi \approx 1$. (2) Redundant Experts (Worst Case): If experts are identical, $C_\Pi = \sqrt{p}$. Crucially, since the gate is typically a lightweight network (e.g., a shallow Transformer) compared to the massive experts (LLMs), $\mathfrak{R}_m(\mathcal{G}_1) \ll \mathfrak{R}_m(\Pi)$. This implies that the modular approach requires significantly fewer samples to learn a robust policy than retraining a monolithic model.

# 4. Optimization Algorithms

The existence result (Theorem 3.3) guarantees a robust gate $g^*$ but is non-constructive. To compute this gate, we must solve the minimax game. Originally, we formulated the problem as $\min_{g \in \mathcal{G}_1} \max_{\lambda \in \Delta} L(\lambda, g)$ with payoff $L(\lambda, g) = D_{\mathrm{KL}}(\widehat{\mathsf{p}}_\lambda \parallel \pi_g)$. This payoff is convex in both parameters (since $\widehat{\mathsf{p}}_\lambda$ is linear in $\lambda$ and $D_{\mathrm{KL}}$ is convex in first argument), preventing the direct application of standard descent-ascent guarantees. See Appendix E for a more detailed discussion.

## 4.1. Reformulation via Linearization

To derive a tractable algorithm, we reformulate the problem into an equivalent convex-concave game. Since the function $\lambda \mapsto L(\lambda, g)$ is convex, its maximum over the simplex $\Delta$ is always achieved at a vertex. Thus, $\max_\lambda L(\lambda, g) = \max_k D_{\mathrm{KL}}(\widehat{\mathsf{p}}_k \parallel \pi_g)$. This observation allows us to introduce a linearized payoff function:

$$\widetilde{L}(\lambda, g) = \sum_{k=1}^p \lambda_k D_{\mathrm{KL}}(\widehat{\mathsf{p}}_k \parallel \pi_g).$$

This new game shares the same value as the original problem but is *convex-concave*: linear in $\lambda$ and convex in $g$. This structure allows us to apply standard no-regret dynamics. Specifically, if the $\lambda$-player uses Exponentiated Gradient and the $g$-player uses Online Gradient Descent, the system is guaranteed to converge (Algorithm 1 in Appendix E.1).

**Theorem 4.1** (Convergence of Dynamics). *Let $\widetilde{V} = \min_{g \in \mathcal{G}_1} \max_{\lambda \in \Delta} \widetilde{L}(\lambda, g)$. If the projection $\Pi_{\mathcal{G}_1}$ onto the normalized gate space can be computed, then with step sizes $\eta_\lambda \propto 1/\sqrt{T}$ and $\eta_g \propto 1/\sqrt{T}$, the time-averaged gate $\overline{g}_T$ converges to the optimal robust solution:*

$$\max_{\lambda \in \Delta} \widetilde{L}(\lambda, \overline{g}_T) - \widetilde{V} \le O\left(\sqrt{\frac{\log p}{T}}\right).$$

See Appendix E.2 for the proof. This theorem provides a solid theoretical foundation: if we could enforce the constraints exactly, we would provably find the robust gate. However, the projection $\Pi_{\mathcal{G}_1}$ is computationally intractable for large sequence models because the global normalization constraint $Z_g = \sum_{x \in \mathcal{X}_0} \pi_g(x) = 1$ couples the updates across all inputs $x$ in the support.

## 4.2. Scalable Primal-Dual Algorithm

To scale to large generative models, we parameterize the gate $g_\theta$ (e.g., as a Transformer Encoder) and enforce the global constraint via Lagrangian relaxation. We introduce a dual variable $\mu \in \mathbb{R}$ corresponding to the equality constraint $Z_g = 1$, transforming the problem into a *3-player primal-dual game*:

$$\min_\theta \max_{\lambda \in \Delta, \mu \in \mathbb{R}} \mathcal{L}(\theta, \lambda, \mu) = \underbrace{\sum_{k=1}^p \lambda_k \mathcal{L}_{\mathrm{NLL}}(k, \theta)}_{\text{Robust NLL}} + \underbrace{\mu(Z_{g_\theta} - 1)}_{\text{Penalty}}.$$

The system simulates dynamics between three players:

**1. $\lambda$-player (Adversary):** Maximizes the mixture difficulty using Exponentiated Gradient. This effectively upweights experts where the gate is currently underperforming.

**2. $\mu$-player (Constraint):** Performs Dual Ascent to enforce global normalization. If the total mass $Z_{g_\theta} > 1$, $\mu$ increases, penalizing the gate; if $Z_{g_\theta} < 1$, $\mu$ decreases.

**3. $g$-player (Gate):** Updates parameters $\theta$ to minimize the Lagrangian via AdamW.

We solve this using stochastic estimates, (see Algorithm 2 in Appendix E.4) and a detailed description in Algorithm 3 (Appendix F.2).

## 4.3. Efficiency and Convergence

The Primal-Dual formulation fundamentally alters the computational profile of the problem, making it feasible for LLMs.

**Optimization Complexity.** The standard projection onto $\mathcal{G}_1$ requires solving a constrained quadratic program over the entire support $\mathcal{X}_0$, which is impossible for sequence models. In contrast, our constraint is enforced via a scalar update $\mu$, costing $O(1)$ per parameter. The partition function $Z$ is estimated efficiently using the training batch itself as the importance sampling proposal, avoiding auxiliary data generation.

**Theoretical Guarantee.** Crucially, replacing the hard projection with a Lagrangian penalty does not sacrifice the convergence guarantee. The Primal-Dual dynamics approximate the solution to the constrained game with the same asymptotic rate.

**Theorem 4.2** (Convergence of Primal-Dual Dynamics). *Consider the Lagrangian payoff $\mathcal{L}(g, \lambda, \mu) = \widetilde{L}(\lambda, g) + \mu(Z_g - 1)$. Under the same convexity assumptions as Theorem 4.1, the time-averaged iterates $(\overline{g}_T, \overline{\lambda}_T)$ generated by Algorithm 2 converge to the optimal robust solution with error $O(1/\sqrt{T})$, and the constraint violation decays at rate*

$O(1/\sqrt{T})$:

$$\max_\lambda \widetilde{L}(\lambda, \overline{g}_T) - \widetilde{V} \le O\left(\frac{1}{\sqrt{T}}\right) \quad and \quad |Z_{\overline{g}_T} - 1| \le O\left(\frac{1}{\sqrt{T}}\right).$$

*Proof sketch.* The proof follows from standard regret analysis for Lagrangian games (e.g., (Cherukuri et al., 2017)). The total regret of the primal-dual system decomposes into the regret of the $\lambda$-player (simplex), the $g$-player (convex set), and the $\mu$-player (unconstrained linear). Since all three use no-regret algorithms (Exponentiated Gradient, Adam/OGD, and Gradient Ascent), the average duality gap and the constraint violation are bounded by the average regret, which scales as $O(1/\sqrt{T})$. $\square$

The proof details are given in Appendix E.4. This theorem ensures that even without the expensive projection step, the algorithm provably recovers the robust modular gate.

**Gap between Theory and Practice.** The convergence guarantees provided in Theorem 4.1 and Theorem 4.2 rely on the convexity of the optimization problem with respect to the gate $g$. In our scalable implementation, the gate $g_\theta$ is parameterized by a deep neural network, rendering the objective non-convex with respect to $\theta$. While strict no-regret guarantees do not apply to this non-convex setting, we empirically observe that the Primal-Dual algorithm converges to effective robust solutions, consistent with the success of similar game-theoretic optimization dynamics in deep learning (e.g., GANs or adversarial training).

### 4.4. Practical Implementation

Translating the theoretical algorithm into a stable training loop requires addressing two specific numerical challenges: estimating the global partition function $Z_g$ and avoiding underflow. See Appendix F for a more detailed discussion.

**Estimating the Partition Function.** Calculating the global sum $Z_g = \sum_{x \in \mathcal{X}_0} \pi_g(x)$ exactly is intractable. We rely on a Monte Carlo estimate using the current training batch $B$. Crucially, to avoid expensive auxiliary sampling, we use the training batch itself as the proposal distribution for Importance Sampling. The batch is constructed by sampling uniformly from the $p$ source datasets, effectively drawing $x \sim \frac{1}{p}\sum_k \widehat{p}_k$. Under the assumption that experts approximate their sources ($\widehat{\pi}_k \approx \widehat{p}_k$), the empirical mixture closely matches the model mixture proposal $q(x) = \frac{1}{p}\sum_k \widehat{\pi}_k(x)$. The estimator becomes: $\widehat{Z} = \frac{1}{|B|}\sum_{x \in B}\frac{\pi_g(x)}{q(x)}$. This allows us to reuse the logits computed during the forward pass, estimating the global constraint with zero additional inference cost. To reduce variance in the $\mu$-update, we track $\widehat{Z}$ using an Exponential Moving Average (EMA).

**Log-Space Stability.** The mixture probability $\pi_g(x) = \sum_k g(x,k)\widehat{\pi}_k(x)$ involves summing probabilities that may

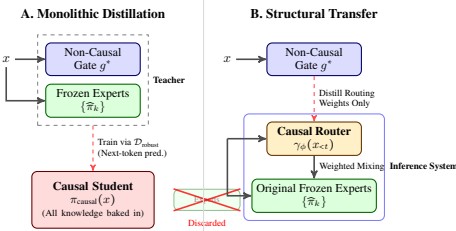

*Figure 3.* Efficiency Strategies. (A) Monolithic Distillation trains a single large model to mimic the ensemble, discarding the original experts. (B) Structural Distillation trains a lightweight Causal Router to mimic only the gating decisions ($g^*$), preserving the original experts. This maintains modularity: upgrading an expert in (B) improves the system immediately.

be extremely small (e.g., $10^{-100}$ for long sequences). Direct computation leads to catastrophic underflow. We strictly perform all operations in log-space using the LogSumExp trick: $\log \pi_g(x) = \text{LogSumExp}_k(\log g(x,k) + \log \widehat{\pi}_k(x))$.

**Quadratic Penalty.** An alternative to the Lagrangian method is to relax the hard constraint into a soft quadratic penalty, $\min_g \max_\lambda L'(\lambda, g) + \beta(Z_g - 1)^2$. This eliminates the need for the $\mu$-player, reducing the problem to a standard regularized minimax optimization. However, it only guarantees approximate normalization. We find the Primal-Dual approach superior as it dynamically adjusts the penalty strength $\mu$ to satisfy the constraint exactly in the limit.

## 5. Sampling from the Robust Gated Model

The optimization procedure yields a robust gate $g^* \in \mathcal{G}_1$ that guarantees the mixture model $\pi_{g^*}(x) = \sum_k g^*(x,k)\widehat{\pi}_k(x)$ is globally normalized. However, sampling from this model presents a unique challenge: the optimal gate $g^*(x,\cdot)$ is *non-causal*. It determines the mixture weights based on the *complete* sequence $x$, meaning the probability of the first token theoretically depends on the last. This breaks the standard autoregressive property required for efficient token-by-token generation. To sample from $\pi_{g^*}$, we must rely on methods that treat the model as an unnormalized density or a re-weighted approximation. We explore two Monte Carlo strategies:

**Sampling-Importance-Resampling (SIR).** This method generates samples asymptotically from the target distribution. We first draw $N$ candidate sequences $x^{(1)}, \ldots, x^{(N)}$ from a proposal distribution $q(x)$ that is easy to sample from—specifically, the uniform mixture of experts $q(x) = \frac{1}{p}\sum_k \widehat{\pi}_k(x)$. We then compute importance weights $w_i = \pi_{g^*}(x^{(i)})/q(x^{(i)})$. Finally, we resample a single sequence $x^*$ from the candidates with probability proportional to $w_i$. As $N \to \infty$, $x^*$ converges to a draw from $\pi_{g^*}$. For full details, see Appendix F.4.1.

**Exact Rejection Sampling.** Unlike SIR, Rejection Sampling provides *exact* samples from $\pi_{g^*}$ for any finite $N$. This

requires an envelope constant $M$ such that $\pi_{g^*}(x) \leq Mq(x)$ for all $x$. A crucial property of our normalized gate space $\mathcal{G}_1$ is that $g^*(x, k) \leq 1$, which implies: $\pi_{g^*}(x) = \sum_{k=1}^{p} g^*(x, k)\widehat{\pi}_k(x) \leq \sum_{k=1}^{p} \widehat{\pi}_k(x) = p \cdot q(x)$. Thus, we can use the number of experts $M = p$ as a strict envelope. The algorithm samples a candidate $x \sim q(x)$ and accepts it with probability $A(x) = \pi_{g^*}(x)/(p \cdot q(x))$. The acceptance rate is $1/p$, meaning we must generate approximately $p$ candidates to obtain one valid sample (see Appendix F.4.2).

**Inference Bottleneck.** While these methods preserve the theoretical robustness guarantees, their inference cost scales linearly with the number of experts $p$. Evaluating the acceptance probability or importance weight requires running a forward pass on *all* $p$ experts for every candidate sequence. For large ensembles (e.g., $p = 100$), this cost is prohibitive for real applications, motivating the need for distillation.

## 6. Efficient Inference: Structural Distillation

While the robust gate $g^*$ guarantees optimal performance, its non-causal nature requires expensive sampling methods like SIR or Rejection Sampling (Section 5) during inference. With *Structural Distillation* we recover efficient $O(1)$ autoregressive generation while preserving the benefits of modularity.

### 6.1. Monolithic vs. Structural Distillation

Standard distillation would involve training a single large student model to mimic the input-output behavior of the ensemble $\pi_{g^*}$ (see Appendix F.5). While efficient at inference time, this *Monolithic Distillation* (Figure 3(A)) discards the modular structure: if one expert is updated or a new domain is added, the entire student model must be retrained from scratch. In contrast, our *Structural Distillation* approach (Figure 3(B)) preserves the pre-trained experts (see detailed analysis in Appendix F.6.2). We distill the robust, non-causal gate $g^*$ into a lightweight *Causal Router* $\gamma_\phi$. The inference system remains a mixture of experts, but the routing decisions are now made causally.

### 6.2. The Causal Router & Objective

We define the student model $\pi_\gamma$ as a causal mixture of the frozen experts, parameterized by a learnable router $\gamma_\phi$: $\pi_\gamma(x) = \prod_{t=1}^{T} \pi_\gamma(x_t \mid x_{<t}) = \prod_{t=1}^{T} \sum_{k=1}^{p} \gamma_\phi(x_{<t}, k)\widehat{\pi}_k(x_t \mid x_{<t})$. Here, $\gamma_\phi(x_{<t}, \cdot) \in \Delta$ is a distribution over experts predicted by a small causal network (e.g., a shallow Transformer) given only the history. Our goal is to train $\phi$ to minimize the KL divergence from the robust teacher $\pi_{g^*}$ to the student $\pi_\gamma$ over the sequence space $\mathcal{X}$:

$$\min_\phi \mathcal{J}(\phi) = \mathrm{D}_{\mathrm{KL}}(\pi_{g^*} \parallel \pi_\gamma) = \mathbb{E}_{x \sim \pi_{g^*}}\left[\log \frac{\pi_{g^*}(x)}{\pi_\gamma(x)}\right].$$

Crucially, this global sequence-level objective decomposes into a tractable token-level optimization.

**Proposition 6.1** (Decomposition of Structural Distillation)**.** *Minimizing the sequence-level divergence $\mathrm{D}_{\mathrm{KL}}(\pi_{g^*} \parallel \pi_\gamma)$ is equivalent to maximizing the expected log-likelihood of the student model on trajectories sampled from the robust teacher. Specifically, the gradient is:*

$$\nabla_\phi \mathcal{J}(\phi) = -\mathbb{E}_{x \sim \pi_{g^*}}\left[\sum_{t=1}^{T} \nabla_\phi \log\left(\sum_{k=1}^{p} \gamma_\phi(x_{<t}, k)\widehat{\pi}_k(x_t \mid x_{<t})\right)\right].$$

See Appendix F.6.1 for the proof. This result allows us to train the router using standard MLE on a dataset of "robust sequences" generated by the teacher (using Rejection Sampling). Furthermore, we prove in Theorem F.2 (Appendix F.6.2) that this objective minimizes the *Router Approximation Error*, with no irreducible structural mismatch.

### 6.3. Cached-Logit Distillation Algorithm

A naive gradient update requires evaluating all $p$ experts at every step. To avoid this bottleneck, we exploit the fact that experts are frozen and propose a *Cached-Logit* training loop (Algorithm 6 in Appendix F.7). First, we generate a dataset $\mathcal{D}$ from $\pi_{g^*}$ using Rejection Sampling or SIR, caching the expert probability vectors $\mathbf{P}_t = [\widehat{\pi}_1(x_t|x_{<t}), \ldots, \widehat{\pi}_p(x_t|x_{<t})]$ for every token. While generating this robust synthetic dataset requires evaluating all experts, it is strictly a one-time, offline process that is highly parallelizable across multiple GPUs. As established by our generalization bound (Theorem 3.5), the sample complexity scales with the lightweight Causal Router rather than the complex experts, meaning we only need to generate a relatively small dataset to distill the robust policy. Second, we train the router $\phi$ iteratively to maximize the likelihood of these cached sequences by minimizing $\mathcal{L} = -\log(\gamma_\phi(x_{<t}) \cdot \mathbf{P}_t)$. This decouples the expensive expert evaluation (one-time cost) from router training and bypasses the need for repeated, expensive forward passes through the experts during the distillation optimization loop, yielding a system that is robust, modular, and efficient.

## 7. Experiments

### 7.1. Empirical Comparison: Gate vs. Monolithic

In this section, we compare our gated model against standard retrained baselines on synthetic data. Before presenting the results, we discuss the nuances of a fair comparison.

**Fairness and Gradient Conflict.** Comparing a modular architecture (frozen experts) with a monolithic model (trained from scratch) is non-trivial. Standard metrics like parameter count are insufficient. While a gated model might have a larger *total* parameter count, its effective hypothesis space is constrained to the convex hull of the experts. A retrained

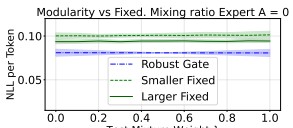 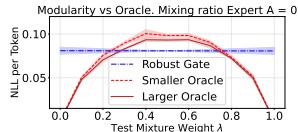

*Figure 4.* **Modularity overcomes gradient conflict.** Left: The Robust Gate (blue) outperforms Fixed monolithic baselines (green) across all mixtures. Right: Even compared to Oracle models (red) trained on the exact test distribution, the Gate wins in the high-interference region ($\lambda \approx 0.5$). The concave shape of the Oracle curves indicates destructive interference.

monolithic model theoretically enjoys greater flexibility, as it can move freely in weight space to minimize aggregate loss. However, this flexibility comes at a cost: *gradient conflict*. When source distributions contain conflicting signals (e.g., distinct tasks or contradictory rules), a single model trained on the aggregate objective suffers from destructive interference. The optimization settles for a high-entropy *compromise* that underperforms on individual components. Our modular architecture structurally orthogonalizes these conflicts. Therefore, we frame our comparison not just on capacity, but on *robustness to distribution shift*.

**Experimental Protocol.** We define $p = 2$ experts ($N_k$ parameters each), pre-trained to convergence on source domains $D_k$. We evaluate three model classes:

**Robust Gate Model (Ours):** Combines frozen experts via a gate trained with the Primal-Dual Algorithm (Algorithm 3 in Appendix F.2) on the union dataset. The total size is $N_{\text{gate}} \approx 1.24 \sum N_k$ (only 24% trainable parameters).

**Retrained Model (Fixed $\lambda$):** Monolithic models trained on the aggregate data (fixed $\lambda = 0.5$). We evaluate a *Smaller* version ($N = \sum N_k$) and a *Larger* version ($N = 1.5 \sum N_k$) to test if increased capacity overcomes interference.

**Oracle Model:** "Cheating" baselines retrained from scratch on the *exact* test mixture $\lambda$ for every evaluation point. We again test Smaller and Larger variants.

**Synthetic Verification.** We define a sequence task over vocabulary $V = 100$ with two contradictory domains. Domain A follows $x_{t+1} = (x_t + 1) \pmod{100}$, while Domain B follows $x_{t+1} = (x_t - 1) \pmod{100}$, with $x_0$ chosen uniformly at random. For any token $x_t$, the gradients from A and B are directly opposed. We evaluate performance by varying the test mixture $\lambda \in [0, 1]$ in steps of 0.1. For experimental details, see Appendix G.2.

**Results: The Interference Gap.** Figure 4 (Left) compares the Robust Gate against the Fixed baselines. The Fixed models, trained on the conflict-heavy mixture ($\lambda = 0.5$), learn a high-entropy policy that fails to specialize for either domain. The Gate achieves consistently lower loss, confirming that modularity is superior to ERM when tasks are disjoint.

Figure 4 (Right) reveals a more profound insight: the *In-*

*terference Gap.* In the high-entropy region ($\lambda \in [0.3, 0.7]$), the Gate outperforms even the "cheating" Larger Oracle. This empirically validates Theorem D.1 (The JSD Gap). The Oracle's performance curve is distinctly concave: even with perfect knowledge of $\lambda$, a single set of weights cannot simultaneously master contradictory rules without increasing entropy (divergence). The modular system avoids this penalty because the experts remain disjoint, and the gate simply routes queries to the correct specialist. At the extremes ($\lambda \approx 0$ or $1$), the task collapses to a single domain, allowing the Oracles to specialize and naturally surpass the Gate. Appendix G.1 provides additional results where the distributions of two experts A and B were less contradictory.

## 7.2. Algorithm Stability and Convergence

A key concern with minimax optimization is stability. We monitored the dynamics of the Primal-Dual variables during training. The adversary's mixture weights $\lambda_t$ rapidly converged to $\lambda \approx [0.5, 0.5]$. This indicates that the gate successfully balanced the performance across domains ($\epsilon_A \approx \epsilon_B$), reaching a maximum-entropy equilibrium where the adversary has no incentive to concentrate on a specific task. Simultaneously, the dual variable $\mu_t$, initialized at 0, increased steadily during the first epoch as the gate initialization ($Z \approx 1/p$) violated the constraint, before stabilizing once the gate learned to satisfy the partition unity $Z_g \approx 1$. The system exhibited stable convergence without the oscillations typical of adversarial training, likely due to the convexity of the inner maximization over $\lambda$.

## 7.3. Experiments with Structural Distillation

We also evaluate the Structured Distillation algorithm (Section 6), which distills the robust mixture $\pi_{g^*}$ into a *Causal Router* $\gamma_\phi$. We sampled 5,000 sequences from $\pi_{g^*}$ using rejection sampling. About 3% of these sequences contained inversions (switching between rules), reflecting the nontrivial nature of the task. We trained a causal Transformer router with $d = 10$ (matching the parameter count of the Larger Fixed Retrained baseline) on these sequences.

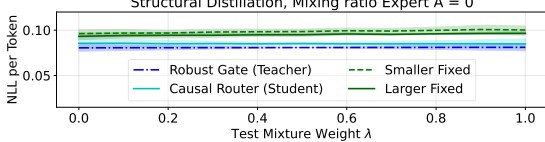

*Figure 5.* **Structured Distillation.** The Causal Router (cyan) effectively recovers the performance of the non-causal Robust Gate (blue), outperforming the Larger Fixed baseline (green).

As shown in Figure 5, the distilled Causal Router (cyan) performs nearly identically to the optimal non-causal Robust Gate (blue), losing minimal performance despite the architectural constraint. It significantly outperforms the monolithic Larger Fixed model (green), demonstrating that

we can transfer the robustness benefits into an efficient autoregressive form.

## 7.4. Modularity for Real-World Data and LLM Scaling

Finally, we experimented with three distinct HuggingFace datasets to test real-world transfer: (1) `wikimedia/wikipedia`: High-quality factual prose; (2) `bigcode/the-stack-smol`: Source code across 30+ languages; (3) `fineweb-edu`: Filtered high-quality educational web content. While Wikipedia and FineWeb share domain characteristics, the Code dataset represents a significant distribution shift. Merging code (strict syntax, high repetition) with natural language (ambiguous, fluid) is known to cause negative transfer in monolithic models.

**Small-Scale Experiments.** We trained 3 experts and a lightweight gate with a combined ~20M parameters. We compared this against a monolithic Retrained model of matching size (19.8M, 3 layers). To address standard architectural baselines, we also evaluated a token-level Mixture-of-Experts (MoE) trained from scratch on the uniform mixture of datasets, using shared attention layers and a token-level routing layer over three expert Feed-Forward Networks, guided by a standard load-balancing loss, matching the capacity (~20M) of the combined modular framework. For additional experimental details, see Appendix G.3.

*Table 1.* NLL per token on Real-World Data.

| Model | NLL (Diff seed) | NLL (Diff data) |
|---|---|---|
| Wiki Expert | 5.122 ± 0.005 | 5.118 ± 0.011 |
| Code Expert | 4.722 ± 0.045 | 5.267 ± 0.788 |
| FineWeb Expert | 5.623 ± 0.004 | 5.623 ± 0.006 |
| Retrained Model | 5.133 ± 0.010 | 5.306 ± 0.257 |
| Standard MoE (Scratch) | 5.080 ± 0.009 | 5.245 ± 0.269 |
| **Gate Model (Ours)** | **4.994** ± 0.013 | **5.087** ± 0.141 |

Table 1 details the results (averages over 5 runs) across two settings: varying initialization seeds and varying data splits. In both cases, the Gate Model outperforms the Retrained model and the Standard MoE model. As theoretically predicted in Section 3.2, joint training over conflicting domains induces gradient interference even in standard MoE architectures due to their shared representation layers and joint optimization. In contrast, our frozen-expert routing approach effectively mitigated this conflict.

Regarding the domain-specific variance, the massive variance of 0.788 stemming from using different training data splits, the "Diff data" column, corresponds to the Code Expert (while the FineWeb Expert exhibited a very low variance of 0.006). This is empirically expected: code datasets are highly heterogeneous, with significant structural and syntactic differences across varying programming languages. Consequently, evaluating the Code Expert across different unseen splits naturally leads to a more volatile NLL. Cru-

cially, despite this inherent instability in the monolithic code domain, our Gate Model robustly gates inputs across experts, smoothing out these domain-specific instabilities and dropping the total variance substantially to 0.141. Keeping the training data fixed but varying the seed for the model training, the "Diff seed" column, exhibits much smaller variations across models.

This result is significant: even with a small-scale experiment (20M parameters), the modular approach navigates the conflict between code and natural language better than a monolithic model trained on the union. This validates our hypothesis that structural modularity acts as a regularizer against negative transfer in real-world deployments. Note that there is no guarantee that a modular model will always outperform a model retrained on all the data. As stated in Theorem D.1, the JSD gap governs the relative performance.

**Scaling to Massive LLMs (1B+ parameters).** To confirm that our theoretical framework scales to modern massive LLMs, we scaled our experiments to an ensemble of massive experts (~1.6B parameters each, GPT-2 XL-style configuration with 48 layers and 1600 embedding dimension) pre-trained on different domains. To ensure a strictly fair comparison, the retrained monolithic baseline model was symmetrically scaled to a massive 4.9B parameter architecture (48 layers, 2832 embedding dimension) to exactly match the combined capacity of the three experts plus the gate model. The Robust Gate, which is highly parameter-efficient at only ~2.5M parameters, successfully learned to route across these massive experts without optimization instability, achieving a test NLL of 4.734 ± 0.006 compared to the monolithic baseline's 4.829 ± 0.012. This confirms that our framework successfully scales to massive LLMs, decoupling geometric interference and yielding a significant performance improvement of 0.095.

We further examined the stability of the models under varying test set compositions, see Table 2 in Appendix G.4. The Gate model consistently exhibits lower loss and hence greater robustness to distribution shifts compared to the monolithic baseline. Details are provided in the appendix.

## 8. Conclusion

We presented a game-theoretic framework for robust generative modeling, deriving a gate $g^*$ with bounded worst-case risk. Our analysis identifies a phase transition: while monolithic models suffer interference proportional to the Jensen-Shannon Divergence, modularity decouples tasks to cancel capacity costs. We proved modularity acts as a "safe" prior, matching optimal retraining in convex regimes while superior for conflicting distributions. Finally, we validated our scalable Primal-Dual algorithm and Structural Distillation on synthetic and real-world datasets.

## Impact Statement

This paper presents work whose goal is to advance the field of Machine Learning. There are many potential societal consequences of our work, none which we feel must be specifically highlighted here.

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

# Contents of Appendix

# A. Extended Content

## A.1. Extended Related Work

Our proposed framework for robust modularity intersects with several active areas of research, including model composition, theoretical routing, and the emerging economics of modular AI ecosystems.

**Robustness and Multiple-Source Adaptation.** Our approach is rooted in the theory of multiple-source domain adaptation (MSA) (Mansour, Mohri, and Rostamizadeh, 2008; 2009; Hoffman, Mohri, and Zhang, 2018; Mohri, Hoffman, and Zhang, 2021; Hoffman, Mohri, and Zhang, 2022; Cortes, Mohri, Suresh, and Zhang, 2021), which seeks to learn predictors robust to mixtures of source domains. Recently, Dann et al. (2025) applied similar minimax principles to the problem of model routing. Their work addresses *value-based* routing, where the goal is to maximize a scalar reward (linear regret). Our work can be viewed as the generative counterpart to this line of research. By moving from linear rewards to the standard KL divergence objective which is needed for tackling generative modeling, we face a fundamentally different mathematical challenge: the resulting optimization problem is convex but non-linear, and crucially, requires enforcing a global normalization constraint ($Z_g = 1$) on the mixture model. This necessitates the constrained minimax analysis developed in this paper, distinguishing our contribution from the unconstrained or locally-constrained optimization found in value-based routing or standard MSA. Our problem formulation also shares historical roots with the *Meta-Pi* network (Hampshire II & Waibel, 1992), which combined expert outputs via a gating mechanism. However, their primary goal was robustness for the *average* dataset, whereas we target the *worst-case* mixture. Furthermore, while they demonstrated source-independence empirically, we provide rigorous existence and convergence guarantees.

**Mixtures, Merging, and Composition.** The concept of combining models has a rich history. *Mixture of Experts (MoE)* (Jacobs et al., 1991; Fedus et al., 2022) trains a routing mechanism jointly with specialized sub-networks. In contrast, our framework operates on *frozen, pre-trained* experts, decoupling the routing learning from the generative training. Another approach is *Model Merging* or *Model Soups* (Wortsman et al., 2022), which averages weights to find a single high-performing static model. Our approach differs by maintaining the experts as discrete entities and using an input-dependent gate $g(x, \cdot)$ to adapt to distribution shifts dynamically.

Recent work explores deeper architectural integration. For example, Bansal et al. (2024) introduce *Composition to Augment Language Models (CALM)*, which leverages cross-attention to merge representations from a base LLM and specialized models, expanding capabilities without full retraining. Distinct from a symmetric modular view, this method designates one model as an *anchor* and the other as an *augmenting* counterpart. It is also not clear how this construction scales beyond two models, as it may require a quadratic number of pairwise cross-attention connections. Complementary to this is *model stitching* (Jiang & Li, 2024), where pre-trained blocks from disparate models, such as BERT and GPT, are integrated directly. Similarly, recent frameworks like StitchLLM (Hu et al., 2025) dynamically route requests across stitched blocks—for instance, feeding the lower layers of one model into the upper layers of another—to optimize the trade-off between latency and accuracy. Crucially, neither approach provides theoretical analysis or guarantees for the resulting composed model. In contrast, our approach preserves experts as black boxes and offers strong theoretical guarantees for a gating mechanism robust to worst-case distribution mixtures.

**Theoretical Routing and Learning to Defer.** Our problem shares conceptual similarities with routing in *learning to defer*, where a learner chooses between predicting or deferring to experts. Foundational work by Cortes, DeSalvo, and Mohri (2016a; 2024a) established the theory for learning with rejection in binary classification. This line of work was significantly expanded to multi-class settings and predictor-rejector frameworks by Mao et al. (2023a; 2024a;b;c;g; 2025a); Mao (2025); DeSalvo et al. (2025); Cortes et al. (2026), supported by $\mathcal{H}$-*consistency* guarantees (Awasthi et al., 2022a;b; Mao et al., 2023f;e;c;b;d; 2024h;d;e;f; Mohri et al., 2024; Cortes et al., 2024b; 2025a;b; Mao et al., 2025c;b; Zhong, 2025; Mohri & Zhong, 2026a;c;b;d; Mohri et al., 2026). Our approach diverges from this literature in three key aspects. First, unlike standard routing which performs a hard selection of a single expert, our gated framework induces a distribution over base models. Second, rather than optimizing for average-case performance, we address *robustness* against adversarial distribution mixtures. Finally, while computational cost is a primary consideration in standard model routing, our current framework focuses purely on statistical performance guarantees.

**Modular Marketplaces and Ecosystems.** Beyond functional integration, the rise of LLMs has spurred interest in the economic dynamics of modular systems. Bhawalkar et al. (2025) analyze "modular marketplaces" from a game-theoretic

perspective, focusing on price equilibria where module owners act strategically to maximize profit. Broader analyses of the AI ecosystem's evolutionary dynamics (Jacobides et al., 2021) further highlight how the interplay between large upstream providers (e.g., cloud and foundation models) and specialized downstream modules is fundamentally reshaping industrial organization. Our work complements these economic and ecosystem perspectives by providing the *statistical* equilibria—ensuring that the aggregated output of these traded modules remains robust regardless of how they are combined.

**Generalization Guarantees.** We provide rigorous generalization bounds for the modular framework using vector-valued Rademacher complexity (Appendix D.4). We show that the sample complexity of learning the robust gate typically admits only a mild logarithmic dependency on the *number* of experts (provided they are diverse), theoretically confirming that modularity acts as a powerful regularizer against overfitting.

## A.2. Problem Formulation Details

Given the setup in Section 2, our objective is to find a single gating function $g \in \mathcal{G}_1$ such that the resulting model $\pi_g$ is a high-quality approximation of any data mixture $\widehat{p}_\lambda$. We use the relative entropy, $D_{KL}$ divergence, as our measure of dissimilarity. This leads to two primary formulations. First, as a preliminary question, we can ask what performance is achievable for a *single, fixed* mixture $\lambda$. This corresponds to the standard convex optimization problem:

$$\min_{g \in \mathcal{G}_1} D_{KL}(\widehat{p}_\lambda \parallel \pi_g).$$

Finding a solution to this problem would provide an optimal gate for a known, static test distribution. Second, and more central to our goal of modularity and robustness, we ask for a *single* gate $g^*$ that performs well against the *worst-case* mixture $\lambda \in \Delta$. This is a robust optimization problem that can be formulated as a minimax game:

$$\min_{g \in \mathcal{G}_1} \max_{\lambda \in \Delta([1,p])} D_{KL}(\widehat{p}_\lambda \parallel \pi_g).$$

The solution $g^*$ to this game would be a truly robust model, providing a uniform performance guarantee across the entire ambiguity set of possible data mixtures.

Our use of the relative entropy, rather than the cross-entropy loss, is essential. While minimizing KL is equivalent to minimizing cross-entropy for a *fixed* target distribution (since entropy is constant), this equivalence breaks down in the robust setting. The entropy term $H(\widehat{p}_\lambda)$ varies with the adversarial choice of $\lambda$. Consequently, minimizing the worst-case cross-entropy $\max_\lambda \mathbb{E}_{\widehat{p}_\lambda}[-\log \pi]$ is not equivalent to minimizing the worst-case divergence $\max_\lambda D_{KL}(\widehat{p}_\lambda \parallel \pi)$. We target the latter to ensure the model approximates the distribution $\widehat{p}_\lambda$ itself, rather than merely covering its support.

Our work aims to answer several fundamental theoretical and algorithmic questions arising from these formulations:

1. Fixed-mixture performance: For a fixed mixture $\lambda$, does there exist a gated solution $\pi_g$ with small divergence $D_{KL}(\widehat{p}_\lambda \parallel \pi_g)$? More specifically, how does this optimal error $\min_{g \in \mathcal{G}_1} D_{KL}(\widehat{p}_\lambda \parallel \pi_g)$ compare to the baseline errors $\epsilon_k$?

2. Robust guarantee: Does there exist a *robust* gated solution $\pi_{g^*}$ that achieves small divergence for *all* $\lambda \in \Delta$? This is a question of existence for the minimax problem $\min_g \max_\lambda D_{KL}(\widehat{p}_\lambda \parallel \pi_g)$.

3. Construction and bounds: If such a robust solution exists, how can we construct it algorithmically? What explicit, non-asymptotic guarantees can we provide for its worst-case performance, $\max_\lambda D_{KL}(\widehat{p}_\lambda \parallel \pi_{g^*})$, in terms of the individual expert guarantees $\epsilon_k$?

4. Comparison to aggregate training: For a fixed mixture $\lambda$, how does the performance of our modular solution $\pi_g$ compare to that of a model $\widehat{\pi}_\lambda$ trained from scratch on the aggregate data $\widehat{p}_\lambda = \sum_{k=1}^p \lambda_k \widehat{p}_k$? Understanding this trade-off is key to justifying the modular approach over the expensive, non-adaptive retraining baseline.

In Section 3, we will address these questions, starting with the existence and bounds for the fixed and robust solutions.

# B. Fundamental Theory (Existence and Generalization)

## B.1. Properties of the Gate Space

**Lemma 2.1.** *The family $\mathcal{G}_1$ is non-empty, compact, and convex.*

*Proof.* $\mathcal{G}_1$ is non-empty since for any $\lambda \in \Delta([1,p])$ it contains the constant gate $g_\lambda(x,k) = \lambda_k$. $\mathcal{G}_1$ is convex since $\mathcal{G}$ is convex, as a product of simplices, and since the affine equality $Z_g = 1$ is preserved by convex combinations. The base family $\mathcal{G} = \prod_{x \in \mathcal{X}_0} \Delta([1,p])$ is compact since each simplex $\Delta([1,p])$ is compact and the product of compact sets (over the finite support $\mathcal{X}_0$, or even countable sets) is compact by Tychonoff's theorem. The constraint function $g \mapsto Z_g = \sum_{x,k} g(x,k)\widehat{\pi}_k(x)$ is continuous since linear. The set $\mathcal{G}_1 = \mathcal{G} \cap \{g: Z_g = 1\}$ is the intersection of a compact set, $\mathcal{G}$, and a closed set, the level set $\{g: Z_g - 1 = 0\}$ of a continuous function. Therefore, $\mathcal{G}_1$ is closed. Since it is a closed subset of a compact set, it is also compact. $\qquad\square$

## B.2. Fixed-Mixture Analysis

### B.2.1. FIXED-MIXTURE GUARANTEE: PROOF OF PROPOSITION 3.1

**Proposition 3.1** (Fixed Mixture Guarantee). *For any fixed $\lambda \in \Delta$, the constant gate $\pi_\lambda = \sum_k \lambda_k \widehat{\pi}_k$ satisfies $D_{\mathrm{KL}}(\widehat{p}_\lambda \parallel \pi_\lambda) \le \sum_{k=1}^p \lambda_k \epsilon_k \le \max_k \epsilon_k$.*

*Proof.* The result follows directly from the joint convexity of the KL divergence:

$$D_{\mathrm{KL}}(\widehat{p}_\lambda \parallel \pi_\lambda) = D_{\mathrm{KL}}\left(\sum_{k=1}^p \lambda_k \widehat{p}_k \,\Big\|\, \sum_{k=1}^p \lambda_k \widehat{\pi}_k\right) \le \sum_{k=1}^p \lambda_k D_{\mathrm{KL}}(\widehat{p}_k \parallel \widehat{\pi}_k) \le \sum_{k=1}^p \lambda_k \epsilon_k.$$

This completes the proof. $\qquad\square$

Note that we could in fact search for the optimal gate function $g^*$ and thus gated solution $\pi_{g^*}$, by solving $g^* = \mathrm{argmin}_{g \in \mathcal{G}_1} D_{\mathrm{KL}}(p_\lambda \parallel \pi_g)$, which is a convex optimization problem since $\mathcal{G}_1$ is convex, $\pi_g$ is linear in $g$, $D_{\mathrm{KL}}$ is jointly convex and composition with a linear function preserves convexity. However, the resulting guarantee for $\pi_{g^*}$ is complex and does not admit a simple expression, as detailed in the following section.

### B.2.2. FIXED-MIXTURE OPTIMAL SOLUTION: CHARACTERIZATION

For a fixed mixture weights vector $\lambda \in \Delta([1,p])$, we consider the convex optimization problem of finding the best normalized gated model:

$$\min_{g \in \mathcal{G}_1} D_{\mathrm{KL}}(\widehat{p}_\lambda \parallel \pi_g).$$

The following lemma characterizes the unique optimal model $\pi_{g^*}$.

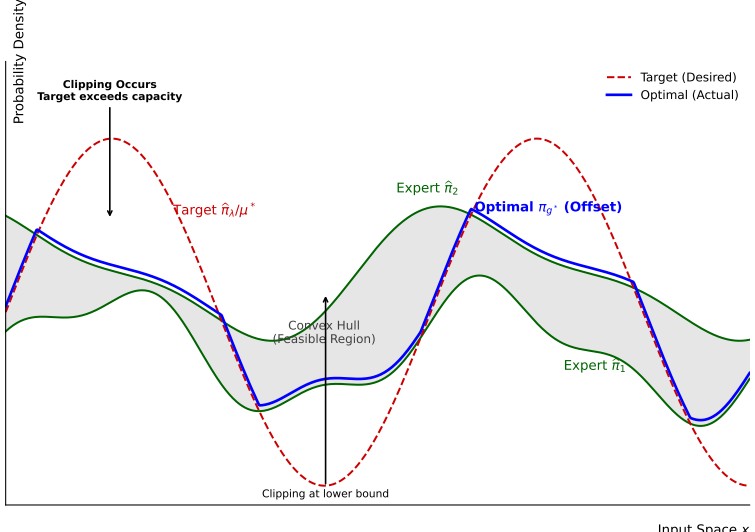

*Figure 6.* Geometric Interpretation of Lemma B.1. The shaded gray region represents the convex hull of the experts $\widehat{\pi}_1$ and $\widehat{\pi}_2$. The target distribution (red dashed) exceeds this feasible capacity. The optimal gate $\pi_{g^*}$ (solid blue) traces the target where possible but is "clipped" to the expert boundaries when the target falls outside the hull.

**Lemma B.1** (Structure of the Optimal Fixed-Mixture Model). *Let $\widehat{\mathsf{p}}_\lambda = \sum_k \lambda_k \widehat{\mathsf{p}}_k$. The optimal model $\pi_{g^*}$ solving the minimization problem is unique and takes the form of a clipped version of the mixture distribution $\widehat{\mathsf{p}}_\lambda$. Specifically, there exists a unique scalar $\mu^* > 0$ such that for all $x \in \mathcal{X}_0$:*

$$\pi_{g^*}(x) = \mathrm{clip}\left(\frac{\widehat{\mathsf{p}}_\lambda(x)}{\mu^*}, m(x), M(x)\right),$$

*where $m(x) = \min_k \widehat{\pi}_k(x)$ and $M(x) = \max_k \widehat{\pi}_k(x)$. The scalar $\mu^*$ is the unique solution to the normalization equation $\sum_{x \in \mathcal{X}_0} \pi_{g^*}(x) = 1$.*

*Proof.* The optimization problem is:

$$\text{minimize} \quad \sum_{x \in \mathcal{X}_0} \widehat{\mathsf{p}}_\lambda(x) \log \frac{\widehat{\mathsf{p}}_\lambda(x)}{\pi_g(x)}$$

$$\text{subject to} \quad \pi_g(x) = \sum_{k=1}^p g(x, k) \widehat{\pi}_k(x) \quad \forall x,$$

$$\sum_{k=1}^p g(x, k) = 1, \quad g(x, k) \geq 0 \quad \forall x, k,$$

$$\sum_{x \in \mathcal{X}_0} \pi_g(x) = 1.$$

Minimizing the $\mathrm{D}_{\mathrm{KL}}$ divergence is equivalent to maximizing the expected log-likelihood: $\sum_x \widehat{\mathsf{p}}_\lambda(x) \log \pi_g(x)$. The local constraints on $g(x, \cdot)$ imply that for any $x$, the value $\pi_g(x)$ must lie in the convex hull of the expert predictions $\{\widehat{\pi}_1(x), \ldots, \widehat{\pi}_p(x)\}$. Since these are scalars, the convex hull is simply the interval $[m(x), M(x)]$. Thus, we can reformulate the problem in terms of the model values $\mathsf{q}_x = \pi_g(x)$:

$$\text{maximize} \quad \sum_{x \in \mathcal{X}_0} \widehat{\mathsf{p}}_\lambda(x) \log \mathsf{q}_x$$

$$\text{subject to} \quad m(x) \leq \mathsf{q}_x \leq M(x) \quad \forall x,$$

$$\sum_{x \in \mathcal{X}_0} \mathsf{q}_x = 1.$$

This is a convex optimization problem. We introduce a Lagrange multiplier $\mu$ for the global equality constraint $\sum_x \mathsf{q}_x = 1$. The Lagrangian is:

$$\mathcal{L}(\mathsf{q}, \mu) = \sum_{x \in \mathcal{X}_0} \widehat{\mathsf{p}}_\lambda(x) \log \mathsf{q}_x - \mu \left[\sum_{x \in \mathcal{X}_0} \mathsf{q}_x - 1\right].$$

We solve this by maximizing $\mathcal{L}$ with respect to $\mathsf{q}_x$ subject to the local interval constraints. The problem decomposes for each $x$:

$$\max_{m(x) \leq \mathsf{q}_x \leq M(x)} \widehat{\mathsf{p}}_\lambda(x) \log \mathsf{q}_x - \mu \mathsf{q}_x.$$

Let $f_x(q) = \widehat{\mathsf{p}}_\lambda(x) \log \mathsf{q} - \mu \mathsf{q}$. The derivative is $f'_x(\mathsf{q}) = \frac{\widehat{\mathsf{p}}_\lambda(x)}{\mathsf{q}} - \mu$. Setting this to zero gives the unconstrained optimum $\mathsf{q}^* = \frac{\widehat{\mathsf{p}}_\lambda(x)}{\mu}$. Since $f_x(\mathsf{q})$ is concave, the constrained optimum is the projection of the unconstrained optimum onto the interval $[m(x), M(x)]$. This is exactly the clipping operation (see Figure 6):

$$\mathsf{q}_x^*(\mu) = \mathrm{clip}\left(\frac{\widehat{\mathsf{p}}_\lambda(x)}{\mu}, m(x), M(x)\right).$$

The optimal $\mu^*$ is found by enforcing the global constraint $Z(\mu) = \sum_x \mathsf{q}_x^*(\mu) = 1$. The function $Z(\mu)$ is continuous and monotonically decreasing in $\mu$. Since $\sum_x m(x) \leq 1$ and $\sum_x M(x) \geq 1$ (as each expert $\pi_k$ sums to 1), there exists a unique $\mu^*$ such that $Z(\mu^*) = 1$. $\qquad\square$

### B.3. Capacity Lower Bound

**Theorem 3.2** (Fundamental Capacity Lower Bound for Static Gating). *Assume the datasets $D_k$ have mutually disjoint supports. Consider the class of static gating functions $\mathcal{G}_{const} \subset \mathcal{G}_1$, defined as gates where $g(x, k) = w_k$ is independent of $x$*

*for all $k$. For any static gate $\mathbf{w} \in \mathcal{G}_{const}$, the worst-case Kullback-Leibler divergence is lower-bounded by:*

$$\max_{k \in \{1,\dots,p\}} D_{\mathrm{KL}}(\widehat{\mathsf{p}}_k \| \pi_{\mathbf{w}}) \geq \log\left(\sum_{j=1}^{p} e^{\epsilon_j}\right).$$

*Proof.* Let $\mathbf{w} = [w_1, \dots, w_p]$ be the weight vector. The global normalization constraint on $\mathcal{G}_1$ requires:

$$\sum_{k=1}^{p} \int g(x,k)\widehat{\pi}_k(x)dx = \sum_{k=1}^{p} w_k \underbrace{\int \widehat{\pi}_k(x)dx}_{1} = \sum_{k=1}^{p} w_k = 1. \tag{1}$$

Thus, the static weights must lie on the simplex. Under the disjoint support assumption, for any $x \in \mathrm{supp}(D_k)$, the other experts have zero density ($\widehat{\pi}_j(x) = 0$ for $j \neq k$). Thus, the mixture density simplifies exactly to:

$$\pi_{\mathbf{w}}(x) = w_k \widehat{\pi}_k(x). \tag{2}$$

We evaluate the KL divergence for task $k$ using this exact form:

$$D_{\mathrm{KL}}(\widehat{\mathsf{p}}_k \| \pi_{\mathbf{w}}) = \mathop{\mathbb{E}}_{x \sim \widehat{\mathsf{p}}_k}\left[\log\left(\frac{\widehat{\mathsf{p}}_k(x)}{w_k \widehat{\pi}_k(x)}\right)\right] \tag{3}$$

$$= \underbrace{\mathop{\mathbb{E}}_{x \sim \widehat{\mathsf{p}}_k}\left[\log\left(\frac{\widehat{\mathsf{p}}_k(x)}{\widehat{\pi}_k(x)}\right)\right]}_{\epsilon_k} - \mathop{\mathbb{E}}_{x \sim \widehat{\mathsf{p}}_k}\left[\log w_k\right] \tag{4}$$

$$= \epsilon_k - \log w_k. \tag{5}$$

Let $\delta = \max_k D_{\mathrm{KL}}(\widehat{\mathsf{p}}_k \| \pi_{\mathbf{w}})$ be the worst-case risk. Then for all $k$:

$$\delta \geq \epsilon_k - \log w_k \Rightarrow w_k \geq e^{\epsilon_k - \delta}. \tag{6}$$

Summing the weights over all $p$ experts:

$$1 = \sum_{k=1}^{p} w_k \geq \sum_{k=1}^{p} e^{\epsilon_k - \delta} = e^{-\delta}\sum_{k=1}^{p} e^{\epsilon_k}. \tag{7}$$

Rearranging to solve for the risk $\delta$:

$$e^{\delta} \geq \sum_{k=1}^{p} e^{\epsilon_k} \implies \delta \geq \log\left(\sum_{k=1}^{p} e^{\epsilon_k}\right). \tag{8}$$

This establishes that no static weighting scheme can surpass this capacity limit. $\square$

### B.4. Robust Existence Proof

We define the *linearized game* as the one with payoff: $\widetilde{L}(\lambda, g) = \sum_{k=1}^{p} \lambda_k D_{\mathrm{KL}}(\widehat{\mathsf{p}}_k \| \pi_g)$.

**Theorem 3.3** (Robust Existence). *The linearized modular game admits a saddle point $(\lambda^*, g^*) \in \Delta([1,p]) \times \mathcal{G}_1$. For any mixture $\lambda \in \Delta$, the robust gate $g^*$ satisfies for* any $\lambda \in \Delta([1,p])$:

$$D_{\mathrm{KL}}(\widehat{\mathsf{p}}_\lambda \| \pi_{g^*}) \leq \log\left[\sum_{k=1}^{p} e^{\epsilon_k}\right] - H_\sigma^{\lambda^*}(K|X) - D_{\mathrm{JSD}}^{\lambda}(\{\widehat{\mathsf{p}}_k\}),$$

*where $H_\sigma^{\lambda^*}(K|X) = \sum_{k=1}^{p} \lambda_k^* \mathop{\mathbb{E}}_{x \sim \widehat{\mathsf{p}}_k}\left[-\log\frac{\sigma_k \widehat{\pi}_k(x)}{\pi_\sigma(x)}\right]$ is the target-weighted conditional entropy of the expert assignment under the robust constant gate $\pi_\sigma = \sum_{k=1}^{p} \sigma_k \widehat{\pi}_k$ defined by the softmax weights $\sigma_k = e^{\epsilon_k} / \sum_{j=1}^{p} e^{\epsilon_j}$.*

*Proof.* The proof consists of casting the problem as a two-player, zero-sum game and showing the existence of a saddle point via Kakutani's Fixed Point Theorem.

The original payoff function is $L(\lambda, g) = D_{\mathrm{KL}}(\widehat{\mathsf{p}}_\lambda \| \pi_g)$. Since $L$ is convex in $\lambda$, its maximizers lie strictly at the vertices. To satisfy the convexity requirement of Kakutani's theorem, we consider the *linearized game* with payoff:

$$\widetilde{L}(\lambda, g) = \sum_{k=1}^{p} \lambda_k D_{\mathrm{KL}}(\widehat{\mathsf{p}}_k \| \pi_g).$$

We define the best-response functions:

$$\Lambda^*(g) = \underset{\lambda' \in \Delta([1,p])}{\operatorname{argmax}} \widetilde{L}(\lambda', g), \qquad G^*(\lambda) = \underset{g' \in \mathcal{G}_1}{\operatorname{argmin}} \widetilde{L}(\lambda, g').$$

We show that the correspondence $T(\lambda, g) = (\Lambda^*(g), G^*(\lambda))$ satisfies Kakutani's conditions: For $G^*(\lambda)$, $\widetilde{L}$ is convex in $g$ (as a convex combination of convex KL terms), so the set of minimizers is convex. For $\Lambda^*(g)$, $\widetilde{L}$ is linear in $\lambda$, so the set of maximizers is a convex face of the simplex. Since $\widetilde{L}$ is continuous and the domains are compact, Berge's Maximum Theorem implies $\Lambda^*$ and $G^*$ have closed graphs. Kakutani's fixed-point theorem thus guarantees the existence of a fixed point $(\lambda^*, g^*) \in (\Lambda^*(g^*), G^*(\lambda^*))$, which is a saddle point for $\widetilde{L}$.

**Bounding the Value.** We now bound the worst-case risk of this optimal solution. By the saddle point property, $\widetilde{L}(\lambda^*, g^*) = \max_\lambda \widetilde{L}(\lambda, g^*)$. We use the fundamental identity relating the mixture risk $L$ and the linearized risk $\widetilde{L}$: for any $\lambda \in \Delta([1,p])$,

$$\begin{aligned}
D_{\mathrm{KL}}(\widehat{p}_\lambda \parallel \pi_{g^*}) &= \widetilde{L}(\lambda, g^*) - D_{\mathrm{JSD}}^\lambda(\widehat{p}_1, \ldots, \widehat{p}_p) \\
&\le \widetilde{L}(\lambda^*, g^*) - D_{\mathrm{JSD}}^\lambda(\widehat{p}_1, \ldots, \widehat{p}_p).
\end{aligned} \tag{9}$$

We must bound $\widetilde{L}(\lambda^*, g^*)$. Since $g^*$ is the minimizer of $\widetilde{L}(\lambda^*, \cdot)$ over $\mathcal{G}_1$, its loss is bounded by that of *any* specific witness gate. We choose the Robust Constant Gate $\pi_\sigma$ defined by the softmax weights $\sigma_k = e^{\epsilon_k}/Z$, where $Z = \sum e^{\epsilon_j}$ (the solution of the proof of Theorem 3.2):

$$\widetilde{L}(\lambda^*, g^*) \le \widetilde{L}(\lambda^*, \pi_\sigma) = \sum_{k=1}^p \lambda_k^* D_{\mathrm{KL}}(\widehat{p}_k \parallel \pi_\sigma).$$

We expand the component KL divergence $D_{\mathrm{KL}}(\widehat{p}_k \parallel \pi_\sigma)$. Since $\pi_\sigma(x) \ge \sigma_k \widehat{\pi}_k(x)$:

$$\begin{aligned}
D_{\mathrm{KL}}(\widehat{p}_k \parallel \pi_\sigma) &= \underset{x \sim \widehat{p}_k}{\mathbb{E}} \left[ \log \frac{\widehat{p}_k(x)}{\pi_\sigma(x)} \right] \\
&= \underset{x \sim \widehat{p}_k}{\mathbb{E}} \left[ \log \frac{\widehat{p}_k(x)}{\sigma_k \widehat{\pi}_k(x)} + \log \frac{\sigma_k \widehat{\pi}_k(x)}{\pi_\sigma(x)} \right] \\
&= \underbrace{D_{\mathrm{KL}}(\widehat{p}_k \parallel \widehat{\pi}_k)}_{\epsilon_k} - \log \sigma_k - \underbrace{\underset{x \sim \widehat{p}_k}{\mathbb{E}} \left[ -\log \frac{\sigma_k \widehat{\pi}_k(x)}{\pi_\sigma(x)} \right]}_{H_k(K|x)}.
\end{aligned}$$

Substituting $\sigma_k = e^{\epsilon_k}/Z$, we have $-\log \sigma_k = \log Z - \epsilon_k$. The $\epsilon_k$ terms cancel:

$$D_{\mathrm{KL}}(\widehat{p}_k \parallel \pi_\sigma) = \log Z - H_k(K|x).$$

Averaging over $\lambda^*$:

$$\widetilde{L}(\lambda^*, g^*) \le \sum_{k=1}^p \lambda_k^* (\log Z - H_k(K|x)) = \log Z - H_\sigma^{\lambda^*}(K|X).$$

Substituting this upper bound back into Eq. (9) completes the proof. $\qquad\square$

## B.5. Geometric Interpretation of the Bound

The upper bound in Theorem 3.3 reveals how the robust gate leverages the geometry of the task distributions. We analyze the bound $V^* \le \mathrm{LSE} - \mathrm{Overlap} - \mathrm{Diversity}$ in three limiting regimes.

**Case 1: The Specialization Limit (Disjoint Experts).** Consider the case where task supports are mutually disjoint $(\mathrm{supp}(\widehat{p}_k) \cap \mathrm{supp}(\widehat{p}_j) = \varnothing)$.

- **Geometry:** The expert assignment is deterministic, so the overlap gain vanishes: $H_\sigma^{\lambda^*}(K|X) = 0$.

- **Diversity:** The diversity gain depends on the test mixture $\lambda$. For disjoint supports, it is equal to the entropy: $D_{\mathrm{JSD}}^\lambda = H(\lambda)$.

- **Result:** The bound becomes $D_{\mathrm{KL}}(\widehat{p}_\lambda \parallel \pi_{g^*}) \le \log(\sum e^{\epsilon_k}) - H(\lambda)$. If the test mixture is difficult (high entropy, e.g., balanced tasks), then $H(\lambda) \approx \log p$, canceling the capacity cost. This guarantees that the modular model incurs no capacity penalty precisely when the task is most complex.

**Case 2: The Redundancy Limit (Identical Experts).** Consider the simplified case where all experts and targets are identical and have equal error $\epsilon$.

- **Geometry:** Tasks are indistinguishable ($D_{\mathrm{JSD}} = 0$) and weights are uniform ($H(\sigma) = \log p$).

- **Result:** The capacity cost $\log(pe^\epsilon) = \epsilon + \log p$ is exactly refunded by the overlap gain ($\log p$).

$$V^* \leq (\epsilon + \log p) - \log p = \epsilon.$$

Thus, the modular system recovers the performance of a single expert.

**Case 3: The Ensemble Mechanism (Exact Cancellation).** We now analyze the general mechanism of overlap for arbitrary errors. Consider the case where experts overlap fully (identical targets, $D_{\mathrm{JSD}} = 0$) but have different errors $\epsilon_k$. The overlap gain becomes the entropy of the robust weights $H(\sigma)$. We recall that for the robust gate, $H(\sigma) = \log Z - \sum_k \sigma_k \epsilon_k$. Substituting this into the bound:

$$V^* \leq \underbrace{\log Z}_{\text{Capacity Cost}} - \underbrace{\left(\log Z - \sum_{k=1}^{p} \sigma_k \epsilon_k\right)}_{\text{Overlap Gain } H(\sigma)} - \underbrace{0}_{\text{Diversity}} \tag{10}$$

$$V^* \leq \sum_{k=1}^{p} \sigma_k \epsilon_k. \tag{11}$$

This derivation proves that in the high-overlap regime, the "Capacity Cost" (LogSumExp) is *exactly cancelled* by the ambiguity of the gate. The bound collapses to the *weighted average error* of the experts. The modular system effectively transforms into a static ensemble, pooling the experts to minimize risk.

# C. Extensions: Prior Knowledge

### C.1. Guarantees with prior knowledge on mixture weights

In certain applications, we may have prior knowledge suggesting that the mixture weights encountered at test time will be restricted to a convex subset $\Lambda \subset \Delta([1,p])$. This knowledge can be leveraged to derive a more specialized solution with significantly stronger performance guarantees.

The existence result from Theorem 3.3 extends directly to this scenario. By considering the linearized game restricted to the compact convex set $\Lambda$, the convexity of the best-response sets is preserved, ensuring the existence of a saddle point via Kakutani's theorem. The value of this restricted game, $V_\Lambda^*$, is guaranteed to be no worse than the original game value, $V_\Delta^*$. The adversary ($\lambda$-player) has a smaller set of strategies, which limits their ability to find high-loss mixtures. This results in a lower worst-case loss for our solution:

$$V_\Lambda^* = \min_{g \in \mathcal{G}_1} \max_{\lambda \in \Lambda} L(\lambda, g) \leq \min_{g \in \mathcal{G}_1} \max_{\lambda \in \Delta([1,p])} L(\lambda, g) = V_\Delta^*.$$

Intuitively, the gating function no longer needs to defend against unrealistic mixture weights outside of $\Lambda$. It can therefore specialize its performance for the known set of likely scenarios.

We now formalize the superiority of this specialized solution. Let $g_\Delta^*$ be the optimal robust gate found by solving the original problem over the full simplex $\Delta$, and let $g_\Lambda^*$ be the optimal gate found by solving the restricted problem over $\Lambda$ (see Figure 7).

**Theorem C.1** (Dominance of the Specialized Solution). *Let $g_\Delta^*$ be the optimal robust gate over the full simplex $\Delta([1,p])$, and let $g_\Lambda^*$ be the optimal gate over the restricted convex set $\Lambda \subset \Delta([1,p])$. Then the worst-case performance of $g_\Lambda^*$ over $\Lambda$ is at least as good as the performance of $g_\Delta^*$ over the same set:*

$$\max_{\lambda \in \Lambda} D_{\mathrm{KL}}(\widehat{p}_\lambda \parallel \pi_{g_\Lambda^*}) \leq \max_{\lambda \in \Lambda} D_{\mathrm{KL}}(\widehat{p}_\lambda \parallel \pi_{g_\Delta^*}).$$

The proof is presented in Appendix C.2.1. To make the benefit more concrete, we can quantify the improvement under a Lipschitz assumption, see proof in Appendix C.2.2.

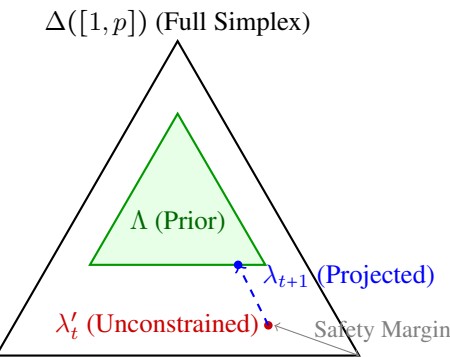

*Figure 7.* Geometry of Prior Knowledge (Section C.1). The outer triangle represents the full probability simplex $\Delta$. The green region $\Lambda$ represents the subset of valid mixtures defined by prior knowledge. The algorithm projects the adversary's updates (red point) back onto $\Lambda$ (blue point), tightening the worst-case bound as per Theorem C.2.

**Theorem C.2** (Quantitative Improvement in Game Value). *Assume that for any fixed gate $g \in \mathcal{G}_1$, the mapping $\lambda \mapsto D_{KL}(\widehat{p}_\lambda \parallel \pi_g)$ is $L$-Lipschitz with respect to the $\ell_1$-norm:*

$$|D_{KL}(\widehat{p}_\lambda \parallel \pi_g) - D_{KL}(\widehat{p}_{\lambda'} \parallel \pi_g)| \le L\|\lambda - \lambda'\|_1,$$

*for all $\lambda, \lambda' \in \Delta([1, p])$. Let $V_\Delta^* = \min_g \max_{\lambda \in \Delta} D_{KL}(\widehat{p}_\lambda \parallel \pi_g)$ be the minimax value over the full simplex, and let $V_\Lambda^* = \min_g \max_{\lambda \in \Lambda} D_{KL}(\widehat{p}_\lambda \parallel \pi_g)$ be the value over the restricted set. The improvement is bounded by:*

$$0 \le V_\Delta^* - V_\Lambda^* \le L \cdot d_H(\Lambda, \Delta([1, p])),$$

*where $d_H(\Lambda, \Delta([1, p])) = \max_{\lambda \in \Delta([1, p])} \min_{\lambda' \in \Lambda} \|\lambda - \lambda'\|_1$ is the Hausdorff distance between the sets.*

**Explicit Lipschitz Constant.** In practice, if the support $\mathcal{X}_0$ is finite and all probabilities are strictly positive, an explicit Lipschitz constant is

$$L = \max_{k \in [1, p]} \max_{g \in \mathcal{G}_1} \sum_{x \in \mathcal{X}_0} \widehat{p}_k(x)\left|\log \frac{\widehat{p}_k(x)}{\pi_g(x)}\right|.$$

This constant bounds the maximum gradient of the loss with respect to the mixture weights $\lambda$. Crucially, it depends only on the extremal geometry of the experts and is independent of the mixture $\lambda$.

**Example.** Consider a case with two experts: a high-quality model $\widehat{\pi}_1$ ($\epsilon_1 = 0.01$) and a low-quality model $\widehat{\pi}_2$ ($\epsilon_2 = 0.5$). The general solution $g_\Delta^*$ must be robust against the worst-case mixture $\lambda = (0, 1)$, so its guaranteed performance $V^*$ will be close to $0.5$. However, if we have prior knowledge that the second source will never constitute more than 5% of the mixture (i.e., $\Lambda = \{\lambda \mid \lambda_2 \le 0.05\}$), the specialized solution $g_\Lambda^*$ can largely ignore this worst-case scenario. Its guaranteed performance $V_\Lambda^*$ will be dramatically lower, focusing on mixtures dominated by the high-quality expert.

**Robustness to Mis-specification.** The specialized gate is naturally robust to small mis-specifications of the set $\Lambda$. The Lipschitz assumption allows us to bound the performance on a slightly expanded set $\Lambda_\delta = \{\lambda \in \Delta([1, p]) \mid \exists \lambda' \in \Lambda, \|\lambda - \lambda'\|_1 \le \delta\}$. The performance of the gate $g_\Lambda^*$ degrades gracefully:

$$\max_{\lambda \in \Lambda_\delta} L(\lambda, g_\Lambda^*) \le V_\Lambda^* + L\delta.$$

**Algorithmic Adaptation.** The optimization algorithm presented in Section 4 is easily modified to adapt to this scenario. The update step for the $\lambda$-player is simply augmented with a projection back onto the convex set $\Lambda$. First, we compute the standard intermediate update $\lambda_{t+1}'$:

$$\lambda_{t+1}'(k) = \frac{\lambda_t(k)\exp(\eta_\lambda \ell_t(k))}{\sum_{j=1}^{p} \lambda_t(j)\exp(\eta_\lambda \ell_t(j))}.$$

Then, we project this distribution onto the restricted set $\Lambda$ to obtain the new weights:

$$\lambda_{t+1} = P_\Lambda(\lambda_{t+1}') = \underset{q \in \Lambda}{\operatorname{argmin}}\, D_{KL}(q \parallel \lambda_{t+1}').$$

This ensures that the mixture weights always remain within the specified prior constraints.

## C.2. Proofs for Prior Knowledge

### C.2.1. DOMINANCE OF SPECIALIZED SOLUTION

**Theorem C.1** (Dominance of the Specialized Solution). *Let $g_\Delta^*$ be the optimal robust gate over the full simplex $\Delta([1, p])$, and let $g_\Lambda^*$ be the optimal gate over the restricted convex set $\Lambda \subset \Delta([1, p])$. Then the worst-case performance of $g_\Lambda^*$ over $\Lambda$ is at least as good as the performance of $g_\Delta^*$ over the same set:*

$$\max_{\lambda \in \Lambda} \mathsf{D}_{\mathrm{KL}}(\widehat{\mathsf{p}}_\lambda \parallel \pi_{g_\Lambda^*}) \le \max_{\lambda \in \Lambda} \mathsf{D}_{\mathrm{KL}}(\widehat{\mathsf{p}}_\lambda \parallel \pi_{g_\Delta^*}).$$

*Proof.* By definition of the restricted minimax solution $g_\Lambda^*$, we have

$$\max_{\lambda \in \Lambda} \mathsf{D}_{\mathrm{KL}}(\widehat{\mathsf{p}}_\lambda \parallel \pi_{g_\Lambda^*}) = \min_{g \in \mathcal{G}_1} \max_{\lambda \in \Lambda} \mathsf{D}_{\mathrm{KL}}(\widehat{\mathsf{p}}_\lambda \parallel \pi_g).$$

Since $g_\Delta^* \in \mathcal{G}_1$ is a feasible gate, its performance must be greater than or equal to the minimum over all gates:

$$\min_{g \in \mathcal{G}_1} \max_{\lambda \in \Lambda} \mathsf{D}_{\mathrm{KL}}(\widehat{\mathsf{p}}_\lambda \parallel \pi_g) \le \max_{\lambda \in \Lambda} \mathsf{D}_{\mathrm{KL}}(\widehat{\mathsf{p}}_\lambda \parallel \pi_{g_\Delta^*}).$$

Combining the two statements gives the claimed inequality. $\square$

### C.2.2. QUANTITATIVE IMPROVEMENT

**Theorem C.2** (Quantitative Improvement in Game Value). *Assume that for any fixed gate $g \in \mathcal{G}_1$, the mapping $\lambda \mapsto \mathsf{D}_{\mathrm{KL}}(\widehat{\mathsf{p}}_\lambda \parallel \pi_g)$ is L-Lipschitz with respect to the $\ell_1$-norm:*

$$|\mathsf{D}_{\mathrm{KL}}(\widehat{\mathsf{p}}_\lambda \parallel \pi_g) - \mathsf{D}_{\mathrm{KL}}(\widehat{\mathsf{p}}_{\lambda'} \parallel \pi_g)| \le L\|\lambda - \lambda'\|_1,$$

*for all $\lambda, \lambda' \in \Delta([1, p])$. Let $V_\Delta^* = \min_g \max_{\lambda \in \Delta} \mathsf{D}_{\mathrm{KL}}(\widehat{\mathsf{p}}_\lambda \parallel \pi_g)$ be the minimax value over the full simplex, and let $V_\Lambda^* = \min_g \max_{\lambda \in \Lambda} \mathsf{D}_{\mathrm{KL}}(\widehat{\mathsf{p}}_\lambda \parallel \pi_g)$ be the value over the restricted set. The improvement is bounded by:*

$$0 \le V_\Delta^* - V_\Lambda^* \le L \cdot d_H(\Lambda, \Delta([1, p])),$$

*where $d_H(\Lambda, \Delta([1, p])) = \max_{\lambda \in \Delta([1, p])} \min_{\lambda' \in \Lambda} \|\lambda - \lambda'\|_1$ is the Hausdorff distance between the sets.*

*Proof.* For any fixed gate $g$, define the worst-case loss over a set $S$ as $F(g, S) = \max_{\lambda \in S} \mathsf{D}_{\mathrm{KL}}(\widehat{\mathsf{p}}_\lambda \parallel \pi_g)$.

Let $\lambda^* \in \Delta([1, p])$ be a mixture that achieves the maximum for the full simplex, i.e., $\mathsf{D}_{\mathrm{KL}}(\widehat{\mathsf{p}}_{\lambda^*} \parallel \pi_g) = F(g, \Delta)$. Let $\lambda_{proj}$ be the point in $\Lambda$ closest to $\lambda^*$ in the $\ell_1$-norm. By the definition of the Hausdorff distance, $\|\lambda^* - \lambda_{proj}\|_1 \le d_H(\Lambda, \Delta)$.

Using the Lipschitz assumption:

$$\begin{aligned}
F(g, \Delta) - F(g, \Lambda) &= \mathsf{D}_{\mathrm{KL}}(\widehat{\mathsf{p}}_{\lambda^*} \parallel \pi_g) - \max_{\lambda \in \Lambda} \mathsf{D}_{\mathrm{KL}}(\widehat{\mathsf{p}}_\lambda \parallel \pi_g) \\
&\le \mathsf{D}_{\mathrm{KL}}(\widehat{\mathsf{p}}_{\lambda^*} \parallel \pi_g) - \mathsf{D}_{\mathrm{KL}}(\widehat{\mathsf{p}}_{\lambda_{proj}} \parallel \pi_g) &\text{(Since } \lambda_{proj} \in \Lambda) \\
&\le L\|\lambda^* - \lambda_{proj}\|_1 \\
&\le L \cdot d_H(\Lambda, \Delta([1, p])).
\end{aligned}$$

This inequality holds for *any* gate $g$. Therefore:

$$F(g, \Delta) \le F(g, \Lambda) + L \cdot d_H(\Lambda, \Delta).$$

Taking the minimum over $g \in \mathcal{G}_1$ on both sides preserves the inequality:

$$\min_g F(g, \Delta) \le \min_g F(g, \Lambda) + L \cdot d_H(\Lambda, \Delta).$$

Substituting the definitions $V_\Delta^* = \min_g F(g, \Delta)$ and $V_\Lambda^* = \min_g F(g, \Lambda)$ yields the upper bound. The lower bound $V_\Delta^* - V_\Lambda^* \ge 0$ follows immediately because $\Lambda \subset \Delta$, so the maximum over $\Lambda$ can never exceed the maximum over $\Delta$. $\square$

# D. Theoretical Comparison: Modular Gating vs. Monolithic Retraining

In this appendix, we rigorously contrast our proposed Modular Robustness with standard Monolithic training (or retraining on a mixture). We analyze the trade-off from four perspectives:

1. **Qualitative Scenarios (Section D.1):** We illustrate the trade-off between specialization and flexibility with concrete examples.

2. **Optimization Geometry (Section D.2):** We prove that monolithic models face a fundamental "Interference Floor" determined by the Jensen-Shannon Divergence.

3. **Architectural Safety (Section D.3):** We show that for linear models, our approach coincides with the optimal retrained model.

4. **Statistical Generalization (Section D.4):** We derive bounds showing that modular sample complexity scales with the lightweight gate.

## D.1. Qualitative Scenarios: Retraining vs. Gating

We compare the optimal gated model for a fixed $\lambda$, $\pi_{g_\lambda^*} = \operatorname{argmin}_{\pi \in \Pi_G} \mathsf{D}_{\mathrm{KL}}(\widehat{\mathsf{p}}_\lambda \parallel \pi)$ (where $\Pi_G = \{\pi_g \mid g \in \mathcal{G}_1\}$), with the optimal retrained model $\widehat{\pi}_\lambda = \operatorname{argmin}_{\pi \in \Pi} \mathsf{D}_{\mathrm{KL}}(\widehat{\mathsf{p}}_\lambda \parallel \pi)$. Both approaches seek the best model for $\widehat{\mathsf{p}}_\lambda$ within their respective model classes ($\Pi_G$ for gating, $\Pi$ for retraining).

For a fixed $\lambda$, both optimal solutions are, by definition, at least as good as the simple, non-adaptive mixture $\pi_\lambda = \sum_k \lambda_k \widehat{\pi}_k$ (which is a feasible point in both $\mathcal{G}_1$ and $\Pi$, assuming $\Pi$ is closed under convex combinations). Thus, by Proposition 3.1, both optimal solutions have their performance upper-bounded by the same quantity:

$$\mathsf{D}_{\mathrm{KL}}(\widehat{\mathsf{p}}_\lambda \parallel \pi_{g_\lambda^*}) \le \mathsf{D}_{\mathrm{KL}}(\widehat{\mathsf{p}}_\lambda \parallel \pi_\lambda) \le \sum_k \lambda_k \epsilon_k$$

$$\mathsf{D}_{\mathrm{KL}}(\widehat{\mathsf{p}}_\lambda \parallel \widehat{\pi}_\lambda) \le \mathsf{D}_{\mathrm{KL}}(\widehat{\mathsf{p}}_\lambda \parallel \pi_\lambda) \le \sum_k \lambda_k \epsilon_k$$

The key difference lies in which model class, $\Pi_G$ (the $x$-dependent mixtures of experts) or $\Pi$ (the original model family), is better suited for approximating the specific mixture distribution $\widehat{\mathsf{p}}_\lambda$. This leads to a trade-off between specialization and flexibility, illustrated in the following scenarios.

- **Scenario 1: Retraining Wins (Shuffling of the data may introduce new structures in the mixture).** The retrained model $\widehat{\pi}_\lambda$ will outperform the best gated model $\pi_{g_\lambda^*}$ when shuffling of the mixture distribution $\widehat{\mathsf{p}}_\lambda$ introduces significant new patterns or structures that are not present in the individual source distributions $\widehat{\mathsf{p}}_k$ and cannot be approximated by an $x$-dependent mixture of their expert models $\{\widehat{\pi}_k\}$.

  **Abstract Case:** Let $\Pi$ be a powerful, universal model class. The optimal model for the mixture, $\widehat{\pi}_\lambda \in \Pi$, may be very accurate (low $\mathsf{D}_{\mathrm{KL}}$). However, if this optimal $\widehat{\pi}_\lambda$ is fundamentally different from any model in $\Pi_G = \{\sum_k g(x,k)\widehat{\pi}_k \mid g \in \mathcal{G}_1\}$, then $\widehat{\pi}_\lambda \notin \Pi_G$. The best gated model $\pi_{g_\lambda^*}$ will be a poor approximation, and its error $V_1(\lambda) = \min_{\pi \in \Pi_G} \mathsf{D}_{\mathrm{KL}}(\widehat{\mathsf{p}}_\lambda \parallel \pi)$ will be large.

  **Concrete Example:** Let $\widehat{\mathsf{p}}_1$ be text describing only cats, making $\widehat{\pi}_1$ a "cat expert". Let $\widehat{\mathsf{p}}_2$ be text describing only dogs, making $\widehat{\pi}_2$ a "dog expert". Let the mixture $\widehat{\mathsf{p}}_\lambda$ contain both $\widehat{\mathsf{p}}_1$ and $\widehat{\mathsf{p}}_2$, but from shuffling of the data, possibly also a new set of documents that *compare* cats and dogs. The retrained model $\widehat{\pi}_\lambda \in \Pi$ sees these new comparison articles and learns the corresponding language patterns. The gated model $\pi_{g_\lambda^*} \in \Pi_G$, however, has no "comparison expert" to choose from. It can only mix the cat expert and the dog expert, resulting in a poor approximation of the new comparison texts. In this case, we expect $\mathsf{D}_{\mathrm{KL}}(\widehat{\mathsf{p}}_\lambda \parallel \widehat{\pi}_\lambda) < \mathsf{D}_{\mathrm{KL}}(\widehat{\mathsf{p}}_\lambda \parallel \pi_{g_\lambda^*})$.

- **Scenario 2: Gated Model Wins (Conflicting tasks).** The gated model $\pi_{g_\lambda^*}$ will outperform the retrained model $\widehat{\pi}_\lambda$ when the source distributions $\widehat{\mathsf{p}}_k$ represent distinct or conflicting tasks that are "easier" to learn separately than together, especially if the base model class $\Pi$ has limited capacity or suffers from optimization issues like catastrophic forgetting.

  **Abstract Case:** Let $\widehat{\mathsf{p}}_1$ and $\widehat{\mathsf{p}}_2$ have largely disjoint supports. A retrained model $\widehat{\pi}_\lambda \in \Pi$ trained on the mixture $\widehat{\mathsf{p}}_\lambda$ may find a poor compromise, modeling neither $\widehat{\mathsf{p}}_1$ nor $\widehat{\mathsf{p}}_2$ well (a phenomenon known as parameter interference or catastrophic forgetting). Its error $\mathsf{D}_{\mathrm{KL}}(\widehat{\mathsf{p}}_\lambda \parallel \widehat{\pi}_\lambda)$ could be high, potentially $\gg \max(\epsilon_1, \epsilon_2)$. The optimal gate $g_\lambda^*$,

however, can learn to be a perfect "router": it sets $g^*(x, 1) \approx 1$ for $x \in \operatorname{supp}(\widehat{p}_1)$ and $g^*(x, 2) \approx 1$ for $x \in \operatorname{supp}(\widehat{p}_2)$. The resulting model $\pi_{g_\lambda^*}(x)$ effectively *is* $\widehat{\pi}_1(x)$ on the first part of the data and $\widehat{\pi}_2(x)$ on the second. The performance $V_1(\lambda) = D_{\mathrm{KL}}(\widehat{p}_\lambda \parallel \pi_{g_\lambda^*})$ will be approximately the average of the individual expert errors: $V_1(\lambda) \approx \lambda_1 D_{\mathrm{KL}}(\widehat{p}_1 \parallel \widehat{\pi}_1) + \lambda_2 D_{\mathrm{KL}}(\widehat{p}_2 \parallel \widehat{\pi}_2) \le \sum_k \lambda_k \epsilon_k \le \max_k \epsilon_k$.

**Concrete Example:** Let $\Pi$ be a transformer. Let $\widehat{p}_1$ be an English→French translation dataset ($\widehat{\pi}_1$ is the expert, low $\epsilon_1$) and $\widehat{p}_2$ be English→German ($\widehat{\pi}_2$ is the expert, low $\epsilon_2$). Retraining $\widehat{\pi}_\lambda$ on the 50/50 mixture may cause parameter interference, resulting in a model that confuses French and German vocabulary. The $x$-dependent gate $g_\lambda^*$ can learn to read the prompt (e.g., "Translate to French:") and set $g^*(x, 1) = 1$, perfectly routing to the correct expert. In this common scenario, we expect $D_{\mathrm{KL}}(\widehat{p}_\lambda \parallel \pi_{g_\lambda^*}) < D_{\mathrm{KL}}(\widehat{p}_\lambda \parallel \widehat{\pi}_\lambda)$.

## D.2. The Optimization Gap: Interference vs. Decoupling

### D.2.1. THE MONOLITHIC BARRIER: DIVERSITY AS INTERFERENCE

Consider a monolithic model $\widehat{\pi}_\lambda$ retrained to minimize the loss on the aggregate mixture $\widehat{p}_\lambda = \sum_{k=1}^{p} \lambda_k \widehat{p}_k$. While the model may achieve low training loss on the mixture distribution, its performance on individual tasks is governed by the *Jensen-Shannon Decomposition Identity*. For any model $\pi$, the average task risk decomposes exactly into two terms:

$$\underbrace{\sum_{k=1}^{p} \lambda_k D_{\mathrm{KL}}(\widehat{p}_k \parallel \pi)}_{\text{Average Task Risk}} = \underbrace{D_{\mathrm{KL}}(\widehat{p}_\lambda \parallel \pi)}_{\text{Mixture Fit}} + \underbrace{D_{\mathrm{JSD}}^{\lambda}(\widehat{p}_1, \ldots, \widehat{p}_p)}_{\text{Interference}}. \tag{12}$$

This equality reveals a fundamental limitation. Even in the limit of infinite capacity where the model fits the global mixture perfectly ($D_{\mathrm{KL}}(\widehat{p}_\lambda \parallel \pi) \to 0$), the average risk is strictly lower-bounded by the task diversity. We formalize this in the following theorem.

**Theorem D.1** (The JSD Gap). *Let $\{\widehat{p}_k\}_{k=1}^{p}$ be source distributions and let $\epsilon_k = \min_{\pi \in \Pi} D_{\mathrm{KL}}(\widehat{p}_k \parallel \pi)$ be the best-in-class error for each source. Then, the risk of the optimal retrained model $\widehat{\pi}_\lambda$ satisfies:*

$$D_{\mathrm{KL}}(\widehat{p}_\lambda \parallel \widehat{\pi}_\lambda) \ge \sum_{k=1}^{p} \lambda_k \epsilon_k - D_{\mathrm{JSD}}^{\lambda}(\widehat{p}_1, \ldots, \widehat{p}_p).$$

*Proof.* The proof relies on a fundamental identity for the KL divergence of a mixture. For any model $\pi$ and letting $\mathcal{X} = \cup_k \operatorname{supp}(\widehat{p}_k)$, the following equality holds:

$$\begin{aligned}
\sum_{k=1}^{p} \lambda_k D_{\mathrm{KL}}(\widehat{p}_k \parallel \pi) &= \sum_{k=1}^{p} \lambda_k \sum_{x \in \mathcal{X}} \widehat{p}_k(x) \log \frac{\widehat{p}_k(x)}{\pi(x)} \\
&= \sum_{k=1}^{p} \lambda_k \sum_{x \in \mathcal{X}} \widehat{p}_k(x) \left( \log \frac{\widehat{p}_\lambda(x)}{\pi(x)} + \log \frac{\widehat{p}_k(x)}{\widehat{p}_\lambda(x)} \right) \\
&= \sum_{x \in \mathcal{X}} \left( \sum_{k=1}^{p} \lambda_k \widehat{p}_k(x) \right) \log \frac{\widehat{p}_\lambda(x)}{\pi(x)} + \sum_{k=1}^{p} \lambda_k \left( \sum_{x \in \mathcal{X}} \widehat{p}_k(x) \log \frac{\widehat{p}_k(x)}{\widehat{p}_\lambda(x)} \right) \\
&= D_{\mathrm{KL}}(\widehat{p}_\lambda \parallel \pi) + \sum_{k=1}^{p} \lambda_k D_{\mathrm{KL}}(\widehat{p}_k \parallel \widehat{p}_\lambda) \\
&= D_{\mathrm{KL}}(\widehat{p}_\lambda \parallel \pi) + D_{\mathrm{JSD}}^{\lambda}(\widehat{p}_1, \ldots, \widehat{p}_p).
\end{aligned}$$

Rearranging gives the identity:

$$D_{\mathrm{KL}}(\widehat{p}_\lambda \parallel \pi) = \sum_{k=1}^{p} \lambda_k D_{\mathrm{KL}}(\widehat{p}_k \parallel \pi) - D_{\mathrm{JSD}}^{\lambda}(\widehat{p}_1, \ldots, \widehat{p}_p).$$

This holds for any $\pi \in \Pi$. We select the optimal model for the mixture, $\widehat{\pi}_\lambda$, and use the fact that $D_{\mathrm{KL}}(\widehat{p}_k \parallel \widehat{\pi}_\lambda) \ge \epsilon_k$ for all $k$:

$$D_{\mathrm{KL}}(\widehat{p}_\lambda \parallel \widehat{\pi}_\lambda) = \sum_{k=1}^{p} \lambda_k D_{\mathrm{KL}}(\widehat{p}_k \parallel \widehat{\pi}_\lambda) - D_{\mathrm{JSD}}^{\lambda}(\{\widehat{p}_k\}) \ge \sum_{k=1}^{p} \lambda_k \epsilon_k - D_{\mathrm{JSD}}^{\lambda}(\{\widehat{p}_k\}).$$

This completes the proof. $\qquad\square$

This theorem shows that the retrained model's performance is lower-bounded by the average best-case error $\sum_k \lambda_k \epsilon_k$, offset by the $D_{JSD}$ divergence, which measures the diversity of the sources. For a monolithic architecture, task diversity manifests as **Geometric Interference**. The model is forced to collapse distinct distributions into a single centroid. Consequently, as tasks become more dissimilar (increasing $D_{JSD}$), the monolithic performance necessarily degrades.

In the theorem, we chose $\epsilon_k$ to represent the smallest possible error. Alternatively, one can simply assume that the error of $\widehat{\pi}_\lambda$ on domain $k$ is never smaller than that of the specialist $\widehat{\pi}_k$. This is the only natural assumption required for the proof. Under this condition, the theorem establishes a fundamental limit on the retrained model: its performance is bounded below by the average specialist error minus a diversity term.

### D.2.2. STRICTNESS OF THE GAP (CONDITION FOR EQUALITY)

One might ask if the retrained model can ever overcome this bound. We show that the inequality is strict unless the tasks are degenerate.

**Proposition D.2** (Condition for Equality). *The lower bound on the retrained model's error is met with equality if and only if the optimal model for the mixture, $\widehat{\pi}_\lambda$, is simultaneously an optimal model for every individual source distribution $\widehat{p}_k$ for which $\lambda_k > 0$. That is, $D_{KL}(\widehat{p}_k \parallel \widehat{\pi}_\lambda) = \epsilon_k$ for all $k$.*

*Proof.* Assume first that $D_{KL}(\widehat{p}_k \parallel \widehat{\pi}_\lambda) = \epsilon_k$ for all $k$ with $\lambda_k > 0$. We start with the general identity for the KL divergence of a mixture:

$$D_{KL}(\widehat{p}_\lambda \parallel \widehat{\pi}_\lambda) = \sum_{k=1}^{p} \lambda_k D_{KL}(\widehat{p}_k \parallel \widehat{\pi}_\lambda) - D_{JSD}^\lambda(\widehat{p}_1, \ldots, \widehat{p}_p).$$

Substituting our assumption into this identity directly yields the desired equality:

$$D_{KL}(\widehat{p}_\lambda \parallel \widehat{\pi}_\lambda) = \sum_{k=1}^{p} \lambda_k \epsilon_k - D_{JSD}^\lambda(\widehat{p}_1, \ldots, \widehat{p}_p).$$

Assume now that the equality holds:

$$D_{KL}(\widehat{p}_\lambda \parallel \widehat{\pi}_\lambda) = \sum_{k=1}^{p} \lambda_k \epsilon_k - D_{JSD}^\lambda(\widehat{p}_1, \ldots, \widehat{p}_p).$$

By equating this assumption with the general identity from the first part, we get:

$$\sum_{k=1}^{p} \lambda_k \epsilon_k - D_{JSD}^\lambda(\{\widehat{p}_k\}) = \sum_{k=1}^{p} \lambda_k D_{KL}(\widehat{p}_k \parallel \widehat{\pi}_\lambda) - D_{JSD}^\lambda(\{\widehat{p}_k\}).$$

Canceling the $-D_{JSD}^\lambda(\{\widehat{p}_k\})$ term and rearranging gives:

$$\sum_{k=1}^{p} \lambda_k \left( D_{KL}(\widehat{p}_k \parallel \widehat{\pi}_\lambda) - \epsilon_k \right) = 0.$$

By the definition of $\epsilon_k$ as the minimum error, the term $D_{KL}(\widehat{p}_k \parallel \widehat{\pi}_\lambda) - \epsilon_k$ must be greater than or equal to zero for all $k$. Since each $\lambda_k$ is also non-negative, the expression is a sum of non-negative terms. Such a sum can only equal zero if every term is individually zero. Therefore, for every $k$ with $\lambda_k > 0$, it must be that:

$$D_{KL}(\widehat{p}_k \parallel \widehat{\pi}_\lambda) - \epsilon_k = 0,$$

which implies $D_{KL}(\widehat{p}_k \parallel \widehat{\pi}_\lambda) = \epsilon_k$. This completes the proof. $\square$

This implies that for equality to hold, the single model $\widehat{\pi}_\lambda$ must simultaneously be an optimal specialist for every individual source. This requires that the optimal specialist models themselves be identical ($\widehat{\pi}_1 = \cdots = \widehat{\pi}_p$), a scenario plausible only if the source distributions are not meaningfully distinct. Since this represents a highly restrictive, degenerate condition, we can conclude that for any practical application involving distinct sources, the equality will not hold.

### D.2.3. QUANTIFYING THE AMBIGUITY COST

We now examine how the performance gap varies as a function of how well-separated the source datasets are. Recall that the Jensen-Shannon Divergence coincides with the mutual information:

$$\mathrm{D}_{\mathrm{JSD}}^{\lambda}(\widehat{\mathsf{p}}_1, \ldots, \widehat{\mathsf{p}}_p) = I(X; K) = H(\lambda) - H(K \mid X)$$

where $H(K \mid X)$ is the conditional entropy of the source-index random variable $K$ with $P(K = k) = \lambda_k$ given $X$, and where $H(\lambda) = -\sum_{k=1}^{p} \lambda_k \log \lambda_k$ is the entropy. $H(K \mid X)$ measures the degree of *overlap* or *ambiguity* between the mixture components. It quantifies the expected uncertainty about which component $K$ generated a given sample $X$. When the supports of the component distributions are disjoint, the component index $K$ is fully determined by the observation $X$, so $H(K \mid X) = 0$. As the supports begin to overlap, certain points $x$ can be explained by multiple components, creating ambiguity and a nonzero conditional entropy $H(K \mid X = x)$. In this sense, $H(K \mid X)$ captures how much the mixture *forgets* about its origin; it represents the information lost about the latent source $K$ after observing $X$. The more indistinguishable the components become over their regions of overlap, the larger $H(K \mid X)$ grows, up to a maximum when all $\widehat{\mathsf{p}}_k$ coincide and $K$ is completely unidentifiable from $X$.

The story revealed by this analysis is not about a gap between two static bounds, but rather the dynamic behavior of a *single performance boundary* that separates the two modeling approaches. This boundary, given by $B(\lambda) = \sum_{k=1}^{p} \lambda_k \epsilon_k - \mathrm{D}_{\mathrm{JSD}}^{\lambda}$, is not fixed; its position is a direct function of the data's ambiguity, $H(K \mid X)$. It varies from $\sum_{k=1}^{p} \lambda_k \epsilon_k - H(\lambda)$ to $\sum_{k=1}^{p} \lambda_k \epsilon_k$. For perfectly separated sources where the ambiguity is zero, the boundary is at its lowest $\sum_{k=1}^{p} \lambda_k \epsilon_k - H(\lambda)$, guaranteeing a significant performance gap. As the sources begin to overlap, ambiguity grows, causing the JSD to shrink and the entire boundary to *rise* up to $\sum_{k=1}^{p} \lambda_k \epsilon_k$. This upward shift quantifies the increasing "cost of ambiguity" for both models, culminating in the boundary reaching its highest point when the sources are identical and the advantage of gating vanishes.

To make this relationship more concrete and locate a specific dataset along this trajectory, we now provide an explicit lower bound on the overlap ambiguity term $H(K \mid X)$ inside the JSD.

**Proposition D.3** (Bounds on Overlap Ambiguity). *The overlap ambiguity term $H(K \mid X)$ admits the following lower bound:*

$$H(K \mid X) \geq 1 - \sum_{k=1}^{p} \lambda_k^2 \sum_x \frac{[\widehat{\mathsf{p}}_k(x)]^2}{\widehat{\mathsf{p}}_{\lambda}(x)}.$$

*Proof.* The conditional entropy $H(K \mid X)$ can be expressed as follows:

$$H(K \mid X) = -\sum_x P_X(x) \sum_{k=1}^{p} P(K = k \mid X = x) \log P(K = k \mid X = x)$$

$$= -\sum_x \widehat{\mathsf{p}}_{\lambda}(x) \sum_{k=1}^{p} \frac{\lambda_k \widehat{\mathsf{p}}_k(x)}{\widehat{\mathsf{p}}_{\lambda}(x)} \log \frac{\lambda_k \widehat{\mathsf{p}}_k(x)}{\widehat{\mathsf{p}}_{\lambda}(x)}$$

$$= \sum_{k=1}^{p} \sum_x \lambda_k \widehat{\mathsf{p}}_k(x) \log \frac{\widehat{\mathsf{p}}_{\lambda}(x)}{\lambda_k \widehat{\mathsf{p}}_k(x)}.$$

Thus, we have $H(K \mid X) = \sum_k \lambda_k \sum_x \widehat{\mathsf{p}}_k(x) \log(1 + t_k(x))$, where $t_k(x) = \frac{\sum_{j \neq k} \lambda_j \widehat{\mathsf{p}}_j(x)}{\lambda_k \widehat{\mathsf{p}}_k(x)}$. Using the inequality $\log(1 + t) \geq \frac{t}{1+t}$ for $t \geq 0$:

$$H(K \mid X) \geq \sum_{k=1}^{p} \lambda_k \sum_x \widehat{\mathsf{p}}_k(x) \left( \frac{\sum_{j \neq k} \lambda_j \widehat{\mathsf{p}}_j(x)}{\lambda_k \widehat{\mathsf{p}}_k(x) + \sum_{j \neq k} \lambda_j \widehat{\mathsf{p}}_j(x)} \right)$$

$$= \sum_k \lambda_k \sum_x \widehat{\mathsf{p}}_k(x) \left( \frac{\widehat{\mathsf{p}}_{\lambda}(x) - \lambda_k \widehat{\mathsf{p}}_k(x)}{\widehat{\mathsf{p}}_{\lambda}(x)} \right)$$

$$= \sum_x \frac{1}{\widehat{\mathsf{p}}_{\lambda}(x)} \sum_{k=1}^{p} \left( \lambda_k \widehat{\mathsf{p}}_k(x) \widehat{\mathsf{p}}_{\lambda}(x) - \lambda_k^2 [\widehat{\mathsf{p}}_k(x)]^2 \right)$$

$$= \sum_x \frac{\widehat{\mathsf{p}}_{\lambda}(x) \left( \sum_k \lambda_k \widehat{\mathsf{p}}_k(x) \right) - \sum_k \lambda_k^2 [\widehat{\mathsf{p}}_k(x)]^2}{\widehat{\mathsf{p}}_{\lambda}(x)}$$

$$= \sum_x \frac{[\widehat{\mathsf{p}}_{\lambda}(x)]^2 - \sum_k \lambda_k^2 [\widehat{\mathsf{p}}_k(x)]^2}{\widehat{\mathsf{p}}_{\lambda}(x)}.$$

The Chi-squared-like form follows from splitting the fraction: $\sum_x \widehat{\mathsf{p}}_\lambda(x) - \sum_x \frac{\sum_k \lambda_k^2 [\widehat{\mathsf{p}}_k(x)]^2}{\widehat{\mathsf{p}}_\lambda(x)} = 1 - \sum_k \lambda_k^2 \sum_x \frac{[\widehat{\mathsf{p}}_k(x)]^2}{\widehat{\mathsf{p}}_\lambda(x)}$. $\qquad\square$

By substituting this explicit lower bound for the ambiguity $H(K \mid X)$ back into our main theorem, we arrive at a concrete guarantee for the retrained model $\mathsf{D}_{\mathrm{KL}}(\widehat{\mathsf{p}}_\lambda \parallel \widehat{\pi}_\lambda) \geq$:

$$\sum_{k=1}^p \lambda_k \epsilon_k - H(\lambda) + \left( \sum_x \frac{\sum_{i \neq j} \lambda_i \lambda_j \widehat{\mathsf{p}}_i(x) \widehat{\mathsf{p}}_j(x)}{\widehat{\mathsf{p}}_\lambda(x)} \right).$$

The last term, an interaction score, is a non-negative quantity that is zero if and only if the sources are disjoint. This final inequality makes the counter-intuitive conclusion undeniable: the guaranteed error of a single retrained model does not decrease with overlap, but rather increases by a computable amount directly related to the "cross-talk" between conflicting sources. This interaction score is the tangible price the single model pays for ambiguity, a price the gated model avoids, thus making the gated model's advantage even more pronounced in the face of messy, overlapping data.

### D.2.4. THE MODULAR ADVANTAGE: DECOUPLING

In contrast, the Modular Gating network effectively inverts this relationship. Theorem 3.3 establishes that the risk of the robust gate is bounded by:

$$\mathrm{Risk}_{\mathrm{mod}} \leq \underbrace{\log\left(\sum e^{\epsilon_k}\right)}_{\text{Capacity Cost}} - \underbrace{\mathsf{D}_{\mathrm{JSD}}^\lambda(\widehat{\mathsf{p}}_1, \ldots, \widehat{\mathsf{p}}_p)}_{\text{Separability Gain}} - \mathrm{Overlap}.$$

Here, the divergence term appears with a *negative* sign. For the modular system, task diversity acts as a **Geometric Gain**.

**The Decoupling Hypothesis.** This analysis identifies a structural phase transition.

- **High Diversity** ($\mathsf{D}_{\mathrm{JSD}} \gg 0$)**:** When tasks are distinct (disjoint supports), the Monolithic model hits the interference floor (Risk $\geq \mathsf{D}_{\mathrm{JSD}}$). However, for the Modular model, the separability gain becomes maximal ($\mathsf{D}_{\mathrm{JSD}}^\lambda \to H(\lambda)$), effectively canceling the capacity cost. The bound converges to $V^* \leq \max_k \epsilon_k$.

While the explicit diversity "discount" disappears, the result is profound: the modular architecture has successfully **decoupled** the tasks. The risk is no longer lower-bounded by the geometry ($\mathsf{D}_{\mathrm{JSD}}$), but is instead upper-bounded strictly by the intrinsic error of the experts.

### D.3. Safety in Convex Regimes: A Geometric Perspective

One might ask: does modularity sacrifice performance when tasks are simple and compatible? We prove that in convex settings, the answer is no. We assume $\Pi$ is a linear model family (e.g., exponential families).

Our analysis leverages the *Pythagorean equality theorems* of (Csiszár, 1975) and (Csiszár & Matus, 2003), which hold for *linear models* defined as follows. We assume throughout this section that $\Pi$ is a *linear model* in the sense of (Csiszár, 1975). That is, there exists a finite-dimensional vector space of statistics $\phi = (\phi_1, \ldots, \phi_m)$ and an open convex set $\mathcal{E} \subseteq \mathbb{R}^m$ such that $\Pi = \{\pi \in \mathcal{P}(\mathcal{X}) : \mathbb{E}_\pi[\phi] \in \mathcal{E}\}$. The map $\eta(\pi) = \mathbb{E}_\pi[\phi]$ is called the *expectation parameter*. The set $\mathcal{E}$ is affine, and the mapping $\pi \mapsto \eta(\pi)$ is one-to-one in the interior of $\Pi$. The family $\Pi$ thus includes both affine families and regular exponential families in their expectation parametrization.

For any $\mathsf{p}$, we denote by $\pi^*(\mathsf{p})$ its *I-projection* on $\Pi$, that is $\pi^*(\mathsf{p}) = \mathrm{argmin}_{\pi \in \Pi} \mathsf{D}_{\mathrm{KL}}(\mathsf{p} \parallel \pi)$. The following fundamental equality is due to (Csiszár, 1975; Csiszár & Matus, 2003).

**Lemma D.4** (Pythagorean equality for linear models)**.** *For any $\mathsf{p}$ and any $\pi \in \Pi$,*

$$\mathsf{D}_{\mathrm{KL}}(\mathsf{p} \parallel \pi) = \mathsf{D}_{\mathrm{KL}}(\mathsf{p} \parallel \pi^*(\mathsf{p})) + \mathsf{D}_{\mathrm{KL}}(\pi^*(\mathsf{p}) \parallel \pi).$$

This equality characterizes the linear models: it expresses the fact that the projection $\pi^*(\mathsf{p})$ is orthogonal to $\Pi$ in the geometry induced by $\mathsf{D}_{\mathrm{KL}}$.

**Theorem 3.4** (Gated Model Coincides with Retraining)**.** *Let $\Pi$ be a linear model. For a mixture $\widehat{\mathsf{p}}_\lambda$, let $\pi_k = \pi^*(\widehat{\mathsf{p}}_k)$ denote the projection of each component. Then, the best model trained on the mixture $\pi^*(\widehat{\mathsf{p}}_\lambda)$ coincides exactly with the gated mixture of the best component models:*

$$\pi^*(\widehat{\mathsf{p}}_\lambda) = \sum_{k=1}^p \lambda_k \pi_k.$$

*Proof.* By the Pythagorean equality (Lemma D.4) for linear models applied to each component $\widehat{p}_k$ and to an arbitrary model $\pi \in \Pi$; denoting $\pi_k = \pi^*(\widehat{p}_k)$ we obtain for each $k$,

$$D_{\mathrm{KL}}(\widehat{p}_k \parallel \pi) = D_{\mathrm{KL}}(\widehat{p}_k \parallel \pi_k) + D_{\mathrm{KL}}(\pi_k \parallel \pi).$$

Multiplying by $\lambda_k$ and summing up yields

$$\sum_{k=1}^{p} \lambda_k D_{\mathrm{KL}}(\widehat{p}_k \parallel \pi) = \sum_{k=1}^{p} \lambda_k D_{\mathrm{KL}}(\widehat{p}_k \parallel \pi_k) + \sum_{k=1}^{p} \lambda_k D_{\mathrm{KL}}(\pi_k \parallel \pi). \tag{13}$$

The functions $\pi \mapsto D_{\mathrm{KL}}(\widehat{p}_\lambda \parallel \pi)$ and $\pi \mapsto \sum_k \lambda_k D_{\mathrm{KL}}(\widehat{p}_k \parallel \pi)$ differ by a constant independent of $\pi$. Specifically,

$$\sum_{k=1}^{p} \lambda_k D_{\mathrm{KL}}(\widehat{p}_k \parallel \pi) - D_{\mathrm{KL}}(\widehat{p}_\lambda \parallel \pi) = D_{\mathrm{JSD}}^\lambda(\widehat{p}_1, \ldots, \widehat{p}_p).$$

This constant difference implies they share the same unique minimizer over $\pi \in \Pi$. The minimizer of $\pi \mapsto \sum_k \lambda_k D_{\mathrm{KL}}(\widehat{p}_k \parallel \pi)$ over $\pi \in \Pi$ is the same as the minimizer of $\pi \mapsto \sum_k \lambda_k D_{\mathrm{KL}}(\pi_k \parallel \pi)$, which is equivalent to minimizing $-\mathbb{E}_{\pi_g}[\log \pi]$, where $\pi_g = \sum_k \lambda_k \pi_k$. Since $\Pi$ is a linear model, the convex combination $\pi_g$ lies in $\Pi$. Therefore, the unique minimizer of $D_{\mathrm{KL}}(\pi_g \parallel \pi)$ is $\pi = \pi_g$. This establishes that $\pi_g$ is the minimizer of $D_{\mathrm{KL}}(\widehat{p}_\lambda \parallel \pi)$, proving the first claim:

$$\pi_g = \pi^*(\widehat{p}_\lambda).$$

Moreover, setting $\pi = \pi_g$ in (13) yields the following useful error decomposition:

$$\sum_{k=1}^{p} \lambda_k D_{\mathrm{KL}}(\widehat{p}_k \parallel \pi_g) = \sum_{k=1}^{p} \lambda_k D_{\mathrm{KL}}(\widehat{p}_k \parallel \pi_k) + \sum_{k=1}^{p} \lambda_k D_{\mathrm{KL}}(\pi_k \parallel \pi_g).$$

The second term on the right-hand side is, by definition, the Jensen-Shannon divergence of the projections, $D_{\mathrm{JSD}}^\lambda(\pi_1, \ldots, \pi_p)$. The left-hand side can be rewritten using the constant-shift identity we established earlier:

$$\sum_{k=1}^{p} \lambda_k D_{\mathrm{KL}}(\widehat{p}_k \parallel \pi_g) = D_{\mathrm{KL}}(\widehat{p}_\lambda \parallel \pi_g) + D_{\mathrm{JSD}}^\lambda(\widehat{p}_1, \ldots, \widehat{p}_p).$$

Combining these two equalities gives the decomposition:

$$D_{\mathrm{KL}}(\widehat{p}_\lambda \parallel \pi_g) = \sum_{k=1}^{p} \lambda_k D_{\mathrm{KL}}(\widehat{p}_k \parallel \pi_k) + D_{\mathrm{JSD}}^\lambda(\pi_1, \ldots, \pi_p) - D_{\mathrm{JSD}}^\lambda(\widehat{p}_1, \ldots, \widehat{p}_p).$$

$\square$

The proof also yields the following decomposition of the gated model's error.

**Corollary D.5** (Error Decomposition for Linear Models). *Under the assumptions of Theorem 3.4, the error of the gated model decomposes as:*

$$D_{\mathrm{KL}}(\widehat{p}_\lambda \parallel \pi^*(\widehat{p}_\lambda)) = \sum_{k=1}^{p} \lambda_k D_{\mathrm{KL}}(\widehat{p}_k \parallel \pi_k) + D_{\mathrm{JSD}}^\lambda(\pi_1, \ldots, \pi_p) - D_{\mathrm{JSD}}^\lambda(\widehat{p}_1, \ldots, \widehat{p}_p).$$

This theorem ensures that modularity is a "safe" architectural prior: it loses nothing in convex regimes while providing strictly superior robustness guarantees in the presence of conflicting, non-convex distributions. The best model trained on the mixture $\widehat{p}_\lambda$ coincides with the gated model obtained by mixing the best component models $\pi_k$. In particular, retraining on the mixture does not improve upon gating.

### D.4. The Generalization Gap: Sample Efficiency

Beyond optimization, modularity offers a distinct statistical advantage. The guarantees for the robust gate transfer efficiently from empirical data to the true population. To establish these generalization bounds, we first introduce the necessary assumption and formally define the vector-valued Rademacher complexity.

**Assumption D.6** (Bounded Expert Likelihoods). We assume that the pre-trained experts are bounded away from zero on the support of the data distribution. That is, there exists a constant $M > 0$ such that for all $x \in \mathcal{X}_0$ and all $k \in [1, p]$, the negative log-likelihood is bounded:

$$|\log \widehat{\pi}_k(x)| \leq M.$$

This implies that for any valid probability mixture $\pi_g(x)$, the probability is lower-bounded by $e^{-M}$.

In the context of Large Language Models (LLMs) with finite vocabulary sizes, this assumption is naturally satisfied. The standard Softmax function yields strictly positive probabilities for all tokens. Unless an expert explicitly masks a token to $-\infty$ (assigning it zero probability), the log-likelihood remains finite.

**Vector-Valued Rademacher Complexity.** Let $\mathcal{F}$ be a class of functions mapping $\mathcal{X}_0$ to $\mathbb{R}^p$ and denote by $f_j$ the $j$-th component of $f \in \mathcal{F}$. The empirical Rademacher complexity associated to such a vector-valued class of functions is defined by

$$\widehat{\mathfrak{R}}_S(\mathcal{F}) = \frac{1}{m} \mathbb{E}_{\boldsymbol{\sigma}} \left[ \sup_{f \in \mathcal{F}} \sum_{i=1}^{m} \sum_{j=1}^{p} \sigma_{i,j} f_j(x_i) \right],$$

with $\sigma_{i,j}$'s independent Rademacher variables, that is independent uniformly distributed random variables taking values in $\{-1, +1\}$. Its expectation, $\mathfrak{R}_m(\mathcal{F}) = \mathbb{E}_S[\widehat{\mathfrak{R}}_S(\mathcal{F})]$, is the Rademacher complexity of $\mathcal{F}$. Note that for $p = 1$, this coincides with the standard notion of Rademacher complexity for real-valued functions. The vectorial extension is also called *factor graph Rademacher complexity* in (Cortes et al., 2016b), in the context of structured prediction.

We will denote by $\mathsf{p}_k$ the true distribution according to which the dataset $D_k$ is drawn (or a sample $S_k$), and will denote by $\mathfrak{R}_{m_k}^k(\mathcal{G}_1)$ the Rademacher complexity of $\mathcal{G}_1$ for the distribution $\mathsf{p}_k$ and a sample size $m_k$. Note that Theorem 3.5 considers the simplified case where all experts have equal sample sizes $m_k = m$ and uniform complexity across distributions.

**Theorem 3.5** (Generalization Bound for Modular Gate Models). *Under Assumption D.6, for any $\delta > 0$, with probability at least $1 - \delta$ over the draw of samples $S_k \sim \mathsf{p}_k^m$, the following inequality holds simultaneously for all $g \in \mathcal{G}_1$ and $\lambda \in \Delta$: for the generalization gap $\Gamma = \mathbb{E}_{x \sim \mathsf{p}_\lambda}[-\log \pi_g(x)] - \mathbb{E}_{x \sim \widehat{\mathsf{p}}_\lambda}[-\log \pi_g(x)]$: $\Gamma \leq 2\sqrt{2} C_\Pi e^M \mathfrak{R}_m(\mathcal{G}_1) + M \sqrt{\frac{\log(p/\delta)}{2m}}$, where $\mathfrak{R}_m(\mathcal{G}_1)$ is the Rademacher complexity of the gate class and $C_\Pi = \sup_x \|(\widehat{\pi}_1(x), \ldots, \widehat{\pi}_p(x))\|_2$ is the* expert coincidence norm.

*Proof.* Let $\mathcal{L}_{\mathcal{G}_1} = \{x \mapsto -\log \pi_g(x) \mid g \in \mathcal{G}_1\}$ be the loss class associated with the gating hypothesis space $\mathcal{G}_1$. For a fixed $k \in [p]$, by the standard Rademacher complexity generalization bounds (see e.g., Mohri et al., 2018) for functions bounded by $M$, for any $\delta > 0$, with probability at least $1 - \delta$, the following inequality holds for all $g \in \mathcal{G}_1$:

$$\mathbb{E}_{x \sim \mathsf{p}_k}[-\log \pi_g(x)] \leq \mathbb{E}_{x \sim \widehat{\mathsf{p}}_k}[-\log \pi_g(x)] + 2\mathfrak{R}_{m_k}^k(\mathcal{L}_{\mathcal{G}_1}) + M\sqrt{\frac{\log \frac{1}{\delta}}{2m_k}}.$$

Thus, by the union bound, the following inequality holds for all $k \in [p]$ with probability at least $1 - \delta$:

$$\mathbb{E}_{x \sim \mathsf{p}_k}[-\log \pi_g(x)] \leq \mathbb{E}_{x \sim \widehat{\mathsf{p}}_k}[-\log \pi_g(x)] + 2\mathfrak{R}_{m_k}^k(\mathcal{L}_{\mathcal{G}_1}) + M\sqrt{\frac{\log \frac{p}{\delta}}{2m_k}}.$$

Multiplying each inequality by $\lambda_k$ and summing up yields that with probability at least $1 - \delta$, the following holds for all $g \in \mathcal{G}_1$ and $\lambda \in \Delta$:

$$\mathbb{E}_{x \sim \mathsf{p}_\lambda}[-\log \pi_g(x)] \leq \mathbb{E}_{x \sim \widehat{\mathsf{p}}_\lambda}[-\log \pi_g(x)] + \sum_{k=1}^{p} \lambda_k \left[ 2\mathfrak{R}_{m_k}^k(\mathcal{L}_{\mathcal{G}_1}) + M\sqrt{\frac{\log \frac{p}{\delta}}{2m_k}} \right].$$

We now bound $\mathfrak{R}_{m_k}^k(\mathcal{L}_{\mathcal{G}_1})$ in terms of $\mathfrak{R}_{m_k}^k(\mathcal{G}_1)$, using the vector contraction established in (Cortes et al., 2016b) (Lemma A.1) and (Maurer, 2016). This inequality holds for $\ell_2$-Lipschitz functions.

We view the gate function class $\mathcal{G}_1$ as a vector-valued hypothesis class mapping inputs $x \in \mathcal{X}_0$ to the simplex $\Delta \subset \mathbb{R}^p$. The loss function for a function $g \in \mathcal{G}_1$ and a sample $x_i$ is defined as $-\log(\pi_g(x_i)) = \Psi_i(\mathbf{u}) = -\log(\mathbf{u} \cdot \widehat{\boldsymbol{\pi}}(x_i))$, where

$\widehat{\boldsymbol{\pi}}(x_i) = (\widehat{\pi}_1(x_i), \ldots, \widehat{\pi}_p(x_i))$ and $\mathbf{u} = (g(x_i, 1), \ldots, g(x_i, p))$. Under Assumption D.6, for any fixed sample $x_i$, $\Psi_i$ is Lipschitz continuous with respect to the $\ell_2$ norm. The gradient of $\Psi_i$ with respect to the vector $\mathbf{u}$ is:

$$\nabla \Psi_i(\mathbf{u}) = \frac{-1}{\mathbf{u} \cdot \widehat{\boldsymbol{\pi}}(x_i)} \widehat{\boldsymbol{\pi}}(x_i).$$

To determine the Lipschitz constant, we examine the $\ell_2$ norm of the gradient:

$$\|\nabla \Psi_i(\mathbf{u})\|_2 = \frac{\|\widehat{\boldsymbol{\pi}}(x_i)\|_2}{|\mathbf{u} \cdot \widehat{\boldsymbol{\pi}}(x_i)|}.$$

First, consider the numerator. By the definition of the coincidence norm, we have $\|\widehat{\boldsymbol{\pi}}(x_i)\|_2 \leq C_\Pi$. Second, consider the denominator. By Assumption D.6, the mixture probability is lower-bounded by $e^{-M}$. Thus:

$$\|\nabla \Psi_i(\mathbf{u})\|_2 \leq \frac{C_\Pi}{e^{-M}} = C_\Pi \, e^M.$$

The function is therefore $C_\Pi \, e^M$-Lipschitz with respect to the $\ell_2$ norm. Thus, by the vector contraction lemma, we have

$$\mathfrak{R}_{m_k}^k(\mathcal{L}_{\mathcal{G}_1}) \leq \sqrt{2} C_\Pi \, e^M \mathfrak{R}_{m_k}^k(\mathcal{G}_1).$$

Plugging in the right-hand side in the inequality previously proven completes the proof. $\qquad\square$

### D.4.1. COMPARISON WITH MONOLITHIC GENERALIZATION

Let $\widehat{\pi}_{\text{scratch}}$ be a monolithic model retrained on the aggregate data. Standard learning theory bounds its generalization error by the complexity of the full model class $\Pi$:

$$\text{GenGap}(\widehat{\pi}_{\text{scratch}}) \approx O(\mathfrak{R}_m(\Pi)).$$

In contrast, our modular solution scales with $\mathfrak{R}_m(\mathcal{G}_1)$. Since the gate is typically a lightweight network (e.g., a shallow Transformer with $10^6$ parameters) compared to the massive experts (LLMs with $10^9+$ parameters), we have $\mathfrak{R}_m(\mathcal{G}_1) \ll \mathfrak{R}_m(\Pi)$. The term $C_\Pi$ measures the effective overlap of the experts. In the worst case of redundant experts, $C_\Pi \approx \sqrt{p}$. However, in the ideal modular setting where experts are specialized (disjoint supports), $C_\Pi \approx 1$. Thus, even with the overlap factor, the modular approach requires significantly fewer samples to learn a robust policy than retraining a monolithic model.

## E. Optimization and Algorithms

Let us analyze the $\min_g \max_\lambda$ game with this new payoff. The $\lambda$-player seeks to maximize $\widetilde{L}(\lambda, g)$ over $\lambda \in \Delta([1, p])$. Since $\widetilde{L}(\lambda, g)$ is linear in $\lambda$, its maximum is also achieved at a vertex. This gives:

$$\max_{\lambda \in \Delta([1,p])} \widetilde{L}(\lambda, g) = \max_{k \in [1,p]} \mathsf{D}_{\text{KL}}(\widehat{\mathsf{p}}_k \parallel \pi_g).$$

This is the *exact same* objective function for the $g$-player as in our equivalent problem. Therefore, the solution to this new game is the same as the solution to our original problem:

$$\min_{g \in \mathcal{G}_1} \max_{\lambda \in \Delta} \widetilde{L}(\lambda, g) = \min_{g \in \mathcal{G}_1} \max_{k \in [1,p]} \mathsf{D}_{\text{KL}}(\widehat{\mathsf{p}}_k \parallel \pi_g) = \min_{g \in \mathcal{G}_1} \max_{\lambda \in \Delta} L(\lambda, g).$$

This new game $\widetilde{L}(\lambda, g)$ is convex-concave:

- Convex in $g$: $\widetilde{L}$ is a positive, weighted sum of functions $\mathsf{D}_{\text{KL}}(\widehat{\mathsf{p}}_k \parallel \pi_g)$, each of which is convex in $g$.

- Concave in $\lambda$: $\widetilde{L}$ is linear in $\lambda$, which is a special case of concavity.

We can therefore find the robust solution $g^*$ by applying standard no-regret algorithms to this tractable convex-concave game $\widetilde{L}(\lambda, g)$.

---

**Algorithm 1** Robust Gate via No-Regret Dynamics (EG + OGD)

---

1: **Input:** Models $\widehat{\pi}_1, \ldots, \widehat{\pi}_p$; datasets $D_1, \ldots, D_p$; learning rates $\eta_\lambda, \eta_g$; iterations $T$.
2: **Initialize:** $\lambda_0(k) = 1/p$; $g_0(x, k) = 1/p$ for all $x, k$.
3: **for** $t = 0$ **to** $T - 1$ **do**
4:     (**Compute Gains**) For each expert $k \in [1, p]$, compute its gain for the $\lambda$-player: $\ell_t(k) = D_{\mathrm{KL}}(\widehat{p}_k \| \pi_{g_t})$.
5:     ($\lambda$-**update**) Update mixture weights using Exponentiated Gradient on gains $\ell_t$: $\lambda_{t+1}(k) \propto \lambda_t(k) \exp(\eta_\lambda \ell_t(k))$ and re-normalize.
6:     ($g$-**update**) Construct mixture $\widehat{p}_{\lambda_{t+1}}$ from new $\lambda_{t+1}$.
7:     Compute gradient $v_t(x, k) = -(\widehat{p}_{\lambda_{t+1}}(x)/\pi_{g_t}(x))\widehat{\pi}_k(x)$ for all $(x, k)$.
8:     Compute intermediate update $g'_{t+1} = g_t - \eta_g v_t$.
9:     Project onto constrained space: $g_{t+1} = \Pi_{\mathcal{G}_1}(g'_{t+1})$.
10: **end for**
11: **Output:** The time-averaged gate $\overline{g}_T = \frac{1}{T}\sum_{t=1}^{T} g_t$.

---

## E.1. Game Dynamics and Reformulation

### E.1.1. ALGORITHM AND PLAYER DYNAMICS

The algorithm simulates the game $\widetilde{L}(\lambda, g)$ where two players, a $\lambda$-player and a $g$-player, iteratively update their strategies.

The $\lambda$-player (maximizer) chooses a mixture distribution $\lambda_t \in \Delta([1, p])$. This is a standard "learning from experts" problem. At each round, given the opponent's gate $g_t$, the "gain" associated with selecting expert $k$ is its contribution to the payoff:

$$\ell_t(k) = D_{\mathrm{KL}}(\widehat{p}_k \| \pi_{g_t}).$$

The $\lambda$-player's goal is to find the mixture $\lambda$ that maximizes $\sum_k \lambda_k \ell_t(k)$. We use the Exponentiated Gradient algorithm, which updates the weights to favor experts with higher gain:

$$\lambda_{t+1}(k) = \frac{\lambda_t(k) \exp(\eta_\lambda \ell_t(k))}{\sum_{j=1}^{p} \lambda_t(j) \exp(\eta_\lambda \ell_t(j))}.$$

The $g$-player (minimizer) chooses a gate $g_t \in \mathcal{G}$. Given the $\lambda$-player's new mixture $\lambda_{t+1}$, its goal is to minimize the loss $\widetilde{L}(\lambda_{t+1}, g_t)$. The $g$-player updates its gate using Online Gradient Descent. The gradient of the loss $\widetilde{L}(\lambda_{t+1}, g_t)$ with respect to the gate weights $g_t(x, k)$ is:

$$
\begin{aligned}
v_t(x, k) &= \nabla_{g_t(x,k)} \widetilde{L}(\lambda_{t+1}, g_t) \\
&= \nabla_{g_t(x,k)} \sum_{j=1}^{p} \lambda_{t+1}(j) D_{\mathrm{KL}}(\widehat{p}_j \| \pi_{g_t}) \\
&= \sum_{j=1}^{p} \lambda_{t+1}(j) \nabla_{g_t(x,k)} D_{\mathrm{KL}}(\widehat{p}_j \| \pi_{g_t}) \\
&= \sum_{j=1}^{p} \lambda_{t+1}(j)\left(-\frac{\widehat{p}_j(x)\widehat{\pi}_k(x)}{\pi_{g_t}(x)}\right) \\
&= -\frac{\left(\sum_{j=1}^{p} \lambda_{t+1}(j)\widehat{p}_j(x)\right)\widehat{\pi}_k(x)}{\pi_{g_t}(x)} \\
&= -\frac{\widehat{p}_{\lambda_{t+1}}(x)\widehat{\pi}_k(x)}{\pi_{g_t}(x)}.
\end{aligned}
$$

The gate is then updated by taking a step in the negative gradient direction and projecting back onto the feasible set, for each $x \in \mathcal{X}_0$:

$$g_{t+1}(x, \cdot) \leftarrow \Pi_{\mathcal{G}_1}(g_t(x, \cdot) - \eta_g v_t(x, \cdot)).$$

The final solution is the time-average of the iterates, as this is what is guaranteed to converge to the equilibrium: $\overline{g}_T = \frac{1}{T}\sum_{t=1}^{T} g_t$.

### E.1.2. PROJECTION CHALLENGE

The primary difficulty lies in Line 9 of Algorithm 1. The projection $\Pi_{\mathcal{G}_1}$ requires finding the closest point $g$ to $g'_{t+1}$ (typically in Euclidean norm) such that $g$ satisfies both:

1. Local normalization: $g(x, \cdot) \in \Delta([1, p])$ for all $x \in \mathcal{X}_0$.

2. Global normalization: $\sum_{x \in \mathcal{X}_0} \sum_{k=1}^{p} g(x, k) \widehat{\pi}_k(x) = 1$.

This is a projection onto the intersection of the product space of simplices $\mathcal{G}$ and an affine subspace defined by the linear equality $Z_g = 1$.

### E.2. Convergence analysis

Assuming an efficient oracle for the projection $\Pi_{\mathcal{G}_1}$, the standard convergence theorems for no-regret dynamics in convex-concave games apply to our reformulated game $\widetilde{L}(\lambda, g)$.

**Theorem 4.1** (Convergence of Dynamics). *Let $\widetilde{V} = \min_{g \in \mathcal{G}_1} \max_{\lambda \in \Delta} \widetilde{L}(\lambda, g)$. If the projection $\Pi_{\mathcal{G}_1}$ onto the normalized gate space can be computed, then with step sizes $\eta_\lambda \propto 1/\sqrt{T}$ and $\eta_g \propto 1/\sqrt{T}$, the time-averaged gate $\overline{g}_T$ converges to the optimal robust solution:*

$$\max_{\lambda \in \Delta} \widetilde{L}(\lambda, \overline{g}_T) - \widetilde{V} \leq O\left(\sqrt{\frac{\log p}{T}}\right).$$

*Proof.* The proof relies on the regret bounds for the players' algorithms and the connection between average regret and the duality gap for time-averaged strategies in convex-concave games. Let $R_T^\lambda$ be the regret of the $\lambda$-player (using EG) and $R_T^g$ be the regret of the $g$-player (using OGD over $\mathcal{G}_1$). Standard bounds yield: $R_T^\lambda \leq M_\lambda \sqrt{2T \log p}$ and $R_T^g \leq D_{\mathcal{G}_1} M_g \sqrt{T}$. Let $\overline{\lambda}_T = \frac{1}{T} \sum_{t=1}^{T} \lambda_t$ and $\overline{g}_T = \frac{1}{T} \sum_{t=1}^{T} g_t$. By the properties of convex-concave games, the duality gap of the time-averaged strategies is bounded by the sum of average regrets:

$$\max_{\lambda \in \Delta} \widetilde{L}(\lambda, \overline{g}_T) - \min_{g \in \mathcal{G}_1} \widetilde{L}(\overline{\lambda}_T, g) \leq \frac{R_T^\lambda + R_T^g}{T}.$$

The value of the game is $\widetilde{V} = \min_g \max_\lambda \widetilde{L}(\lambda, g)$. By weak duality, we know that for any $\overline{\lambda}_T$, $\min_{g \in \mathcal{G}_1} \widetilde{L}(\overline{\lambda}_T, g) \leq \widetilde{V}$. Therefore, we can bound the suboptimality of $\overline{g}_T$:

$$\begin{aligned}
\max_{\lambda \in \Delta} \widetilde{L}(\lambda, \overline{g}_T) - \widetilde{V} &\leq \max_{\lambda \in \Delta} \widetilde{L}(\lambda, \overline{g}_T) - \min_{g \in \mathcal{G}_1} \widetilde{L}(\overline{\lambda}_T, g) \\
&\leq \frac{R_T^\lambda + R_T^g}{T} \\
&\leq \frac{M_\lambda \sqrt{2T \log p} + D_{\mathcal{G}_1} M_g \sqrt{T}}{T} \\
&= \frac{D_{\mathcal{G}_1} M_g}{\sqrt{T}} + \frac{M_\lambda \sqrt{2 \log p}}{\sqrt{T}}.
\end{aligned}$$

This proves convergence of the time-averaged strategy $\overline{g}_T$ to the value of the game $\widetilde{V}$. As established in the reformulation, $\widetilde{V}$ is the value of the original problem, and thus $\overline{g}_T$ is a near-optimal robust gate. $\square$

The analysis extends to the case where a subset $\Lambda \subseteq \Delta([1, p])$ is used (Section C.1).

**Theorem E.1** (Algorithmic Convergence on $\Lambda$). *Let $\widetilde{V}_\Lambda = \min_{g \in \mathcal{G}_1} \max_{\lambda \in \Lambda} \widetilde{L}(\lambda, g)$ be the value of the restricted convex-concave game, where $\widetilde{L}(\lambda, g) = \sum_k \lambda_k D_{\mathrm{KL}}(\widehat{p}_k \| \pi_g)$. Let $\overline{g}_T$ be the time-averaged gate obtained by running the no-regret algorithm (Algorithm 1, modified with $\lambda$-updates projected onto $\Lambda$). Assume the gains and gradients are bounded. Then, the algorithm admits the following convergence guarantee:*

$$\mathbb{E}\left[\max_{\lambda \in \Lambda} \widetilde{L}(\lambda, \overline{g}_T)\right] - \widetilde{V}_\Lambda \leq O\left(\sqrt{\frac{\log p}{T}}\right).$$

*Proof.* The proof is a direct extension of Theorem 4.1. The game remains convex-concave. The $g$-player's algorithm and regret bound are unchanged. The $\lambda$-player now runs a projected online mirror descent (OMD) algorithm, specifically, Exponentiated Gradient with a projection. The regret of this algorithm is still bounded relative to the best fixed strategy in $\Lambda \subseteq \Delta([1, p])$. Since the OMD algorithm uses the negative entropy regularizer (which leads to EG), the regret bound remains $R_T^\lambda \le O(\sqrt{T \log p})$. The standard analysis bounding the duality gap by the average regret, $\frac{1}{T}(R_T^\lambda + R_T^g)$, applies directly, yielding the $O(\sqrt{\log p / T})$ convergence rate. $\qquad\square$

### E.3. Leveraging the Least-Favorable Mixture

Our minimax solution provides two crucial outputs: the robust gate $g^*$, and the least-favorable mixture $\lambda^*$. It is important to note that $\lambda^*$ corresponds to the saddle point of the *linearized* game $\widetilde{L}(\lambda, g) = \sum_k \lambda_k D_{\mathrm{KL}}(\widehat{\mathsf{p}}_k \parallel \pi_g)$, rather than the original convex-convex payoff $L$.

Consequently, $\lambda^*$ identifies the specific weighting of source domains that is maximally challenging for the modular ensemble in the linearized regime. This insight is highly valuable in practical scenarios where engineers must train a single, static model $\widehat{\pi}$ on the aggregated data $\widehat{\mathsf{p}}_\lambda$ (e.g., for reduced inference latency). Instead of resorting to heuristic choices for the mixture weights $\lambda$ (such as uniform $\lambda_k = 1/p$ or weights based on dataset size), the optimal $\lambda^*$ resulting from the no-regret algorithm provides a statistically principled alternative.

By training a new model on the data mixture $\widehat{\mathsf{p}}_{\lambda^*} = \sum_k \lambda_k^* \widehat{\mathsf{p}}_k$, the resulting model $\widehat{\pi}_{\lambda^*}$ is optimized for the distribution where the underlying expert ensemble is most vulnerable (in terms of the upper bound $\widetilde{L}$). This strategy effectively turns the worst-case scenario for the gated model into the training objective for the static model.

### E.4. Convergence of Primal-Dual Dynamics: Proof of Theorem 4.2

The pseudocode of the Primal-Dual algorithm is given in Algorithm 2.

---

**Algorithm 2** Stochastic Primal-Dual Training Loop

---

1: **for** iteration $t = 1$ **to** $T$ **do**
2:     **Data Sampling:** Sample batch $B = \cup_k B_k$ from experts.
3:     **Forward Pass:** Compute logits $g_\theta(x)$ and expert log-probs for $x \in B$.
4:     **Constraint Est.:** Estimate $\widehat{Z} \approx \frac{1}{|B|} \sum_{x \in B} \frac{\pi_g(x)}{q(x)}$ via Importance Sampling, where $q(x) = \frac{1}{p} \sum \widehat{\pi}_k(x)$ is the uniform mixture proposal.
5:     $\lambda$-**Step (Adversary):** Update mixture weights:
6:         $\lambda_k \leftarrow \lambda_k \cdot \exp(\eta_\lambda \cdot \ell_k)$ $\{\ell_k$: loss on domain $k\}$
7:     $\mu$-**Step (Constraint):** $\mu \leftarrow \mu + \eta_\mu(\widehat{Z} - 1)$.
8:     $g$-**Step (Gate):** Update $\theta$ to minimize $\mathcal{L}$ via AdamW.
9: **end for**

---

**Theorem 4.2** (Convergence of Primal-Dual Dynamics)**.** *Consider the Lagrangian payoff $\mathcal{L}(g, \lambda, \mu) = \widetilde{L}(\lambda, g) + \mu(Z_g - 1)$. Under the same convexity assumptions as Theorem 4.1, the time-averaged iterates $(\overline{g}_T, \overline{\lambda}_T)$ generated by Algorithm 2 converge to the optimal robust solution with error $O(1/\sqrt{T})$, and the constraint violation decays at rate $O(1/\sqrt{T})$:*

$$\max_\lambda \widetilde{L}(\lambda, \overline{g}_T) - \widetilde{V} \le O\left(\frac{1}{\sqrt{T}}\right) \quad and \quad |Z_{\overline{g}_T} - 1| \le O\left(\frac{1}{\sqrt{T}}\right).$$

*Proof.* The proof relies on viewing the optimization of the Lagrangian $\mathcal{L}(\theta, \lambda, \mu)$ as a zero-sum game between a primal player (controlling $g_\theta$) and a dual player (controlling $\lambda, \mu$). We analyze the convergence using the framework of online convex optimization (OCO) and regret bounds.

**1. The Lagrangian and Duality Gap.**     Recall the Lagrangian of the reformulated game:

$$\mathcal{L}(g, \lambda, \mu) = \widetilde{L}(\lambda, g) + \mu(Z_g - 1) = \sum_{k=1}^p \lambda_k D_{\mathrm{KL}}(\widehat{\mathsf{p}}_k \parallel \pi_g) + \mu\left(\sum_{x \in \mathcal{X}_0} \pi_g(x) - 1\right).$$

This function is convex in the primal variable $g$ (as established in Lemma 3.1 and Section 5.1) and linear (concave) in the dual variables $\lambda, \mu$. Let $w = (\lambda, \mu)$ denote the combined dual variables. The algorithm generates a sequence of iterates

$(g_t, w_t)_{t=1}^T$. We define the *duality gap* for the time-averaged iterates $(\overline{g}_T, \overline{w}_T)$ as:

$$\text{Gap}(\overline{g}_T, \overline{w}_T) = \max_{w \in \mathcal{W}} \mathcal{L}(\overline{g}_T, w) - \min_{g \in \mathcal{G}_1} \mathcal{L}(g, \overline{w}_T),$$

where $\mathcal{W}$ is a compact subset of the dual space containing the optimal dual solution $w^*$.

**2. Regret Decomposition.** A standard result in game dynamics (e.g., Freund & Schapire, 1999; Cesa-Bianchi & Lugosi, 2006) states that the duality gap is bounded by the sum of the average regrets of the players. Let $R_T^g$ be the regret of the $g$-player minimizing $\mathcal{L}(\cdot, w_t)$ and $R_T^w$ be the combined regret of the dual players maximizing $\mathcal{L}(g_t, \cdot)$:

$$\text{Gap}(\overline{g}_T, \overline{w}_T) \le \frac{R_T^g + R_T^w}{T}.$$

**3. Bounding the Regrets.** We analyze the regret for each player based on their specific update rules in Algorithm 2:

- **The $\lambda$-player (Simplex):** Updates $\lambda$ using Exponentiated Gradient (EG). For linear losses with gradients bounded by $M_\lambda$, the regret of EG over the simplex is bounded by:

$$R_T^\lambda \le M_\lambda \sqrt{2T \log p}.$$

- **The $\mu$-player (Scalar Constraint):** Updates $\mu$ using Gradient Ascent (Dual Ascent). Assuming the constraint violation (gradient w.r.t $\mu$) is bounded by $M_\mu = \max_g |Z_g - 1|$ and the optimal $\mu^*$ lies in a bounded range $[-D_\mu, D_\mu]$, standard Gradient Ascent bounds give:

$$R_T^\mu \le D_\mu M_\mu \sqrt{T}.$$

- **The $g$-player (Gate Parameters):** Updates $g$ (via $\theta$) using AdamW (a variant of Online Mirror Descent). Under the convexity assumption of $\widetilde{L}$ w.r.t $g$ and bounded gradients $M_g$, the regret is bounded by:

$$R_T^g \le D_{\mathcal{G}_1} M_g \sqrt{T}.$$

**4. Convergence Rate.** Summing these terms, the total average regret scales as:

$$\frac{R_T^{total}}{T} \le \frac{C\sqrt{T}}{T} = O\left(\frac{1}{\sqrt{T}}\right).$$

Thus, the duality gap decays at a rate of $O(1/\sqrt{T})$.

**5. Recovering the Objectives.** The convergence of the duality gap implies convergence of both the objective value and the constraint satisfaction:

- **Robust Loss:** $\max_\lambda \widetilde{L}(\lambda, \overline{g}_T) - \widetilde{V} \le \text{Gap}(\overline{g}_T, \overline{w}_T) \le O(1/\sqrt{T})$.

- **Constraint Violation:** The Lagrangian term $\mu(Z_g - 1)$ implies that if the constraint is violated ($|Z_{\overline{g}_T} - 1| > \epsilon$), the dual player $\mu$ would exploit this to maximize the gap. Therefore, the constraint violation is also bounded by the gap:

$$|Z_{\overline{g}_T} - 1| \le \frac{\text{Gap}(\overline{g}_T, \overline{w}_T)}{|\mu^*|} \le O\left(\frac{1}{\sqrt{T}}\right).$$

This completes the proof that the Primal-Dual algorithm converges to the optimal robust solution while asymptotically satisfying the normalization constraint. $\square$

# F. Scalable Implementation and Inference

To scale the robust modular framework to high-dimensional generative models such as Transformers, we must address two practical challenges: characterizing the functional form of the gate $g$, and enforcing the global normalization constraint $Z_g = 1$ during stochastic optimization. This section details the system architecture and the Primal-Dual algorithm used to solve the minimax game.

## F.1. Architecture Parameterization

We parameterize the components of the modular system as follows:

**1. The Experts ($\widehat{\pi}_k$):** The ensemble consists of $p$ pre-trained, frozen autoregressive models (e.g., GPT-style Causal Transformers). For a sequence $x = (x_1, \dots, x_T)$, each expert $k$ provides a conditional probability distribution $\widehat{\pi}_k(x_t \mid x_{<t})$. The total log-probability of a sequence is $\log \widehat{\pi}_k(x) = \sum_{t=1}^{T} \log \widehat{\pi}_k(x_t \mid x_{<t})$.

**2. The Gate ($g_\theta$):** Unlike the experts, the gate function is *non-causal*. It observes the entire sequence $x$ to determine the optimal routing weights. We parameterize $g_\theta$ as a **Transformer Encoder** (e.g., BERT-style) with parameters $\theta$.

Here's a mathematical definition of the gate function. We parameterize the gate function $g_\theta$ as a non-causal, bidirectional Transformer Encoder. Unlike the experts, which must be causal to generate text, the gate observes the full input sequence $x = (x_1, \dots, x_T)$ to determine the optimal mixing weights. The computation is defined as follows:

$$
\begin{aligned}
H^{(0)} &= \text{Embed}(x) + \text{PosEnc} && \in \mathbb{R}^{T \times d} \\
H^{(L)} &= \text{TransformerEncoder}_\theta\left(H^{(0)}\right) && \in \mathbb{R}^{T \times d} \\
v &= \text{Pool}(H^{(L)}) = \frac{1}{T} \sum_{t=1}^{T} H_t^{(L)} && \in \mathbb{R}^d \quad \text{(Global Mean Pooling)} \\
w &= W_{\text{out}} v + b_{\text{out}} && \in \mathbb{R}^p \\
g_\theta(x) &= \text{Softmax}(w) && \in \Delta([1, p]).
\end{aligned}
$$

Here, $d$ is the hidden dimension of the gate model, $L$ is the number of encoder layers, and $p$ is the number of experts. The global pooling step aggregates the bidirectional context into a single vector $v$, ensuring that the routing decision $g_\theta(x)$ is based on the entire sequence content.

## F.2. The Stochastic Primal-Dual Algorithm

The algorithm solves the saddle-point problem defined by the Lagrangian:

$$
\min_\theta \max_{\lambda \in \Delta, \mu \in \mathbb{R}} \left[ \sum_{k=1}^{p} \lambda_k \mathcal{L}_{\text{NLL}}(k, \theta) + \mu(Z_{g_\theta} - 1) \right]
$$

**Hyperparameters:**

- $\eta_g$: Learning rate for Gate (e.g., $10^{-4}$, using AdamW).

- $\eta_\lambda$: Learning rate for Adversary (e.g., $0.1$, using SGD/Exponentiated Gradient).

- $\eta_\mu$: Learning rate for Constraint (e.g., $10^{-2}$, using SGD).

- $\alpha$: Moving average factor for estimating global $Z$ (e.g., $0.9$).

**Initialization:**

- Initialize Gate parameters $\theta$.

- Initialize $\log \lambda = [0, \dots, 0]$ (uniform distribution).

---

**Algorithm 3** Stochastic Primal-Dual Training Loop

---

1: **for** iteration $t = 1$ **to** $T$ **do**
2:     **1. Data Sampling:**
3:     Sample a batch $B_k$ of size $M$ from each source dataset $D_k$.
4:     Combine into a super-batch $B = \bigcup_k B_k$ of size $p \times M$.
5:     **2. Forward Pass (Gate & Experts):**
6:     **for** every $x \in B$ **do**
7:         Compute expert log-probs: $L_k(x) = \log \widehat{\pi}_k(x)$ for all $k$.
8:         Compute gate logits $g_\theta(x)$ and weights $w(x) = \text{Softmax}(g_\theta(x))$.
9:         Compute mixture log-prob via LogSumExp:
10:           $\log \pi_g(x) = \text{LogSumExp}_k \left( \log w_k(x) + L_k(x) \right)$.
11:         Compute unnormalized mass density: $m(x) = \exp(\log \pi_g(x))$.
12:     **end for**
13:     **3. Constraint Estimation (Importance Sampling):**
14:     The proposal distribution is the uniform mixture $q(x) = \frac{1}{p} \sum \widehat{\pi}_k(x)$.
15:     Note: Samples $x \in B$ follow the empirical mixture $\frac{1}{p} \sum \widehat{p}_k$.
16:     Assumption: $\widehat{\pi}_k \approx \widehat{p}_k$, so $B$ serves as samples from $q(x)$.
17:     Estimate IS weights: $w_{IS}(x) = \pi_g(x)/q(x)$.
18:     Estimate Z: $\widehat{Z} = \frac{1}{|B|} \sum_{x \in B} w_{IS}(x)$.
19:     Update moving average: $\overline{Z} \leftarrow \alpha \overline{Z} + (1 - \alpha)\widehat{Z}$.
20:     **4. $\lambda$-Player Update (Adversary):**
21:     Calculate loss per domain $k$: $\ell_k = \frac{1}{|B_k|} \sum_{x \in B_k} - \log \pi_g(x)$.
22:     Update $\lambda$ (Exponentiated Gradient):
23:         $\lambda_k \leftarrow \lambda_k \cdot \exp(\eta_\lambda \cdot \ell_k)$.
24:     Renormalize: $\lambda \leftarrow \lambda / \sum_j \lambda_j$.
25:     **5. $\mu$-Player Update (Dual Ascent):**
26:     Goal: Maximize $\mu(\overline{Z} - 1)$.
27:     $\mu \leftarrow \mu + \eta_\mu(\overline{Z} - 1)$.
28:     **6. $g$-Player Update (Primal Minimization):**
29:     Construct Total Loss $\mathcal{J}$:
30:         $\mathcal{J} = \underbrace{\sum_{k=1}^p \lambda_k \ell_k}_{\text{Robust NLL}} + \underbrace{\mu(\widehat{Z} - 1)}_{\text{Lagrangian Penalty}}$
31:     Compute gradients $\nabla_\theta \mathcal{J}$.
32:     Update $\theta$ using Optimizer (AdamW).
33: **end for**

---

- Initialize $\mu = 0$.

- Initialize Global Normalization Estimate $\overline{Z} = 1.0$.

We solve the constrained minimax problem by relaxing the global normalization constraint via a Lagrange multiplier $\mu \in \mathbb{R}$. The objective function is the Lagrangian:

$$\min_\theta \max_{\lambda \in \Delta, \mu \in \mathbb{R}} \mathcal{L}(\theta, \lambda, \mu) = \underbrace{\sum_{k=1}^p \lambda_k \mathop{\mathbb{E}}_{x \sim \widehat{p}_k} \left[ -\log \pi_{g_\theta}(x) \right]}_{\text{Robust NLL}} + \underbrace{\mu \left( \sum_{x \in \mathcal{X}_0} \pi_{g_\theta}(x) - 1 \right)}_{\text{Normalization Penalty}}$$

Algorithm 3 details the stochastic updates. We use three distinct optimizers: **Exponentiated Gradient** for the simplex-constrained adversary $\lambda$, **Dual Ascent** for the constraint $\mu$, and **AdamW** for the gate parameters $\theta$ (see Figure 8).

### F.3. Practical Implementation Details

**Log-Space Stability.** The mixture probability $\pi_g(x) = \sum_k g(x, k)\widehat{\pi}_k(x)$ involves summing probabilities that may be extremely small (e.g., $10^{-100}$ for long sequences). Direct computation leads to underflow. We strictly perform all operations

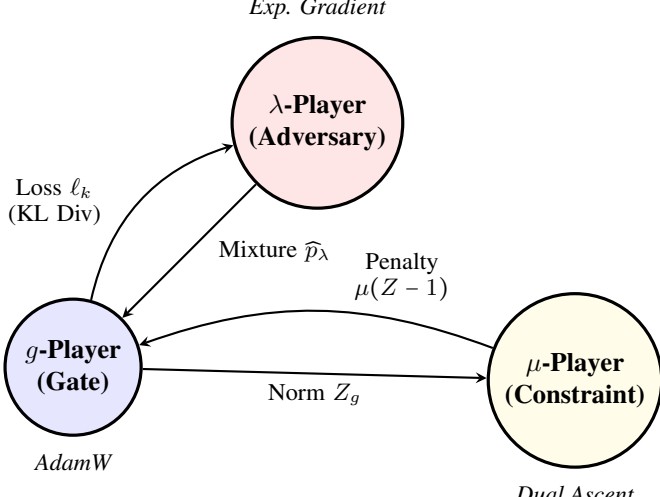

*Figure 8.* Dynamics of the Primal-Dual Game (Algorithm 3). The optimization is modeled as a 3-player game. The $\lambda$-player maximizes the mixture difficulty using Exponentiated Gradient. The $g$-player minimizes the robust loss. The $\mu$-player enforces the global normalization constraint ($Z_g = 1$) via Dual Ascent.

in log-space using the LogSumExp:

$$\log \pi_g(x) = \log \left[ \sum_k \exp \left( \log g(x, k) + \log \widehat{\pi}_k(x) \right) \right].$$

**Estimating the Partition Function** $Z_g$. Calculating the global sum $Z_g = \sum_{x \in \mathcal{X}_0} \pi_g(x)$ exactly is intractable. We rely on a Monte Carlo estimate using the current training batch $B$. To estimate $Z$, consistent with Algorithm 3, we use importance sampling as in Section F.4.1:

$$\widehat{Z} = \frac{1}{|B|} \sum_{x \in B} \frac{\pi_g(x)}{\frac{1}{p} \sum_{k=1}^{p} \widehat{\pi}_k(x)}.$$

To reduce variance in the $\mu$-update, we maintain an Exponential Moving Average (EMA) of the normalization constant $\overline{Z}$. A warm-up period where $\mu$ is fixed to 0 allows the gate to learn discriminative features before the constraint forces the probability mass to contract.

In our implementation, we use the training batch $B$ itself to compute this estimate. The batch $B$ is constructed by sampling uniformly from the source datasets $D_k$, so $x \sim \frac{1}{p} \sum_{k=1}^{p} \widehat{p}_k$. Under the assumption that the pre-trained experts are reasonable approximations of their training data ($\widehat{\pi}_k \approx \widehat{p}_k$), the empirical mixture closely approximates the model mixture proposal $q(x) \approx \frac{1}{p} \sum_{k=1}^{p} \widehat{p}_k$. This allows us to reuse the forward-pass data for the constraint estimation without generating separate synthetic samples from the experts.

**Variance Bound for Partition Function Estimation.** The assumption that the empirical batch approximates the proposal ($q(x) \approx \widehat{p}_{\text{mix}}(x)$) is a natural consequence of the base experts being well-trained density estimators of their respective domains ($\mathrm{D}_{\text{KL}}(\widehat{p}_k \parallel \widehat{\pi}_k) \le \epsilon_k$). The error in estimating $Z_g$ stems from standard Importance Sampling variance. Since our normalized gate space ensures that $g(x, k) \le 1$, the robust model's likelihood is strictly bounded by the envelope $p \cdot q(x)$ (as discussed in Section F.4.2). Thus, the importance weights $w_{IS}(x) = \pi_g(x)/q(x)$ are strictly bounded by $p$. This provides a rigorous formal guarantee that the variance of our Monte Carlo estimator $\widehat{Z}$ is bounded by $\mathcal{O}(p/|B|)$, where $|B|$ is the batch size. In practice, tracking $\widehat{Z}$ via an Exponential Moving Average (EMA) algorithmically smooths out any remaining batch-level variance, ensuring stable Primal-Dual dynamics.

Crucially, this estimator relies on the batch mean to approximate the expectation over $q(x)$, rather than summing over the entire support $\mathcal{X}_0$. This avoids the need to know the total support size $|\mathcal{X}_0|$ (which is intractable for sequence models), making the constraint enforcement computationally feasible.

## F.4. Sampling Algorithms

### F.4.1. THE SIR ALGORITHM

The primary goal in this section is to draw sequence samples from the robust, gated model $\pi_{g^*}$. While our optimization successfully finds a gate $g^*$ that ensures the distribution $\pi_{g^*}(x) = \sum_k g^*(x, k)\hat{\pi}_k(x)$ is globally normalized, the resulting model presents a unique challenge for standard LLM inference. Standard LLMs generate text autoregressively, predicting the next token based solely on past tokens. However, our optimal gate $g^*(x, \cdot)$ is non-causal: it determines the mixture weights based on the complete sequence $x$, meaning the probability of the first token theoretically depends on the last. Since we cannot generate tokens one by one if the routing logic depends on the finished sentence, we must resort to approximation methods like Sampling-Importance-Resampling (SIR) that generate complete candidates first and score them later.

The SIR algorithm derives its name from its three distinct stages, each addressing a specific part of this non-causal hurdle:

1. Sampling: Since we cannot sample directly from the target $\pi_{g^*}$, we first generate a set of $N$ candidate sequences from a *proposal distribution* $q(x)$ that is easy to sample from. We define this proposal as the uniform mixture of our experts, $q(x) = \frac{1}{p}\sum \hat{\pi}_k(x)$. This allows us to use standard autoregressive generation: we simply pick an expert at random and have it generate a full sequence.

2. Importance: We acknowledge that these candidates were drawn from the *wrong distribution* ($q$ instead of $\pi_{g^*}$). To correct for this, we assign an importance weight $w(x)$ to each candidate, calculated as the likelihood ratio $w(x) = \pi_{g^*}(x)/q(x) = \frac{\sum_{k=1}^{p} g^*(x,k)\hat{\pi}_k(x)}{\frac{1}{p}\sum_{k=1}^{p}\hat{\pi}_k(x)}$. This step is computationally expensive but feasible because we are evaluating completed sequences; we can pass the full candidate $x$ to the non-causal gate $g^*$ to compute its true probability under the robust model.

3. Resampling: Finally, to obtain samples that approximate the robust target distribution, we resample from our pool of candidates. A candidate is selected with probability proportional to its importance weight, ensuring that sequences with high probability under the robust model $\pi_{g^*}$ are more likely to be chosen as the final output.

Here's the pseudocode of the SIR algorithm.

The inclusion of the fallback mechanism (Line 19) is a safeguard against numerical instability (underflow), rather than a theoretical necessity. Structurally, the absolute continuity assumption required for SIR consistency is strictly satisfied: since both the target $\pi_{g^*}$ and the proposal $q$ are mixtures of the *same* set of experts $\{\hat{\pi}_k\}$, the support of the target is contained within the support of the proposal ($\text{supp}(\pi_{g^*}) \subseteq \bigcup_k \text{supp}(\hat{\pi}_k) = \text{supp}(q)$). Therefore, the theoretical case where $q(x) = 0$ and $\pi_{g^*}(x) > 0$ (leading to infinite weights) is impossible. Biased uniform sampling is thus only triggered in rare cases of floating-point underflow. The computational cost is concentrated in the weight calculation (Line 11). Evaluating the target density $\pi_{g^*}(x)$ requires a full forward pass of the gate and all $p$ experts for each of the $N$ candidate sequences. Thus, the inference cost scales as $O(Np)$, motivating the need for the efficient distillation methods proposed in Section F.5.

### F.4.2. AN EXACT SAMPLING ALTERNATIVE: REJECTION SAMPLING

While SIR provides asymptotic guarantees, *Rejection Sampling* offers a method to obtain *exact* samples from the robust distribution $\pi_{g^*}$, provided we can strictly bound the ratio between the target and the proposal.

Recall that Rejection Sampling is a fundamental Monte-Carlo technique used to generate observations from a complex target distribution $\pi(x)$ using a simpler, tractable proposal distribution $q(x)$. The main idea is: if we can find a constant $M$ such that the scaled proposal $Mq(x)$ always *envelopes* the target (i.e., $\pi(x) \leq Mq(x)$ for all $x$), we can sample from $q(x)$ and stochastically accept points that fall under the curve of $\pi(x)$. Samples where $M \cdot q(x)$ is much larger than $\pi(x)$ are rejected more frequently, effectively *carving out* the correct distribution from the proposal.

In our framework, we can again use the uniform mixture of experts as the proposal, $q(x) = \frac{1}{p}\sum_{k=1}^{p}\hat{\pi}_k(x)$. A crucial property of our normalized gate space $\mathcal{G}_1$ allows us to derive the strictly required bound $M$. Since the gating weights $g^*(x, k)$ are probabilities bounded by 1, the robust model's likelihood is strictly bounded by the sum of the individual experts:

$$\pi_{g^*}(x) = \sum_{k=1}^{p} g^*(x, k)\hat{\pi}_k(x) \leq \sum_{k=1}^{p}\hat{\pi}_k(x) = p\, q(x).$$

---

**Algorithm 4** Sampling via Sampling-Importance-Resampling (SIR)

---

1: **Input:** Robust gate $g^* \in \mathcal{G}_1$, expert models $\{\widehat{\pi}_k\}_{k=1}^p$, number of candidates $N$.
2: **Initialize:** Empty lists $C \leftarrow []$ (candidates) and $W \leftarrow []$ (weights).
3: {Step 1: Generate N candidates from the proposal $q(x)$.}
4: **for** $i = 1$ **to** $N$ **do**
5:     Sample expert $k \sim \text{Uniform}(\{1, \ldots, p\})$.
6:     Sample sequence $x^{(i)} \sim \widehat{\pi}_k(x)$ (autoregressively).
7:     Append $x^{(i)}$ to $C$.
8: **end for**
9: {Step 2: Compute importance weights.}
10: **for** $i = 1$ **to** $N$ **do**
11:     $x \leftarrow C[i]$.
12:     {This step requires evaluating $g^*$ and all $p$ experts on $x$.}
13:     Compute $\pi_{g^*}(x) = \sum_{k=1}^p g^*(x, k)\widehat{\pi}_k(x)$.
14:     Compute $q(x) = \frac{1}{p}\sum_{k=1}^p \widehat{\pi}_k(x)$.
15:     $w_i \leftarrow \pi_{g^*}(x)/q(x)$ if $q(x) > 0$ else (0 if $\pi_{g^*}(x) = 0$, $\infty$ otherwise). {Handle $q(x) = 0$ case.}
16:     Append $w_i$ to $W$.
17: **end for**
18: {Step 3: Resample one candidate based on weights.}
19: Filter out candidates with non-finite weights. Let indices be $I_{finite}$.
20: Calculate total weight $W_{sum} = \sum_{j \in I_{finite}} W[j]$.
21: **if** $W_{sum} = 0$ **or not** is finite($W_{sum}$) **then** {If all weights are zero or infinite}
22:     Sample $i^*$ uniformly from $\{1, \ldots, N\}$. {Fallback: uniform choice or error}
23: **else**
24:     Define normalized probabilities $P_i = W[i]/W_{sum}$ for $i \in I_{finite}$ (0 otherwise).
25:     Sample an index $i^* \sim \text{Categorical}(\{P_i\}_{i=1}^N)$.
26: **end if**
27: $x^* \leftarrow C[i^*]$.
28: **Return:** The final sample $x^*$.

---

This provides the strict envelope constant $M = p$.

The efficiency of Rejection Sampling is strictly determined by the constant $M = p$, which represents the ratio of the area under the enveloping proposal $Mq(x)$ to the area under the target $\pi_{g^*}(x)$. Geometrically, the algorithm samples points uniformly under the envelope; the probability that such a point also falls under the target curve is exactly $1/M$. Consequently, the number of trials required to find a successful sample follows a geometric distribution with an expected value of $M = p$.

This reveals a clear trade-off: for a moderate number of experts (e.g., $p \leq 10$), the *computational waste* of rejecting candidates is a reasonable price for obtaining unbiased, exact samples. However, because the bound $M = p$ grows linearly with the ensemble size, the acceptance rate $1/p$ drops rapidly. For large $p$ (e.g., $p = 100$), one would discard approximately 99% of generated candidates, making the method prohibitive. In these high-dimensional regimes, SIR becomes the preferred alternative.

### F.5. Baseline: Efficient Sampling via Monolithic Distillation

While the Rejection Sampling and SIR algorithms provide exact or asymptotically exact samples from the robust model $\pi_{g^*}(x)$, their inference cost scales linearly with the number of experts (requiring $\mathcal{O}(p)$ or $\mathcal{O}(Np)$ operations per sample), which may be prohibitive for large-scale deployment. The bottleneck is the non-causal nature of the optimal gate $g^*(x, k)$, which depends on the complete sequence $x$, preventing efficient caching or standard token-by-token generation.

To achieve efficient autoregressive sampling in constant time with respect to $p$, we can distill the robust knowledge from the non-causal target model $\pi_{g^*}$ into a new *causal student model* $\pi_{\text{causal}}$. This student model is parameterized as a standard causal Transformer with parameters $\theta$, ensuring the factorization $\pi_{\text{causal}}(x) = \prod_{t=1}^T \pi_{\text{causal}}(x_t \mid x_{<t})$.

We train the student model by minimizing the Kullback-Leibler divergence from the robust target $\pi_{g^*}$ to the student $\pi_{\text{causal}}$

---

**Algorithm 5** Exact Sampling via Rejection Sampling

---

1: **Input:** Robust gate $g^* \in \mathcal{G}_1$, expert models $\{\widehat{\pi}_k\}_{k=1}^p$.
2: **Output:** An exact sample $x^* \sim \pi_{g^*}$.
3: **loop**
4:     Sample expert $k \sim \mathrm{Uniform}(\{1, \ldots, p\})$.
5:     Sample candidate $x \sim \widehat{\pi}_k(x)$. {Proposal $x \sim q(x)$}
6:     Compute acceptance probability:
$$A(x) = \frac{\pi_{g^*}(x)}{M\, q(x)} = \frac{\pi_{g^*}(x)}{p\, q(x)}$$
7:     Sample $u \sim \mathrm{Uniform}([0, 1])$.
8:     **if** $u \le A(x)$ **then**
9:         **return** $x$
10:    **end if**
11: **end loop**

---

over the space of sequences:

$$\min_\theta \mathrm{D}_{\mathrm{KL}}(\pi_{g^*} \parallel \pi_{\mathrm{causal}})$$

$$= \min_\theta \mathop{\mathbb{E}}_{x \sim \pi_{g^*}}\left[ -\sum_{t=1}^T \log \pi_{\mathrm{causal}}(x_t \mid x_{<t}) \right] - H(\pi_{g^*}).$$

In practice, this is equivalent to maximizing the log-likelihood of the student model on a dataset of synthetic sequences generated from $\pi_{g^*}$. The training procedure is as follows:

1. **Generate Data:** Use the exact Rejection Sampling method (Algorithm 5) or SIR (Algorithm 4) to generate a large dataset of robust sequences $\mathcal{D}_{\mathrm{robust}} = \{x^{(i)}\}_{i=1}^M$ drawn from $\pi_{g^*}$.

2. **Train Student:** Train the causal Transformer $\pi_{\mathrm{causal}}$ on $\mathcal{D}_{\mathrm{robust}}$ using standard cross-entropy loss (next-token prediction).

This distillation step transfers the robustness guarantees of the non-causal gate into the weights of the causal student. At inference time, the expensive ensemble $\pi_{g^*}$ is discarded, and samples are drawn efficiently from $\pi_{\mathrm{causal}}$ using standard autoregressive decoding.

### F.6. Structural Distillation Theory

#### F.6.1. DECOMPOSITION PROOF

**Proposition 6.1** (Decomposition of Structural Distillation). *Minimizing the sequence-level divergence $\mathrm{D}_{\mathrm{KL}}(\pi_{g^*} \parallel \pi_\gamma)$ is equivalent to maximizing the expected log-likelihood of the student model on trajectories sampled from the robust teacher. Specifically, the gradient is:*

$$\nabla_\phi \mathcal{J}(\phi) = -\mathop{\mathbb{E}}_{x \sim \pi_{g^*}}\left[ \sum_{t=1}^T \nabla_\phi \log\left( \sum_{k=1}^p \gamma_\phi(x_{<t}, k)\, \widehat{\pi}_k(x_t \mid x_{<t}) \right) \right].$$

*Proof.* We expand the definition of the KL divergence:

$$\mathcal{J}(\phi) = \mathop{\mathbb{E}}_{x \sim \pi_{g^*}}\left[ \log \pi_{g^*}(x) \right] - \mathop{\mathbb{E}}_{x \sim \pi_{g^*}}\left[ \log \pi_\gamma(x) \right].$$

The first term is the negative entropy of the teacher distribution $\pi_{g^*}$ and is fixed. Thus, minimizing the KL divergence is equivalent to maximizing the second term. Unlike the teacher, the student model $\pi_\gamma$ is defined to be causal and autoregressive. Therefore, its log-probability factors into a sum of conditional log-probabilities:

$$\log \pi_\gamma(x) = \sum_{t=1}^T \log \pi_\gamma(x_t \mid x_{<t}).$$

Substituting this back into the expectation yields:

$$\max_{\phi} \ \mathbb{E}_{x \sim \pi_{g^*}} \left[ \sum_{t=1}^{T} \log \left( \sum_{k=1}^{p} \gamma_{\phi}(x_{<t}, k) \, \widehat{\pi}_k(x_t \mid x_{<t}) \right) \right].$$

Taking the gradient gives the result. □

### F.6.2. THEORETICAL ANALYSIS OF DISTILLATION

To establish a rigorous theoretical footing for structural distillation (Section 6), here, we analyze the divergence between the distribution induced by the non-causal teacher, $\pi_{g^*}$, and the causal student, $\pi_{\gamma}$. We explicitly decompose this error into the sum of step-wise divergences between the student router and the optimal Bayesian posterior of the teacher, proving that there is no irreducible structural mismatch.

**Definitions and Model Classes.** Let $\mathcal{X}$ be the token vocabulary and $\mathcal{X}^T$ be the space of trajectories.

The Teacher (Mixture of Products). The robust gate $g^* \in \mathcal{G}_1$ defines a mixture over expert trajectories. The likelihood of a sequence $x$ is:

$$\pi_{g^*}(x) = \sum_{k=1}^{p} g^*(x, k) \underbrace{\left( \prod_{t=1}^{T} \widehat{\pi}_k(x_t \mid x_{<t}) \right)}_{\text{Expert } k \text{ trajectory}}.$$

This represents a *Mixture of Products*. The latent expert choice $k$ is sampled once per sequence, maintaining mode consistency (e.g., sticking to one domain for the whole sentence).

The Student (Product of Mixtures). The causal router $\gamma_{\phi}$ defines a distribution where mixing happens at every step $t$:

$$\pi_{\gamma}(x) = \prod_{t=1}^{T} \underbrace{\left( \sum_{k=1}^{p} \gamma_{\phi}(x_{<t}, k) \, \widehat{\pi}_k(x_t \mid x_{<t}) \right)}_{\text{Step-wise mixture}}.$$

This represents a *Product of Mixtures*. The effective expert weight $\gamma$ can change at every token.

**The Bayes-Optimal Causal Router.** We do not merely assume a *good* router exists. Instead, we derive the optimal causal policy $\gamma^*$ that minimizes the approximation error to the teacher.

**Proposition F.1** (The Posterior Mean Router). *For any history $h = x_{<t}$, the optimal causal routing weights $\gamma_k^*(h)$ are given by the* posterior probability *of expert $k$ given the history, under the teacher distribution $\pi_{g^*}$:*

$$\gamma_k^*(x_{<t}) = P_{\pi_{g^*}}(K = k \mid x_{<t}) = \frac{\mathbb{E}_{x' \sim \pi_{g^*}} \left[ \mathbb{I}[x'_{<t} = x_{<t}] \cdot g^*(x', k) \right]}{\pi_{g^*}(x_{<t})}.$$

*Proof.* The student model is a product of mixtures: $\pi_{\gamma}(x_t|h) = \sum_k \gamma_k(h)\widehat{\pi}_k(x_t|h)$. The teacher model, despite being non-causal in parameterization, implies a valid marginal conditional distribution:

$$\pi_{g^*}(x_t \mid h) = \sum_{k=1}^{p} P_{\pi_{g^*}}(k \mid h) \, \widehat{\pi}_k(x_t \mid h).$$

By setting $\gamma_k^*(h) = P_{\pi_{g^*}}(k \mid h)$, the student's conditional distribution becomes identical to the teacher's conditional distribution at every step. Thus, this choice of $\gamma^*$ is optimal (achieving zero local divergence). □

**Exact Decomposition of the Distillation Error.** We now provide an exact decomposition of the total distillation error $D_{\mathrm{KL}}(\pi_{g^*} \parallel \pi_{\gamma})$ using the chain rule of relative entropy. This replaces heuristic approximations with a rigorous bound.

**Theorem F.2** (Exact Chain Rule Decomposition). *Let $\pi_{\gamma}$ be the student model parameterized by $\phi$. The total divergence decomposes exactly into a sum of step-wise divergences:*

$$D_{\mathrm{KL}}(\pi_{g^*} \parallel \pi_{\gamma}) = \sum_{t=1}^{T} \mathbb{E}_{x_{<t} \sim \pi_{g^*}} \left[ D_{\mathrm{KL}}(\pi_{g^*}(\cdot \mid x_{<t}) \parallel \pi_{\gamma}(\cdot \mid x_{<t})) \right].$$

*Furthermore, this error is upper-bounded by the divergence between the routing policies:*

$$\mathsf{D}_{\mathrm{KL}}(\pi_{g^*} \parallel \pi_\gamma) \le \sum_{t=1}^{T} \mathop{\mathbb{E}}_{x_{<t} \sim \pi_{g^*}} [\mathsf{D}_{\mathrm{KL}}(\gamma^*(\cdot \mid x_{<t}) \parallel \gamma_\phi(\cdot \mid x_{<t}))].$$

*Proof.* The first equality is the standard Chain Rule for Kullback-Leibler divergence applied to autoregressive sequence models.

For the inequality, recall that the conditional distributions are mixtures: $P(\cdot|h) = \sum_k \gamma_k^*(h)\widehat{\pi}_k(\cdot|h)$ and $Q(\cdot|h) = \sum_k \gamma_{\phi,k}(h)\widehat{\pi}_k(\cdot|h)$. By the joint convexity of the KL divergence, $\mathsf{D}_{\mathrm{KL}}(\sum_k \lambda_k P_k \parallel \sum_k \mu_k P_k) \le \mathsf{D}_{\mathrm{KL}}(\lambda \parallel \mu)$. Applying this to our mixtures:

$$\mathsf{D}_{\mathrm{KL}}\left(\sum_k \gamma_k^*\widehat{\pi}_k \parallel \sum_k \gamma_{\phi,k}\widehat{\pi}_k\right) \le \mathsf{D}_{\mathrm{KL}}(\gamma^* \parallel \gamma_\phi).$$

Summing this bound over all time steps $t$ completes the proof. □

**Interpretation.** This theorem clarifies that there is no irreducible "structural mismatch" error ($\mathcal{E}_{struct} = 0$) because the teacher's distribution is perfectly realizable by a causal product of mixtures using the posterior weights $\gamma^*$. The total error is driven entirely by the *Router Approximation Error*: the inability of the parameterized router $\gamma_\phi$ (e.g., a small Transformer) to perfectly match the complex posterior distribution $\gamma^*$ induced by the non-causal gate.

**Consistency of the Algorithm.** Finally, we confirm that the standard distillation objective minimized by Algorithm 6 is equivalent to minimizing the router approximation error.

**Corollary F.3.** *Minimizing the sequence-level objective $\mathcal{J}(\phi) = \mathsf{D}_{\mathrm{KL}}(\pi_{g^*} \parallel \pi_\gamma)$ is equivalent to minimizing the expected step-wise divergence between the true posterior router $\gamma^*$ and the student router $\gamma_\phi$.*

*Proof.* From Theorem F.2, the total divergence is exactly the sum of expected local divergences:

$$\mathsf{D}_{\mathrm{KL}}(\pi_{g^*} \parallel \pi_\gamma) = \sum_{t=1}^{T} \mathop{\mathbb{E}}_{x_{<t} \sim \pi_{g^*}} [\mathsf{D}_{\mathrm{KL}}(\gamma^*(\cdot \mid x_{<t}) \parallel \gamma_\phi(\cdot \mid x_{<t}))].$$

The terms $\gamma^*(\cdot|x_{<t})$ are fixed targets derived from the teacher. Therefore, gradient descent on the global objective $\mathcal{J}(\phi)$ directly minimizes the discrepancy between the student's routing decisions and the optimal Bayesian update at every time step. □

### F.7. Efficient Structural Distillation via Cached Logits (Algorithm 6)

---

**Algorithm 6** Efficient Structural Distillation via Cached Logits

---

1: **Input:** Robust gate $g^*$, frozen experts $\{\widehat{\pi}_k\}$, dataset size $M$, router $\gamma_\phi$.
2: **Phase 1: Data Generation & Caching**
3: Generate $M$ sequences $\{x^{(i)}\}$ from $\pi_{g^*}$ using Rejection Sampling (Alg. 5) or SIR.
4: Initialize dataset $\mathcal{D} \leftarrow \varnothing$.
5: **for** each sequence $x^{(i)}$ and time step $t$ **do**
6:     Run all $p$ experts to get next-token probabilities:
7:     $p_{t,k}^{(i)} = \widehat{\pi}_k(x_t^{(i)} \mid x_{<t}^{(i)})$ for $k \in \{1, \ldots, p\}$.
8:     Store tuple $(x_{<t}^{(i)}, x_t^{(i)}, \mathbf{p}_t^{(i)})$ in $\mathcal{D}$. $\{\mathbf{p}_t^{(i)}$ is a vector of size $p\}$
9: **end for**
10: **Phase 2: Router Training**
11: **repeat**
12:     Sample batch of tuples $(h, y, \mathbf{p})$ from $\mathcal{D}$. $\{h$: history, $y$: target token$\}$
13:     Compute router weights: $\mathbf{w} = \gamma_\phi(h) \in \Delta([1, p])$.
14:     Compute mixture probability: $P_{\mathrm{mix}} = \mathbf{w} \cdot \mathbf{p} = \sum_{k=1}^{p} w_k p_k$.
15:     Compute Loss: $\mathcal{L} = -\log(P_{\mathrm{mix}})$.
16:     Update $\phi \leftarrow \phi - \eta\nabla_\phi\mathcal{L}$.
17: **until** Convergence
18: **Output:** Causal Router $\gamma_\phi$.

---

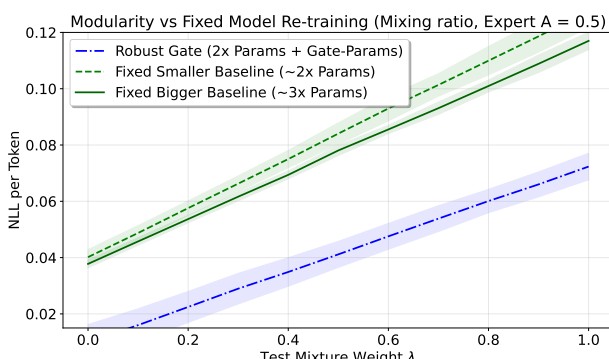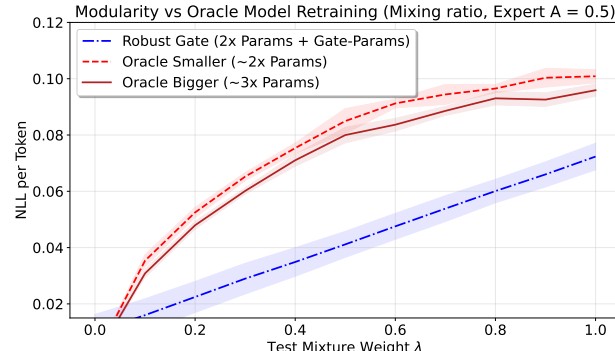

*Figure 9.* **Modularity overcomes gradient conflict at a 50-50% mix of Domain A and a pure Domain B.** Left figure illustrates a comparison to the Fixed Smaller and Larger models (in Green), while the right figure illustrates the comparison to the Oracle models (in Red). The Robust Gate (blue) and the Fixed models in the left figure naturally obtain the best performance for small values of $\lambda$ where the test distribution is predominantly made up by Domain B. As $\lambda$ increases, the test distribution contains more data from Domain A and gets harder for all models. Yet the Robust Gate maintains its clear advantage. The Oracle models in the right figure still has an advantage for the really skewed distribution and $\lambda \sim 0$, but loses to the Robust Gate for larger values of $\lambda$.

### F.8. Discussion

We have presented a hierarchy of sampling strategies for the robust gated model, establishing a trade-off between theoretical exactness, inference latency, and modularity.

**Exactness vs. Efficiency.** The sampling-based methods (SIR and Rejection Sampling) provide the strongest theoretical guarantees. As $N \to \infty$, SIR recovers the exact robust distribution $\pi_{g^*}$, and Rejection Sampling provides exact samples for any $N$. These methods ensure that the worst-case performance bound ($D_{\mathrm{KL}} \le \max \epsilon_k$) established in Theorem 3.3 holds precisely. However, the computational cost of evaluating all $p$ experts for every candidate sample is often prohibitive for real-time applications.

**The Role of Distillation.** The distillation approaches (Sections F.5 and F.7) bridge the gap between theory and practice. By compressing the non-causal knowledge of $g^*$ into a causal student model, we recover standard autoregressive inference speeds. This comes at the cost of introducing a distillation error, $D_{\mathrm{KL}}(\pi_{g^*} \parallel \pi_{\mathrm{student}})$, which represents the loss in robustness due to approximation.

**Modularity and Structural Distillation.** Standard causal distillation results in a monolithic student model, discarding the modular nature of the original experts. In contrast, the Structural Distillation method preserves the pre-trained experts, learning only a lightweight routing policy. This maintains the system's adaptability (if an expert is improved, the overall system improves without full retraining) while significantly reducing the inference overhead compared to the raw non-causal gate. This structural approach represents the most promising direction for deploying robust, modular generative models at scale.

## G. Experimental Details and Additional Results

In this appendix, we provide the detailed experimental setups and additional results regarding domain overlap that complement the main analysis in Section 7.

### G.1. Additional Experimental Results: Robustness to Domain Overlap

We further carried out experiments where the distributions of our two experts A and B were less contradictory. We did so by mixing up Domain A to have a fraction of Domain B. We experimented with fractions of zero (the experiment described in the main text with 'clean' distributions), a fraction of 0.5 and a fraction of 0.75, at which point Domain A only contains 25% of its original data.

The two additional experiments are illustrated in Figure 9 and Figure 10. The Oracle models maintain their advantage for very skewed test distributions, but the Robust Gate model demonstrates the best performance for most test distributions, despite having fewer total parameters than the larger-sized models.

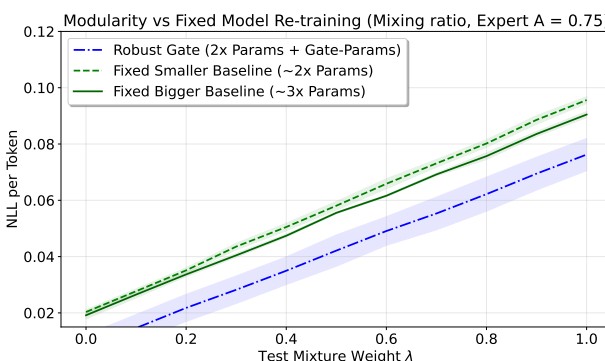 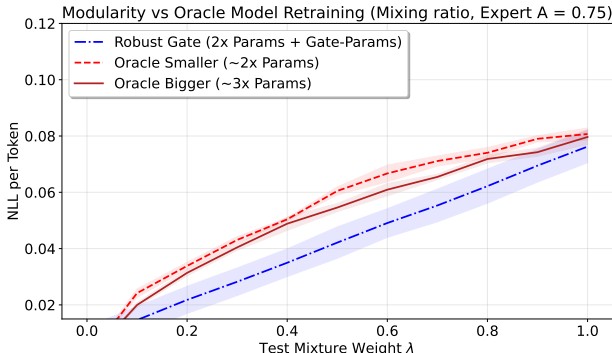

*Figure 10.* **Modularity overcomes gradient conflict at a 25-75% mix of Domain A and Domain B.** Left figure illustrates a comparison to the Fixed Smaller and Larger models (in Green), while the right figure illustrates the comparison to the Oracle models (in Red). The Robust Gate (blue) and the Fixed models in the left figure naturally obtain the best performance for small values of $\lambda$ where the test distribution is predominantly made up by Domain B. As $\lambda$ increases, the test distribution contains more data from Domain A and gets harder for all models. Yet the Robust Gate maintains its clear advantage. Results are illustrated with lines for the mean values over 5 runs and standard deviations indicated with shaded regions. The Oracle models in the right figure still has an advantage for the really skewed distribution and $\lambda \sim 0$, but loses to the Robust Gate for larger values of $\lambda$.

### G.2. Detailed Experimental Setup: Synthetic Data

All models are standard Transformer Encoders (masked for autoregression) with $L = 1$ layer, $H = 2$ attention heads, and feedforward dimension $d_{\text{ff}} = 32$. The different-sized models are obtained by varying the embedding dimension, which is $d = 6$ for the Robust Gate model, $d = 8$ for the basic Expert, $d = 16$ for the Smaller-Retrained model and $d = 20$ for the Larger-Retrained model. Since various parts of the transformers scale either linearly or quadradically (like the feedforward network), the number of parameters do not exactly scale linearly in the embedding dimension. Two expert models plus the small gate match approximately the smaller-retrained model, while the larger retrained model has approximately $1.5\times$ the number of parameters as the combined Robust Gate.

For these experiments, the vocabular size is 100, the sequence length is $T = 10$ and batch size is $B = 64$. Optimization uses AdamW with $\beta_1 = 0.9, \beta_2 = 0.999$ and zero dropout. The experts and baselines are trained for 800 steps with learning rate $\eta = 10^{-2}$. The Robust Gate is trained for 800 steps with $\eta_{\text{gate}} = 5 \times 10^{-3}$, and the dual variables are updated with $\eta_\lambda = 0.2, \eta_\mu = 0.1$. The partition function $Z$ is estimated using a running exponential moving average ($\alpha = 0.9$) for variance reduction. The training set size for each expert was 800 batches of 64 examples or 51,000 examples. Both the gate and the smaller and larger models were trained with the union of 102,000 examples.

We evaluate performance across the full spectrum of distribution shifts by varying the mixture weight $\lambda \in [0, 1]$ in steps of 0.1 in the test distribution $p_\lambda(x) = \lambda p_A(x) + (1 - \lambda)p_B(x)$. That is, $\lambda = 0$ means all the test data comes from Domain B. We illustrate the NLL per token across the mixture range. For all figures, we provide mean values over 5 runs and indicate the standard deviation with shaded regions.

For structural distillation, the Causal Router was implemented with the same transformer architecture as already described. Learning rate and number of training steps are as for the baseline models.

### G.3. Detailed Experimental Setup: Real-World Data

We trained 3 experts each on 80K sequences of length 128 on these dataset using the `gpt2` tokenizer. The experts were chosen as two-layer transformers with 2 heads and an embedding dimension of 256, providing them with about 6.5M parameters. The gate model was implemented as a 2-headed 2-layered transformer with an internal dimension of 256. It has just about 290K parameters, making the combined Robust Gate of size 20M parameters. In comparison, we trained a 19.8M parameter model, a transformer with 4 heads, 3 layers and an internal dimension of 184. The Gate model and the Retrained model were trained on the union of the dataset. All models were tested on a hold out sample of 20K sequences.

The learning rate for the AdamW optimizer was set to $1e - 4$ for the experts, the gate, and the retrained model. For the gate, the additional learning parameters were set to $\eta_\lambda = 0.05, \eta_\mu = 0.02$ and $\alpha = 0.9$.

## G.4. Real-World Robustness to Distribution Shift

To further assess the stability of the method presented in Section 7.4, we tested the models on different compositions of the test data. In Table 2 we provide the results from testing the Retrained and the Gate model on these distributions. The distributions are characterized by $\lambda$-test, with $(1/3, 1/3, 1/3)$ corresponding to the uniform distribution. The order of the distributions is (1) `wikimedia/wikipedia`: High-quality factual prose; (2) `bigcode/the-stack-smol`: Source code across 30+ languages; (3) `fineweb-edu`: Filtered high-quality educational web content.

The performance of the models naturally varies with the distribution, but the Gate model is more robust against these changes and systematically exhibits a lower NLL loss.

*Table 2.* Robustness test results for different test distributions. The results are mean values ± 1 standard deviation, obtained over 5 runs with different initialization of the model training.

| $\lambda$-test | Retrained, NLL ± std.dev. | Gate, NLL ± std.dev. |
|---|---|---|
| 1/3, 1/3, 1/3 | 5.133 ± 0.010 | 4.994 ± 0.013 |
| 1/3, 1/2, 1/6 | 5.190 ± 0.080 | 5.068 ± 0.014 |
| 1/6, 1/3, 1/2 | 5.226 ± 0.011 | 5.099 ± 0.014 |
| 1/2, 1/3, 1/6 | 5.042 ± 0.011 | 4.890 ± 0.005 |
| 1/2, 1/6, 1/3 | 5.298 ± 0.006 | 5.117 ± 0.005 |
| 1/3, 1/6, 1/2 | 5.363 ± 0.054 | 5.187 ± 0.057 |
| 1/6, 1/2, 1/3 | 5.279 ± 0.017 | 5.181 ± 0.020 |

