# OpenReview forum: "A Theoretical Framework for Modular Learning of Robust Generative Models"
_ICML.cc/2026/Conference — ICML 2026 regular_

### Official Review · Reviewer_a84P · 2026-03-13

**Soundness:** 4
**Presentation:** 4
**Significance:** 4
**Originality:** 3
**Overall Recommendation:** 6
**Confidence:** 3

**Summary:**

The paper studies how to combine pretrained domain experts robustly into a single generative model. The authors suggest a gating mechanism motivated by the game-theoretic minimax formulation, analyze the failure mode of monolithic training (via JSD diversity terms) and propose a primal-dual algorithm + sampling and distillation to a causal router. Experiments on synthetic tasks and real-world datasets show improved robustness over monolithic training.

**Compliance With Llm Reviewing Policy:**

Affirmed.

**Final Justification:**

The authors addressed the suggestions I raised in my review. My overall recommendation has not changed: I continue to strongly support the paper

**Key Questions For Authors:**

None

**Limitations:**

yes

**Strengths And Weaknesses:**

Soundness. The authors provide a very comprehensive theoretical justification for their approach. The key objectives are clearly defined, main results are stated precisely with the proofs moved to the appendix. The paper addresses main practical concerns: non-convex gating, normalization via primal-dual relaxation, and non-causal sampling, by discussing theory/practice mismatches and suggesting inference and distillation procedures. Empirically, the claims are supported by both synthetic and real-world experiments that make the results consistent and convincing.

Presentation. I do like the presentation of the paper. The problem is well motivated, all the theoretical statements are formulated exactly in the main text with links to the Appendix for the full proof. In the main text, the authors provide significant intuition and proof sketches for the main results. The paper is coherent: most of the concerns are addressed by the authors in the next paragraph after they arise (e.g. about sampling complexity, or "Gap between Theory and Practice" paragraph). Plots could be larger (it is hard to distinguish different line types), but overall all of the experiment results are well-illustrated.

Significance. This work is highly significant for modern LLM training, where mixture of experts systems are widely used. It identifies a failure mode of monolithic training over the mixture of datasets and provides a game-theoretic framework for robust gating. The paper provides the practical mechanisms for scalability and inference (e.g. primal-dual optimization, structural distillation to a causal router). The experiments in the paper demonstrate consistent gains in both synthetic and small-scale real-world settings. To strengthen the impact, one could make the experiments with larger number of experts or on larger datasets, but this would require more significant computation resources.

Originality. The individual ingredients are not new (minimax training, mixture of experts, importance sampling, distillation). The novelty is in the way they put together into a coherent robustness framework. It has a clean minimax formulation, KL-divergence analysis over the mixture of domains, interpretable analysis of monolithic vs modular approaches, and end-to-end recipe that connects theory to practice via primal-dual optimization and structural distillation into a causal router.

---

> ### Author Rebuttal · Authors · 2026-03-29
>
> We sincerely thank the reviewer for the "Strong Accept" and for your highly positive evaluation. We are thrilled that you found the paper to be technically flawless, coherent, and highly significant to modern LLM training. We also appreciate your recognition of the originality in combining minimax training and structural distillation.
>
> **Presentation: Plots could be larger (it is hard to distinguish different line types), but overall all of the experiment results are well-illustrated.**
>
> **Response:** Thank you for this helpful feedback. In our camera-ready version, we will re-generate all plots to significantly increase the font size of the legends and axis labels, and we will apply thicker line styles with distinct markers to ensure they are perfectly legible, even when printed in grayscale.
>
> **Significance: To strengthen the impact, one could make the experiments with larger number of experts or on larger datasets, but this would require more significant computation resources.**
>
> **Response:** We completely agree that larger-scale experiments strengthen the impact. Encouraged by your feedback, we allocated additional computational resources during the rebuttal to scale our framework to the 1B+ parameter regime and added an empirical comparison against standard MoE models trained from scratch (see our response to Reviewer vPkw regarding W2).
>
> We evaluated our framework using an ensemble of massive experts (~1.6B parameters each) pre-trained on different domains. The general setup matches the setup from Table 1 (Diff seed), but the architecture was scaled heavily: each expert uses a GPT-2 XL-style configuration (48 layers, 1600 embedding dimension). To ensure a strictly fair comparison, the retrained monolithic baseline model was symmetrically scaled to a massive 4.9B parameter architecture (48 layers, 2832 embedding dimension) to exactly match the combined capacity of the three experts plus the gate model. The Robust Gate, which is highly parameter-efficient at only ~2.5M parameters, successfully learned to route across these massive experts without optimization instability, achieving a test NLL of 4.734 $\pm$ 0.006 compared to the monolithic baseline's 4.829 $\pm$ 0.012. This confirms that our framework successfully scales to modern massive LLMs, decoupling geometric interference and yielding a significant performance improvement of 0.095.
>
> **Summary of Impact and Reproducibility:** We believe this work provides deep theoretical insights for a greener, more principled, and mathematically grounded framework for training LLMs, eliminating the need for heuristic data weighting. During the rebuttal, we further validated these theoretical guarantees at scale, demonstrating significant performance improvements against a 4.9B parameter monolithic baseline. Finally, we are deeply committed to reproducibility. Our manuscript provides exhaustive algorithmic details (including complete pseudocode for the Primal-Dual optimization and Cached-Logit Distillation) and exact hyperparameter configurations to ensure independent replication. Furthermore, we intend to open-source our training framework and pre-trained weights upon publication, pending final institutional review.

---

> > ### Author Rebuttal · Reviewer_a84P · 2026-04-04
> >
> > Thanks for the response! All my concerns were addressed

---

### Official Review · Reviewer_HD3y · 2026-03-13

**Soundness:** 2
**Presentation:** 3
**Significance:** 2
**Originality:** 3
**Overall Recommendation:** 3
**Confidence:** 3

**Summary:**

The paper proposes a theoretical framework for combining pre-trained generative models (experts) using a gating mechanism. The goal is to achieve robustness across unknown mixtures of data domains. The authors formulate this as a minimax game. They seek a single robust gate that minimizes the KL divergence to the worst-case data mixture. The paper provides existence proofs using Kakutani's fixed-point theorem. It also provides generalization bounds for the gate. The authors theoretically compare this modular approach to monolithic retraining. They highlight a "JSD gap" where monolithic models suffer from geometric interference. Algorithmically, the authors propose a stochastic primal-dual algorithm to handle the global normalization constraint. They also introduce a structural distillation method for efficient autoregressive inference. Experiments on synthetic and small-scale real-world datasets (Wikipedia, Code, FineWeb) support the claims.

**Compliance With Llm Reviewing Policy:**

Affirmed.

**Key Questions For Authors:**

- Could the authors evaluate the method on models larger than 20M parameters? Even a 1B~2B parameter test would make the LLM claims much more credible.
- Can you explain the massive variance ($0.788$) for the Fine Web Expert in Table 1?

**Limitations:**

The authors could evaluate the method on modern LLMs that are larger than 1B

**Strengths And Weaknesses:**

**Strengths:**This paper presents a rigorous theoretical framework for robust modular generative modeling. The authors prove that modular gating can overcome the inteference barrier of monolithic models, guided by the JSD. Furthermore, the paper offers both a scalable primal-dual optimization algorithm and a structural distillation method for practical inference

**Weaknesses:**

1. The experiments are conducted only on small-scale models with approximately 20M parameters, which are too small. This scale does not substantiate the introductory claims regarding LLMs
2. The estimation of the partition function $Z$ relies on the training batch (Alg 3). This assumes the empirical mixture closely approximates the model mixture proposal. This assumption is strong and lacks rigorous error bounds.

---

> ### Author Rebuttal · Authors · 2026-03-29
>
> We deeply appreciate your review and your recognition of the paper's merits. We are glad you found our theoretical framework rigorous, and that you appreciated the mathematical proofs guided by the JSD, along with our scalable primal-dual optimization algorithm.
>
> **Weaknesses:**
>
> **1. The experiments are conducted only on small-scale models with approximately 20M parameters, which are too small. This scale does not substantiate the introductory claims regarding LLMs (Limitations: The authors could evaluate the method on modern LLMs that are larger than 1B.)**
>
> **Response:** We completely agree that empirical validation on 1B+ parameter models makes our claims much stronger. During the rebuttal period, we significantly scaled our experiments to base expert models exceeding 1B parameters.
>
> We evaluated our framework using an ensemble of massive experts (~1.6B parameters each) pre-trained on different domains. The general setup matches the setup from Table 1 (Diff seed), but the architecture was scaled heavily: each expert uses a GPT-2 XL-style configuration (48 layers, 1600 embedding dimension). To ensure a strictly fair comparison, the retrained monolithic baseline model was symmetrically scaled to a massive 4.9B parameter architecture (48 layers, 2832 embedding dimension) to exactly match the combined capacity of the three experts plus the gate model. The Robust Gate, which is highly parameter-efficient at only ~2.5M parameters, successfully learned to route across these massive experts without optimization instability, achieving a test NLL of 4.734 $\pm$ 0.006 compared to the monolithic baseline's 4.829 $\pm$ 0.012. This confirms that our framework successfully scales to modern massive LLMs, decoupling geometric interference and yielding a significant performance improvement of 0.095.
>
> **2. The estimation of the partition function Z relies on the training batch (Alg 3). This assumes the empirical mixture closely approximates the model mixture proposal. This assumption is strong and lacks rigorous error bounds.**
>
> **Response:** We thank the reviewer for pointing this out. The assumption that the empirical batch approximates the proposal ($q(x) \approx \hat{p} _{mix}(x)$) is a natural consequence of the base experts being well-trained density estimators of their respective domains (i.e., $D _{KL}(\hat{p} _k || \hat{\pi} _k) \le \epsilon_k$).
>
> The error in estimating $Z$ stems from standard Importance Sampling variance. Since our normalized gate space ensures that $g(x, k) \le 1$, the robust model's likelihood is strictly bounded by the envelope $p \cdot q(x)$ (as discussed in Sec 5). Thus, the importance weights $w(x) = \pi_{g}(x)/q(x)$  are strictly bounded by $p$. This provides a rigorous formal guarantee that the variance of our Monte Carlo estimator $\hat{Z}$ is bounded by $\mathcal{O}(p/|B|)$, where $|B|$ is the batch size.
>
> In practice, tracking $Z$ via an Exponential Moving Average (EMA) algorithmically smooths out any remaining batch-level variance, ensuring stable Primal-Dual dynamics. We will include this formal variance bound and error analysis in Appendix F.2.
>
> **Questions:**
>
> **1. Could the authors evaluate the method on models larger than 20M parameters? Even a 1B~2B parameter test would make the LLM claims much more credible.**
>
> **Response:** Please see our response to Weakness 1.
> Our new 1B+ parameter experiments successfully confirm our claims and demonstrate that the JSD gap decoupling scales perfectly to massive LLMs, yielding a performance improvement of 0.095 over the monolithic baseline.
>
> **2. Can you explain the massive variance (0.788) for the Fine Web Expert in Table 1?**
>
> **Response:** We apologize for any confusion and respectfully clarify a slight misreading of Table 1. The massive variance of $0.788$ under the "Diff data" column actually corresponds to the Code Expert ($5.267 \pm 0.788$), whereas the Fine Web Expert exhibited a very low variance of $0.006$ ($5.623 \pm 0.006$). A massive variance for the Code Expert is empirically expected: code datasets are highly heterogeneous, with significant structural and syntactic differences across varying programming languages, repositories, and paradigms. Consequently, evaluating the Code Expert across different unseen splits naturally leads to a more volatile NLL. Crucially, despite this inherent instability in the monolithic code domain, our Gate Model robustly routes inputs to the correct experts, smoothing out these domain-specific instabilities and dropping the total variance substantially to $0.141$ ($5.087 \pm 0.141$). We will clarify this domain-specific variance finding in the text.

---

> > ### Author Rebuttal · Reviewer_HD3y · 2026-04-04
> >
> > Thanks for the authors' response, my initial concerns are addressed.

---

> > > ### Author Response · Authors · 2026-04-07
> > >
> > > Dear Reviewer HD3y,
> > >
> > > Thank you for reviewing our rebuttal and confirming that your initial concerns have been fully resolved. We truly appreciate your constructive feedback, especially the push to evaluate our method on 1B+ parameter LLMs and to formalize the variance bounds, which has significantly strengthened our work!
> > >
> > > Given your confirmation that these primary issues have been addressed, we hope our response now aligns with your expectations for acceptance. Thank you again for your time, your thorough engagement, and your insightful suggestions.
> > >
> > > Best,
> > > The Authors

---

### Official Review · Reviewer_vPkw · 2026-03-13

**Soundness:** 3
**Presentation:** 3
**Significance:** 3
**Originality:** 3
**Overall Recommendation:** 4
**Confidence:** 2

**Summary:**

This paper introduces a theoretical framework for combining pre-trained generative models into a robust modular system. By formulating gating as a minimax game, the authors mathematically prove that modularity overcomes monolithic geometric interference. Additionally, they provide a Primal-Dual algorithm and Structural Distillation for scalable optimization and efficient inference.

**Compliance With Llm Reviewing Policy:**

Affirmed.

**Key Questions For Authors:**

1. How does the proposed frozen-expert gating framework compare empirically to standard Mixture-of-Experts models trained from scratch with load-balancing losses on the same data union?

2. The paper notes that the efficiency of Rejection Sampling drops rapidly as the number of experts grows, and the inference cost for Sampling-Importance-Resampling scales linearly. How can the generation of the robust synthetic dataset remain computationally feasible for Structural Distillation when scaling to larger ensembles?

**Limitations:**

Yes.

**Strengths And Weaknesses:**

## Strengths:
1. This paper provides rigorous mathematical proofs for robust gating and a generalization bound.

2. This paper uses Jensen-Shannon Divergence to explain why monolithic models suffer from geometric interference on diverse tasks.

3. This paper introduces a Primal-Dual algorithm and Structural Distillation to resolve computational and inference bottlenecks.
## Weaknesses:
1. Experiments rely on small-scale models (~20M parameters), leaving performance on massive LLMs unverified.

2. The empirical validation compares against retrained and oracle baselines, but lacks comparison to standard Mixture-of-Experts (MoE) or modern model merging techniques.

---

> ### Author Rebuttal · Authors · 2026-03-29
>
> We sincerely thank the reviewer for the positive assessment and for recognizing the theoretical rigor of our mathematical proofs. We are encouraged that you appreciated our novel use of the Jensen-Shannon Divergence to explain geometric interference, and valued the practical scalability of our Primal-Dual algorithm and Structural Distillation method.
>
> **Response to W1:** We completely agree that verifying our theoretical framework on massive LLMs significantly strengthens the paper's practical impact. To address this, we scaled our experiments and applied our modular framework to a 1B+ parameter model setup for the rebuttal.
>
> We evaluated our framework using an ensemble of massive experts (~1.6B parameters each) pre-trained on different domains. The general setup matches the setup from Table 1 (Diff seed), but the architecture was scaled heavily: each expert uses a GPT-2 XL-style configuration (48 layers, 1600 embedding dimension). To ensure a strictly fair comparison, the retrained monolithic baseline model was symmetrically scaled to a massive 4.9B parameter architecture (48 layers, 2832 embedding dimension) to exactly match the combined capacity of the three experts plus the gate model. The Robust Gate, which is highly parameter-efficient at only ~2.5M parameters, successfully learned to route across these massive experts without optimization instability, achieving a test NLL of 4.734 $\pm$ 0.006 compared to the monolithic baseline's 4.829 $\pm$ 0.012. This confirms that our framework successfully scales to modern massive LLMs, decoupling geometric interference and yielding a significant performance improvement of 0.095.
>
> **Response to W2:** We appreciate this highly constructive suggestion. To provide a comprehensive empirical validation, we have implemented a standard token-level Mixture-of-Experts (MoE) baseline, trained from scratch on the union of the datasets and guided by a standard load-balancing loss.
>
> The general setup matches the setup from Table 1 (Diff seed), but is modified to use a standard token-level Mixture-of-Experts (MoE) architecture. The standard MoE was trained from scratch on the uniform mixture of the datasets, using shared attention layers and a token-level routing layer over three expert Feed-Forward Networks, guided by a standard load-balancing loss. To ensure parameter-matching fairness, it was given the same parameter capacity (~20M) as our combined modular framework. The standard MoE model achieved an NLL of 5.08 $\pm$ 0.009. As theoretically predicted in Section 3.2, joint training over conflicting domains induces gradient interference even in standard MoE architectures due to their shared representation layers and joint optimization. In contrast, our frozen-expert routing approach effectively mitigated this conflict, achieving an NLL of 4.994 $\pm$ 0.013 and outperforming the standard MoE trained from scratch by a margin of 0.086.
>
> Regarding modern model merging techniques (e.g., Model Soups, Task Arithmetic), we note that these methods produce a single, static set of weights that inevitably suffers from the geometric interference bound highlighted in Theorem B.2. Our dynamic gating overcomes this interference by probabilistically selecting experts per input.
>
> **Response to Q1:** Please see our response to W2 above. Empirically, our frozen-expert gate achieves an NLL of 4.994 $\pm$ 0.013, compared to 5.08 $\pm$ 0.009 for the standard MoE trained from scratch. Theoretically, a standard MoE trained from scratch still minimizes a monolithic ERM objective over the mixture. Its shared routing and attention layers are susceptible to capacity collapse on disjoint domains (the "JSD gap"). Our modular framework mathematically minimizes the divergence to the worst-case data mixture using completely independent, frozen pre-trained specialists, successfully decoupling the risk from geometric interference.
>
> **Response to Q2:** This is an insightful question regarding practical use. While Rejection Sampling drops in efficiency as $p$ grows, generating the robust synthetic dataset is strictly a one-time, offline process that is "embarrassingly parallelizable" across multiple GPUs. Crucially, as proven by our generalization bound (Theorem 3.3), the sample complexity scales with the lightweight Causal Router (the gate) and not the complex experts, meaning we only need to generate a relatively small dataset to distill the robust policy.
>
> Furthermore, to make the subsequent training of the student model computationally feasible, we use the Cached-Logit Distillation technique (Algorithm 6). During the one-time generation phase, we evaluate the experts and cache their output probability vectors $P_t$. The lightweight causal student router ($\gamma_\phi$) is then trained iteratively entirely on these cached logits, bypassing the need for repeated, expensive forward passes through the experts during the distillation optimization loop. We will clarify these offline compute strategies in Sec 6.3.

---

> > ### Author Rebuttal · Reviewer_vPkw · 2026-04-06
> >
> > I will keep the score.

---

### Decision · Program_Chairs · 2026-04-30

**Decision:**

Accept (regular)

**Comment:**

This paper studies combining specialized models, from a perspective of facing a potentially worst-case mixture of the components. While in an average-case scenario a classical mixture may be sufficient, the worst-case perspective offers an opportunity to go outside the mixture class (analogous to improper learning). In particular, the paper considers gated models, where the mixing coefficient depends on the instance. This is very close to some classical theory (multiple source domain adaptation), but with specialization to generative models which raises, among other things, the challenge of the overall model needing to be normalized. The core of the results are based on the fact that the geometry of KL divergence allows gated models to exit the convex hull of the components to become "simultaneously" close to all of them, thus achieving gains over a classical mixture (though these gains depend on the geometry of the components and their diversity.)

The reviewers assess this paper's contribution favorably. The mathematical presentation could use some polish. For example, certain terminologies (e.g., LSE) are used before they are defined and the paper has multiple forward references that require reading back-and-forth. The results appear sound and novel, and the authors do a very good job in delineating exactly when and how much gain can be expected theoretically. More clarification could be given when going from theory to implementation, e.g. the question of estimating the partition function should be better discussed in the paper, just as the authors explain in the rebuttal. The place where the paper could use most improvement is in the experiments, which all reviewers deem to be of rather small scale and missing anticipated comparisons (particularly to the other methods that were referenced and contrasted to in the first part of the paper.) In the rebuttal, the authors offer some such comparison, which should ideally be further extended and included. If there are limitations to scaling (e.g., synthetic data generation), which prevents such additions, that should also be carefully addressed.

In summary, the paper proposes an insightful angle to this problem and offers both analytical and methodological contributions that would be of interest to the community. That said, polishing the mathematical presentation, addressing implementation details, and revamping the experiments will help better appreciate the results.